# Classification Under Misspecification: Halfspaces, Generalized Linear Models, and Evolvability

**Sitan Chen**
MIT
sitanc@mit.edu

**Frederic Koehler**
MIT
fkoehler@mit.edu

**Ankur Moitra**
MIT
moitra@mit.edu

**Morris Yau**
UC Berkeley
morrisyau@berkeley.edu

## Abstract

In this paper, we revisit the problem of distribution-independently learning halfspaces under Massart noise with rate $\eta$. Recent work [DGT19] resolved a long-standing problem in this model of efficiently learning to error $\eta+\epsilon$ for any $\epsilon > 0$, by giving an improper learner that partitions space into $\text{poly}(d, 1/\epsilon)$ regions. Here we give a much simpler algorithm and settle a number of outstanding open questions:

(1) We give the first *proper* learner for Massart halfspaces that achieves $\eta + \epsilon$.

(2) Based on (1), we develop a blackbox knowledge distillation procedure to convert an arbitrarily complex classifier to an equally good proper classifier.

(3) By leveraging a simple but overlooked connection to *evolvability*, we show any SQ algorithm requires super-polynomially many queries to achieve $\text{OPT} + \epsilon$.

We then zoom out to study generalized linear models and give an efficient algorithm for learning under a challenging new corruption model generalizing Massart noise.

Lastly, we empirically evaluate our algorithm for Massart halfspaces and find it exhibits some intriguing fairness properties.

## 1   Introduction

A central challenge in theoretical machine learning is to design learning algorithms that are provably robust to noise. We will focus on supervised learning problems, where we are given samples $(\mathbf{X}, Y)$ where the distribution on $\mathbf{X}$ is arbitrary and the label $Y$ is chosen to be either $+1$ or $-1$ according to some unknown ground truth function. We will then allow an adversary to tamper with our samples in various ways. We will be particularly interested in the following models:

(1) Halfspaces: $Y = \text{sgn}(\langle \mathbf{w}^*, \mathbf{X} \rangle)$ for some unknown vector $\mathbf{w}^*$.

(2) Generalized Linear Models: $Y \in \{\pm 1\}$ is a random variable with conditional expectation

$$\mathbb{E}[Y \mid \mathbf{X}] = \sigma(\langle \mathbf{w}^*, \mathbf{X} \rangle)$$

where the *link function* $\sigma$ is odd, monotone, and $L$-Lipschitz.

It is well-known that without noise there are simple, practical, and provable algorithms that work in the PAC learning model [Val84]. For example, the perceptron algorithm [Ros58] learns halfspaces with margin, and the Isotron algorithm [KS09] is an elegant generalization which learns generalized linear models (GLMs), even when $\sigma$ is unknown.

There are many natural models for noise. While it is tempting to gravitate towards the most general models, it turns out that they often make the learning problem computationally hard. For example, we could allow the adversary to change the labels however they want, and ask to find a hypothesis with nearly the best possible agreement over the class. This is called the *agnostic learning* model [Hau92, KSS94]; Daniely [Dan16] recently showed it is computationally hard to learn halfspaces in

this model, even weakly. At the other end of the spectrum, there are overly simplistic models of noise: for example, we could assume that each label is randomly flipped with fixed probability $\eta < 1/2$. This is called the *random classification noise* (RCN) model [AL88]. Bylander [Byl94] and Blum et al. [BFKV98] gave the first algorithms for learning halfspaces under random classification noise. By now, there is a general understanding of how to accommodate such noise (even in generalized linear models, where it can be embedded into the link function), usually by modifying the surrogate loss that we are attempting to minimize.

The modern goal is to find a delicate balance of making the noise model as flexible, expressive, and realistic as possible while maintaining learnability. To that end, the Massart noise model [Slo88, Coh97, MN$^+$06] seems to be an appealing compromise. In this setting, we fix a noise level $\eta < 1/2$ and each sample $(\mathbf{X}, Y)$ has its label flipped independently with some probability $\eta(\mathbf{X}) \leq \eta$.

There are a few interpretations of this model. First, it lets the adversary add *less* noise at a point $\mathbf{x}$ than RCN would. It may seem surprising that this (seemingly helpful!) change can actually break algorithms which work under RCN. However, this is exactly the problem with assuming a fixed noise rate – algorithms that work under RCN are almost always overtuned to this specific noise model.

Second, Massart noise can be interpreted as allowing an adversary[1] to control a random $\eta$ fraction of the examples. Thus it circumvents a major source of hardness in the agnostic learning model – an adversary can no longer control *which* points it gets to corrupt. A natural scenario in which this type of corruption can manifest is crowdsourced data labeling. In such settings, it is common for a central agency to randomly distribute the work of labeling data to a group of trustworthy "turks"– users, federal agencies, etc. However, if a fraction of the turks are untrustworthy, perhaps harboring prejudices and biases, they can try to degrade the learner's performance by injecting adversarially chosen labels for the random subset of examples they are assigned.

Recently, Diakonikolas, Goulekakis and Tzamos [DGT19] resolved a long-standing open question and gave the first efficient algorithm for learning halfspaces under Massart noise, over any distribution on $\mathbf{X}$. However it has some shortcomings. First it is improper: rather than outputting a single halfspace, it outputs a partition of the space into polynomially many regions and a potentially different halfspace on each one. Second, it only achieves classification error $\eta + \epsilon$, though it may be possible to achieve $\mathsf{OPT} + \epsilon$ error, where $\mathsf{OPT}$ is the misclassification error of the best halfspace. Indeed, stronger accuracy guarantees are known under various distributional assumptions [ABHU15, ZLC17, YZ17, ABHZ16, DKTZ20a].

## 1.1 Our Results

In this work we resolve many of the outstanding problems for learning under Massart noise. In doing so, we also make new connections to other concepts in computational learning theory, in particular to Valiant's model of evolvability [Val09]. First, we give the first proper learning algorithm for learning a halfspace under Massart noise without distributional assumptions.

**Theorem 1.1** (Informal, see Theorem C.18). *For any $0 \leq \eta < 1/2$, let $\mathcal{D}$ be a distribution over $(\mathbf{X}, Y)$ given by an $\eta$-Massart halfspace, and suppose $\mathbf{X}$ is supported on vectors of bit-complexity at most $b$. There is an algorithm which runs in time $\mathrm{poly}(d, b, 1/\epsilon)$ and sample complexity $\widetilde{O}(\mathrm{poly}(d, b)/\epsilon^3)$ and outputs a classifier $\mathbf{w}$ whose 0-1 error over $\mathcal{D}$ is at most $\eta + \epsilon$.*

In fact, when the margin is at least inverse polynomial in the dimension $d$, our algorithm is particularly simple. As with all of our proper learning algorithms, it is based on a new and unifying minimax perspective for learning under Massart noise. While the ideal minimax problem is computationally and statistically intractable to solve, we show that by restricting the power of the max player, we get a nonconvex optimization problem such that: (1) any point with sufficiently small loss achieves the desired $\eta + \epsilon$ error guarantee, and (2) gradient descent successfully finds such a point.

An attractive aspect of our formalism is that it is modular: by replacing different building blocks in the algorithm, we can arrive at new guarantees. For example, in the non-margin case we develop a cutting-plane based proper learner that improves the $\widetilde{O}(\mathrm{poly}(d, b)/\epsilon^5)$ sample complexity of the

improper learner of [DGT19] to $\widetilde{O}(\mathrm{poly}(d,b)/\epsilon^3)$ (see Theorem C.18). If we know the margin is only polynomially small, we can swap in a different strategy for the max player and improve the dependence on $d$ and $b$ (Theorem C.15). If we know the underlying halfspace is sparse, we can swap in a mirror descent strategy for the min player and obtain sample complexity depending only logarithmically on $d$ (Theorem C.12). For the dependence on $\epsilon$, note that there is a lower bound of $\Omega(1/\epsilon^2)$ which holds even for random classification noise (see e.g. [MN+06]).

The above result shows that an improper hypothesis is not needed to obtain the guarantees of [DGT19]. In fact, this underlies a more general phenomena: using our proper learner, we develop a blackbox *knowledge distillation* procedure for Massart halfspaces. This procedure converts any classifier, possibly improper and very complex, to a proper halfspace with equally good prediction accuracy.

**Theorem 1.2** (Informal, see Theorem D.3). *Let $\mathcal{D}, b$ be as in Theorem 1.1. There is an algorithm which, given query access to a possibly improper hypothesis $\mathbf{h}$ and $\widetilde{O}(\mathrm{poly}(d,b)/\epsilon^4)$ samples, runs in time $\mathrm{poly}(d,b,1/\epsilon)$ and outputs a proper classifier $\mathbf{w}$ whose 0-1 error over $\mathcal{D}$ exceeds that of $\mathbf{h}$ by at most $\epsilon$. If the underlying halfspace has a margin $\gamma$, there is an algorithm that achieves this but only requires $\widetilde{O}(d/\gamma^2\epsilon^4)$ samples and runs in near-linear time.*

This is surprising as many existing schemes for knowledge distillation in practice (e.g. [HVD14]) require non-blackbox access to the teacher hypothesis. Combining our reduction with known *improper* learning results, we can establish a number of new results for *distribution-dependent* proper learning of Massart halfspaces, e.g. achieving error $\mathsf{OPT} + \epsilon$ over the hypercube $\{\pm1\}^n$ for any fixed $\epsilon > 0$ (see Theorem D.5). We will return to this problem in a bit.

Theorem 1.2 tells us that if we are given access to a "teacher" hypothesis achieving error $\mathsf{OPT} + \epsilon$, we can construct a halfspace with equally good accuracy. Is the teacher necessary? To answer this, we study the problem of achieving $\mathsf{OPT} + \epsilon$ error under Massart noise (without distributional assumptions). Here we make a simple, but previously overlooked, connection to the concept of evolvability in learning theory [Val09]. An implication of this connection is that we automatically can give new evolutionary algorithms, resistant to a small amount of "drift", by leveraging existing distribution-dependent algorithms for learning under Massart noise (see Remark E.9); this improves and extends some of the previous results in the evolvability literature.

The main new implication of this connection, leveraging previous work on evolvability [Fel08], is the first lower bound for the problem of learning Massart halfspaces to error $\mathsf{OPT} + \epsilon$. In particular, we prove super-polynomial lower bounds in the statistical query (SQ) learning framework [KS94]:

**Theorem 1.3** (Informal, see Theorem E.1). *Any SQ algorithm for distribution-independently learning halfspaces to error $\mathsf{OPT} + o(1)$ under Massart noise, requires a super-polynomial number of queries.*

We remark that the SQ framework captures all known algorithms for learning under Massart noise, and the lower bound applies to both proper and improper learning algorithms. Actually, the proof of Theorem 1.3 gives a super-polynomial SQ lower bound for the following natural setting: learning an affine hyperplane over the uniform distribution on the hypercube $\{\pm1\}^d$. Combined with existing works [ABHU15, ZLC17, YZ17, ABHZ16, DKTZ20a], which show polynomial runtime is achievable for e.g. log-concave measures and the uniform distribution on the sphere, we now have a reasonably good understanding of when $\mathsf{OPT} + \epsilon$ error and $\mathrm{poly}(1/\epsilon, d)$ runtime is and is not achievable.

Having resolved the outstanding problems on Massart halfspaces, we move to a more conceptual question: *Can we learn richer families of hypotheses in challenging Massart-like noise models?* In particular, we study generalized linear models (as defined earlier). Unlike halfspaces, these models do not assume that the true label is a deterministic function of $\mathbf{X}$. Rather its expectation is controlled by an odd, monotone, and Lipschitz function that is otherwise arbitrary and unknown and depends upon a projection of the data along an unknown direction. This is a substantially richer family of models, capturing both halfspaces with margin and other fundamental models like logistic regression. In fact, we also allow an even more powerful adversary – one who is allowed to move the conditional expectation of $Y$ in both directions, either further from zero, or, up to some budget $\zeta > 0$, closer to or even *past* zero (see Remark B.3 for a discussion of how this generalizes Massart halfspaces). This can be thought of as taking one more step towards the agnostic learning model (in fact, if $\sigma = 0$ it captures a noisy version of agnostic learning, see Remark B.4), but in a way that still allows for meaningful learning guarantees. We give the first efficient algorithm that works in this setting:

**Theorem 1.4** (Informal, see Theorem F.1). *Let $\sigma : \mathbb{R} \to [-1, 1]$ be any odd, monotone, L-Lipschitz function. For any $\epsilon > 0$ and $0 \le \zeta < 1/2$, there is a polynomial-time algorithm which, given $\mathrm{poly}(L, \epsilon^{-1}, (\zeta \vee \epsilon)^{-1})$ samples from a $\zeta$-misspecified GLM with link function $\sigma$ and true direction $\mathbf{w}^*$, outputs an improper classifier $\mathbf{h}$ whose 0-1 error over $\mathcal{D}$ satisfies*

$$\operatorname*{err}_{\mathcal{X}}(\mathbf{h}) \le \frac{1 - \mathbb{E}_{\mathcal{D}}[\sigma(|\langle \mathbf{w}^*, \mathbf{x} \rangle|)]}{2} + \zeta + \epsilon.$$

When $\zeta = 0$, we can further find a proper halfspace achieving this in time polynomial on an appropriate notion of inverse margin (Theorem F.14), generalizing our result for Massart halfspaces.

Finally in Section G we study our algorithm for learning Massart halfspaces on both synthetic and real data. In the synthetic setting, we construct a natural example where the input distribution is a mixture of two gaussians but the error rates on the two components differ in a way that biases logistic regression to find spurious directions that are close to orthogonal to the true direction used to label the data. In real data we study how varying the noise rates across different demographic groups in the UCI Adult dataset [DG17] can sometimes lead off-the-shelf algorithms to find undesirable solutions that disadvantage certain minority groups. This variation could arise in many ways and there are numerous empirical studies of this phenomenon in the social sciences [Goy19, OW11, LEP+13]. For example, in microcredit systems, certain groups might have lower levels of trust in the system, leading to higher levels of noise in what they self-report [Gui94, Gal97, KRZ14]. It can also come about in models where agents are allowed to manipulate their inputs to a classifier at a cost, but different groups face different costs [HIV19]. In fact we show a rather surprising phenomenon that adding larger noise outside the target group can lead to much worse predictions on the target group in the sense that it leads certain kinds of algorithms to amplify biases present in the data. In contrast, we show that our algorithm, by virtue of being tolerant to varying noise rates across the domain, are able to be simultaneously competitive in terms of overall accuracy and yet avoid the same sorts of adverse effects for minority groups.

## 1.2 Technical Preliminaries

**Generative Models and Notation**    Throughout, given a distribution $\mathcal{D}$ over labeled examples, we will let $(\mathbf{X}, Y)$ and $(\mathbf{x}, y)$ respectively denote the random variable given by $\mathcal{D}$, and a deterministic point in the domain of $\mathcal{D}$. First recall the usual setting of classification under Massart noise.

**Definition 1.5** (Classification Under Massart Noise). *Fix noise rate $0 \le \eta < 1/2$ and domain $\mathcal{X}$. Let $\mathcal{D}_\mathbf{x}$ be an arbitrary distribution over $\mathcal{X}$. Let $\mathcal{D}$ be a distribution over pairs $(\mathbf{X}, Y) \in \mathcal{X} \times \{\pm 1\}$ given by the following generative model. Fix an unknown function $f : \mathcal{X} \to \{\pm 1\}$. Ahead of time, an adversary chooses a quantity $0 \le \eta(\mathbf{x}) \le \eta$ for every $\mathbf{x}$. Then to sample $(\mathbf{X}, Y)$ from $\mathcal{D}$, $\mathbf{X}$ is drawn from $\mathcal{D}_\mathbf{x}$, and $Y = f(\mathbf{X})$ with probability $1 - \eta(\mathbf{X})$, and otherwise $Y = -f(\mathbf{X})$. We will refer to the distribution $\mathcal{D}$ as* arising from concept $f$ with $\eta$-Massart noise.

*In the special case where $\mathcal{X}$ is a Hilbert space and $f(\mathbf{x}) \triangleq \mathrm{sgn}(\langle \mathbf{w}^*, \mathbf{x} \rangle)$ for some unknown $\mathbf{w}^* \in \mathcal{X}$, we will refer to the distribution $\mathcal{D}$ as* arising from an $\eta$-Massart halfspace.

Given $\mathbf{w}$ and $\mathbf{x} \in \mathcal{X}$, let $\mathrm{err}_\mathbf{x}(\mathbf{w}) = \eta(\mathbf{x})$ denote the probability over the Massart-corrupted response $Y$ that $\mathrm{sgn}(\langle \mathbf{w}, \mathbf{x} \rangle) \ne Y$. Given $\mathcal{X}' \subseteq \mathcal{X}$, denote the *zero-one error* of $\mathbf{w}$ over $\mathcal{X}'$ by $\mathrm{err}_{\mathcal{X}'}(\mathbf{w}) \triangleq \mathrm{Pr}_{\mathcal{D}}[\mathrm{sgn}(\langle \mathbf{w}, \mathbf{X} \rangle) \ne Y \mid \mathbf{X} \in \mathcal{X}']$. For any $\mathbf{h}(\cdot) : \mathcal{X}' \to \{\pm 1\}$, we will also overload notation by defining $\mathrm{err}_{\mathcal{X}'}(\mathbf{h}) \triangleq \mathrm{Pr}_{\mathcal{D}}[\mathbf{h}(\mathbf{X}) \ne Y \mid \mathbf{X} \in \mathcal{X}']$. If $\mathbf{h}(\cdot)$ is a *constant* classifier which assigns the same label $s \in \{\pm 1\}$ to every element of $\mathcal{X}'$, then we refer to the zero-one error of $\mathbf{h}(\cdot)$ by $\mathrm{err}_{\mathcal{X}'}(s) \triangleq \mathrm{Pr}_{\mathcal{D}}[Y \ne s \mid \mathbf{X} \in \mathcal{X}']$. When working with a set of samples from $\mathcal{D}$, we will use $\widehat{\mathrm{err}}$ to denote the empirical version of err.

Given $\lambda \ge 0$, let $\ell_\lambda(\mathbf{w}, \mathbf{x}) \triangleq \mathbb{E}_{\mathcal{D}}[\mathrm{LeakyRelu}_\lambda(-Y\langle \mathbf{w}, \mathbf{X} \rangle) \mid \mathbf{X} = \mathbf{x}]$, where $\mathrm{LeakyRelu}_\lambda(z)$ is defined to be $(1 - \lambda)z$ if $z \ge 0$ and $\lambda z$ otherwise. Observe that $\mathrm{LeakyRelu}_\lambda$ is convex for all $\lambda \le 1/2$. Similar to [DGT19], we will work with the convex proxy for 0-1 error given by $L_\lambda(\mathbf{w}) \triangleq \mathbb{E}_{\mathcal{D}_\mathbf{x}}[\ell_\lambda(\mathbf{w}, \mathbf{X})]$. We will frequently condition on the event $\mathbf{X} \in \mathcal{X}'$ for some subsets $\mathcal{X}' \subseteq \mathcal{X}$. Let $L_\lambda^{\mathcal{X}'}$ denote the corresponding loss under this conditioning.

We will consider the following extension of Massart halfspaces.

**Definition 1.6** (Misspecified Generalized Linear Models). *Fix $0 \le \zeta < 1/2$ and Hilbert space $\mathcal{X}$. Let $\sigma : \mathbb{R} \to [-1, 1]$ be any odd, monotone, L-Lipschitz function, not necessarily known to the learner.*

Let $\mathcal{D}_{\mathbf{x}}$ be any distribution over $\mathcal{X}$ supported on the unit ball.[2] Let $\mathcal{D}$ be a distribution over pairs $(\mathbf{X}, Y) \in \mathcal{X} \times \{\pm 1\}$ given by the following generative model. Fix an unknown $\mathbf{w}^* \in \mathcal{X}$. Ahead of time, a $\zeta$-misspecification adversary chooses $\delta : \mathcal{X} \to \mathbb{R}$ for which

$$-2\zeta \le \delta(\mathbf{x})\mathrm{sgn}(\langle \mathbf{w}^*, \mathbf{x} \rangle) \le 1 - |\sigma(\langle \mathbf{w}^*, \mathbf{x} \rangle)|$$

for all $\mathbf{x} \in \mathcal{X}$. Then to sample $(\mathbf{X}, Y)$ from $\mathcal{D}$, $\mathbf{X}$ is drawn from $\mathcal{D}_{\mathbf{x}}$, and $Y$ is sampled from $\{\pm 1\}$ so that $\mathbb{E}[Y \mid \mathbf{X}] = \sigma(\langle \mathbf{w}^*, \mathbf{X} \rangle) + \delta(\mathbf{X})$. We will refer to such a distribution $\mathcal{D}$ as arising from an $\zeta$-misspecified GLM with link function $\sigma$.

We emphasize that the case of $\zeta = 0$ is already nontrivial as the adversary can decrease the noise level arbitrarily at any point; in particular, the $\eta$-Massart adversary in the halfspace model can be equivalently viewed as a 0-misspecification adversary for the link function $\sigma(z) = (1 - 2\eta)\mathrm{sgn}(z)$. While this is not Lipschitz, in the case that the halfspace has a $\gamma$ margin we can make it $O(1/\gamma)$-Lipschitz by making the function linear on $[-\gamma, \gamma]$, turning it into a "ramp" activation [KS09]. The $\zeta = 0$ GLM model is also a special case of the Tsybakov noise model [T$^+$04].

Given i.i.d. samples from $\mathcal{D}$, a learner $\mathcal{A}$'s goal is to output $\mathbf{h} : \mathcal{X} \to \{\pm 1\}$ for which $\mathrm{err}_{\mathcal{X}}(\mathbf{h})$ is as small as possible, with high probability. We say $\mathcal{A}$ is *proper* if $\mathbf{h}$ is given by $\mathbf{h}(\mathbf{x}) \triangleq \langle \widehat{w}, \mathbf{x} \rangle$.

## 2 Properly Learning Halfspaces Under Massart Noise

**An Idealized Zero-Sum Game.** Our proper learning algorithms are all based on the framework of finding approximately optimal strategies for the following zero-sum game:

$$\min_{\|\mathbf{w}\| \le 1} \max_{c:\mathcal{X} \to \mathbb{R}_{\ge 0}} \mathbb{E}[c(\mathbf{X})\ell_\lambda(\mathbf{w}, \mathbf{X})] \tag{1}$$

where $\lambda$ is a fixed parameter chosen slightly larger than $\eta$, and $c(\mathbf{X})$ can be any measurable, nonnegative function such that $\mathbb{E}[c(\mathbf{X})] = 1$. We can think of the $\min$ player as the classifier, whose goal is to output a halfspace $\mathbf{w}$ with loss almost as small as the ground truth halfspace $\mathbf{w}^*$. On the other hand, the $\max$ player is a special kind of *discriminator* whose goal is to prove that $\mathbf{w}$ has inferior predictive power compared to $\mathbf{w}^*$, by finding a reweighting of the data such that $\mathbf{w}$ performs very poorly in the LeakyRelu loss. This is based on the fact that in the Massart model, for *any* reweighting $c(\mathbf{X})$ of the data, $\mathbf{w}^*$ performs well in the sense that $\mathbb{E}[c(\mathbf{X})\ell_\lambda(\mathbf{w}^*, \mathbf{X})] < 0$.

However, directly solving this minimax problem is statistically and computationally intractable. The key to our approach is to fix alternative strategies for the $\max$ player. We then let the $\mathbf{w}$ player play against this adversary and update their strategy in a natural way (e.g. gradient descent), and analyze the resulting dynamics. Thus, our framework naturally yields simple, practical learning algorithms. We note that a similar zero-sum game based approach was used in concurrent work of [DKTZ20b] for a different problem, learning halfspaces under the more general Tsybakov noise model [T$^+$04] with distributional assumptions on $\mathbf{X}$ in quasipolynomial time, and a very recent polynomial time version in the same setting [DKK$^+$20].

**Remark 2.1.** *We briefly explain the approach of [DGT19] in the context of (1) and why their approach only yields an improper learner. In the first step of the algorithm, they minimize the LeakyRelu loss over the entire space (i.e. take $c(\mathbf{X}) = 1$). They show this generates a $\mathbf{w}$ with good zero-one loss on a subset of space $S$, fix this hypothesis on $S$, and then restrict to $\mathcal{X} \setminus S$ (i.e., take $c(\mathbf{X}) = \frac{\mathbb{1}[\mathbf{X} \notin S]}{\Pr[\mathbf{X} \notin S]}$) and restart their algorithm. Because they fix $c$ before minimizing over $\mathbf{w}$, their first step is minimizing a fixed convex surrogate loss. However, by Theorem 3.1 of [DGT19], no proper learner based on minimizing a fixed surrogate loss will succeed in the Massart setting. In contrast, our algorithms choose $c$ adversarially based on $\mathbf{w}$ and thus evade the lower bound of [DGT19].*

**Algorithm and Analysis** For clarity of exposition, we focus here on halfspaces with a margin; we will show in Section C.2 of the supplement how standard techniques allow us to extend to the general case [Coh97, BFKV98]. Our proper learner is based upon the following upper bound on (1):

$$\min_{\|\mathbf{w}\| \le 1} \max_{r > 0} \mathbb{E}[\ell_\lambda(\mathbf{w}, \mathbf{X}) \mid |\langle \mathbf{w}, \mathbf{X} \rangle| \le r], \tag{2}$$

where $r$ will be restricted so that $\Pr[|\langle \mathbf{w}, \mathbf{X} \rangle| \leq r] \geq \epsilon$ for some small $\epsilon > 0$. By (greatly) restricting the possible strategies for the discriminator to "slabs" along the direction of $\mathbf{w}$, we completely fix the problem of intractability for the max-player. In particular, the optimization problem over $r > 0$ is one-dimensional, and the expectation can be accurately estimated from samples.

---

**Algorithm 1:** FINDDESCENTDIRECTION$(\mathbf{w}, \epsilon, \delta, \lambda)$

---

1 Form $\hat{\mathcal{D}}$ from $m = O(\log(2/\delta)/\epsilon^3 \gamma^2)$ samples, let $\hat{L}_\lambda$ denote LeakyRelu loss w.r.t. $\hat{\mathcal{D}}$.
2 $R \leftarrow \{r > 0 : \Pr_{\hat{\mathcal{D}}}[\mathbf{X} \in \mathcal{S}(\mathbf{w}, r)] \geq \epsilon\}$.
3 $r^* \leftarrow \operatorname{argmax}_{r \in R} \hat{L}_\lambda^{\mathcal{S}(\mathbf{w}, r)}(\mathbf{w})$.
4 **return** $\mathbf{g} = \nabla \hat{L}_\lambda^{\mathcal{S}(\mathbf{w}, r^*)}(\mathbf{w})$.

---

---

**Algorithm 2:** FILTERTRON$(\epsilon, \eta, \delta, \lambda, T)$

---

1 Let $\mathbf{w}_1$ be an arbitrary vector in the unit ball.
2 Build an empirical distribution $\hat{\mathcal{H}}$ from $m = \Omega(\log(T/\delta)/\epsilon^2)$ samples (to use as a test set).
3 **for** $t = 1$ *to* $T$ **do**
4      **if** $\widehat{\operatorname{err}}(\mathbf{w}_t) < \eta + \epsilon/2$ **then return** $\mathbf{w}_t$.
5      **else**
6          $\mathbf{g}_t \leftarrow$ FINDDESCENTDIRECTION$(\mathbf{w}_t, \epsilon/6, \delta/2T, \lambda)$.
7          $\mathbf{w}_{t+1} \leftarrow \frac{\mathbf{w}_t - \beta_t \mathbf{g}_t}{\max(1, \|\mathbf{w}_t - \beta_t \mathbf{g}_t\|)}$ for $\beta_t = 1/\sqrt{t}$.

---

However, by doing this we are faced with two new problems: First, computing the optimal $\mathbf{w}$ is a non-convex optimization problem, so it may be difficult to find its global minimum. Second, the value of (2) is only an upper bound on (3), so we need a new analysis to show the optimal $\mathbf{w}$ actually has good prediction accuracy. To solve the latter issue, we prove in Lemma 2.2 that any $\mathbf{w}$ with value $< 0$ for the game (2) achieves prediction error at most $\lambda + O(\epsilon)$ and, since we can take $\lambda = \eta + O(\epsilon)$, we get a proper predictor achieving $\eta + O(\epsilon)$ error, matching the improper learner of [DGT19]:

**Lemma 2.2** (Lemma C.4 in supplement). *Suppose that $(\mathbf{X}, Y) \sim \mathcal{D}$ with $\mathbf{X}$ valued in $\mathbb{R}^d$ and $Y$ valued in $\{\pm 1\}$. Suppose that $\mathbf{w}$ is a vector in the unit ball of $\mathbb{R}^d$ such that $\operatorname{err}_\mathcal{X}(\mathbf{w}) \geq \lambda + 2\epsilon$ for some $\lambda, \epsilon \geq 0$. Then there exists $r \geq 0$ such that $\Pr_\mathcal{D}[\mathbf{X} \in \mathcal{S}(\mathbf{w}, r)] \geq 2\epsilon$ and $L_\lambda^{\mathcal{S}(\mathbf{w}, r)}(\mathbf{w}) \geq 0$.*

Now, knowing that it suffices to find a $\mathbf{w}$ with negative value for (2), we can resolve the issue of optimizing the non-convex objective over $\mathbf{w}$. If gradient descent fails to find a point $\mathbf{w}$ with negative LeakyRelu loss, this means the max player has been very successful in finding convex losses where the current iterate $\mathbf{w}_t$ performs poorly compared to $\mathbf{w}^*$, which achieves negative loss. This cannot go on for too long, because gradient descent is a provably low regret algorithm for online convex optimization [Zin03]. This proves the margin case of Theorem 1.1.

## 3 Learning Misspecified GLMs

We now proceed to the more challenging problem of learning misspecified GLMs. Like [DGT19], our algorithm MISSPECGLM (see Algorithm 9 in the supplement) breaks the domain $\mathcal{X}$ into disjoint regions $\{\mathcal{X}^{(i)}\}$ and assigns a constant label $s^{(i)} \in \{\pm 1\}$ to each. However the key challenge is that GLMs are inherently richer predictive models. They can have regions where they have high and low confidence. What this means for us is that in order to compete with the optimal predictor $\operatorname{sgn}(\mathbf{w}^*, \cdot)$ there is no longer a global upper bound on the error that we can afford in each region. Rather, the target error that we are shooting for can vary wildly (in ways that we cannot directly estimate from the data) across regions.

**Splitting, Merging, and Freezing Regions.** Our algorithm maintains a piecewise-constant predictor over a set of *live* and *frozen* regions; a region is frozen only if we can certify that we achieve nearly optimal prediction error on it. How can we ever certify this? The following tells us that optimizing the convex LeakyRelu loss can indeed prove lower bounds on the optimal zero-one loss:

**Lemma 3.1** (Informal, see Lemma F.4). *Let $\epsilon \geq 0$ be arbitrary, and let $\mathcal{X}'$ be any subset of $\mathcal{X}$. Define $\lambda = \min\{\lambda > 0 : L_\lambda^{\mathcal{X}'}(\mathbf{w}^*) \leq -\frac{2\epsilon}{L} \cdot \mathbb{E}[|\sigma(\langle \mathbf{w}^*, \mathbf{X} \rangle)| \mid \mathbf{X} \in \mathcal{X}']\}$. Then $\operatorname{err}_{\mathcal{X}'}(\mathbf{w}^*) \geq \lambda - \zeta - O(\epsilon)$.*

Based on data we can estimate $\lambda$ as defined in the Lemma. If our current zero-one loss on $\mathcal{X}'$ is at most $\lambda + O(\epsilon)$, we can safely freeze the region $\mathcal{X}'$ because our error rate is close to optimal. If not, we split the region in two (using a similar rounding to [DGT19]) and argue that by doing so we can either (1) improve the accuracy of our predictor (see Lemma F.3), or (2) reduce the amount of *variance in accuracy unexplained by the partition*. Then an "energy decrement" argument guarantees successful termination.

The last complication is that just splitting regions all of the time would lead to exponential runtime and sample complexity. To fix this, we add a crucial *merging* step to keep the number of regions bounded, and show that our progress guarantee holds in spite of this step:

**Lemma 3.2** (Informal, see Lemma F.12). *In the course of running* LEARNMISSPECGLM, *if a region in the current partition $\mathcal{X} = \sqcup \mathcal{X}^{(i)}$ with constant labels $\{s^{(i)}\}$ gets split, but the overall zero-one error of the classifier does not change, then even after some regions of the partition possibly get merged immediately afterwards, the variance $\mathbb{V}_i[\mathrm{err}_{\mathcal{X}^{(i)}}(s^{(i)})]$ increases by* $\mathrm{poly}(\epsilon, 1/L, \zeta)$.

The final predictor is a threshold circuit whose structure records the history of splitting and merging regions over time. The details are rather involved and we defer them to Section F of the supplement.

## 4  Statistical Query Lower Bounds

To prove Theorem 1.3, we establish a surprisingly missed connection between learning under Massart noise and Valiant's notion of evolvability [Val09]. Feldman [Fel08] showed that a concept $f$ is *evolvable* with respect to Boolean loss if and only if it can be efficiently learned by a *correlational SQ* (CSQ) algorithm, i.e. one that only gets access to the data in the following form. Rather than directly getting samples, it is allowed to make noisy queries to statistics of the form $\mathbb{E}_{(\mathbf{X},Y)\sim\mathcal{D}}[Y \cdot G(\mathbf{X})]$ for any $G : \mathcal{X} \to \{\pm 1\}$. See Section E.1 in the supplement for the precise definitions. Note that unlike general SQ algorithms, CSQ algorithms do not get access to statistics like $\mathbb{E}_{(\mathbf{X})\sim\mathcal{D}_{\mathbf{x}}}[G(\mathbf{X})]$, and when $\mathcal{D}_{\mathbf{x}}$ is unknown, this can be a significant disadvantage [Fel11].

At a high level, the connection between learning under Massart noise and learning with CSQs (without label noise) stems from the following simple observation. For any function $G : \mathcal{X} \to \{\pm 1\}$, concept $f$, and distribution $\mathcal{D}$ arising from $f$ with $\eta$-Massart noise

$$\mathbb{E}_{(\mathbf{X},Y)\sim\mathcal{D}}[Y \cdot G(\mathbf{X})] = \mathbb{E}_{(\mathbf{X},Y)\sim\mathcal{D}}[f(\mathbf{X})G(\mathbf{X})(1 - 2\eta(\mathbf{X}))].$$

One can think of the factor $1 - 2\eta(\mathbf{X})$ as, up to a normalization factor $Z$, *tilting* the original distribution $\mathcal{D}_{\mathbf{x}}$ to some other distribution $\mathcal{D}'_{\mathbf{x}}$. If we consider the noise-free distribution $\mathcal{D}'$ over $(\mathbf{X}, Y)$ where $\mathbf{X} \sim \mathcal{D}'_{\mathbf{x}}$ and $Y = f(\mathbf{X})$, then the statistic $\mathbb{E}_{(\mathbf{X},Y)\sim\mathcal{D}}[Y \cdot G(\mathbf{X})]$ is equal, up to a factor of $Z$, to the statistic $\mathbb{E}_{(\mathbf{X},Y)\sim\mathcal{D}'}[Y \cdot G(\mathbf{X})]$ (Fact E.5 in the supplement).

This key fact can be used to show that distribution-independent CSQ algorithms that learn without label noise yield distribution-independent algorithms that learn under Massart noise (Theorem E.4 in the supplement). It turns out a partial converse holds, and we use this in conjunction with known CSQ lower bounds for learning halfspaces [Fel11] to establish Theorem 1.3.

## 5  Numerical Experiments

We evaluated FILTERTRON, gradient descent on the LeakyRelu loss, logistic regression, and random forest (to compare with a less interpretable, non-halfspace classifier) on the UCI Adult dataset, obtained from the UCI Machine Learning Repository [DG17] and originally curated by [Koh96].[3] It consists of demographic information for $N = 48842$ individuals, with a total of 14 attributes including age, gender, education, and race, and the prediction task is to determine whether a given individual has annual income exceeding $\$50K$. Henceforth we will refer to individuals with annual income exceeding (resp. at most) $\$50K$ as *high (resp. low) income*. Some relevant statistics on individuals in the dataset: 23.9% are high-income, 9.6% are African-American, 1.2% are high-income and African-American, 33.2% are female, 3.6% are high-income and female, 10.3% are immigrants, and 2.0% are high-income immigrants. Because our theoretical guarantees are in terms

of zero-one error on the distribution *with* Massart corruptions, we measured the performance using this metric, i.e. both the training and test set labels are corrupted by the Massart adversary.

For various $\eta$ and various predicates $p$ on demographics, we considered the following $\eta$-Massart adversary: for individuals who satisfy the $p$ (the *target* group), do not flip the response, and for all other individuals, flip with probability $\eta$. The intuition is that because most individuals in the dataset are low-income, the corruptions will make those not satisfying $p$ appear to be higher-income on average, which may bias the learner against classifying individuals satisfying $p$ as high-income.

We measured the performance of a classifier under this attack along two axes: A) accuracy over the entire test set, and B) accuracy over the high-income members of the target group in the test set.

For every $p$, we took a five-fold cross validation of the dataset, and for every $\eta \in [0, 0.1, 0.2, 0.3, 0.4]$ we repeated the following five times and took the mean: $(1)$ randomly flip the labels for the training and test set according to the Massart adversary, $(2)$ train on the noisy training set, and $(3)$ evaluate according to $(A)$ and $(B)$. For FILTERTRON and gradient descent on the LeakyRelu loss, we ran for 2000 iterations with step size 0.05 and $\epsilon$ chosen by a naive grid search over $[0.05, 0.1, 0.15, 0.2]$.

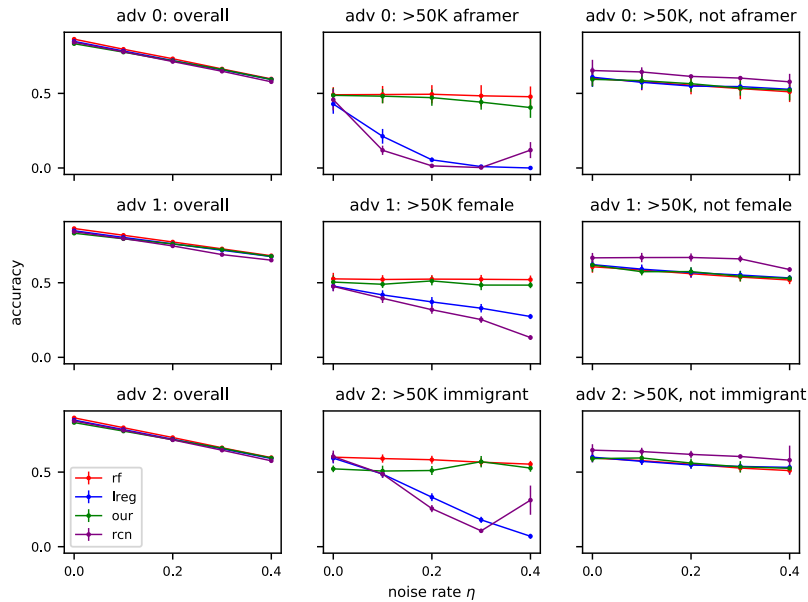

Figure 1: **UCI Adult**: Effect of Massart adversaries, targeting African-Americans, females, immigrants respectively, on accuracy of FILTERTRON and baselines. Left/center/right depict accuracy over entire test set/high-income subset of target group/high-income subset of complement of target group.

The predicates $p$ that we considered were (1) African-American, (2) females, and (3) individuals whose native country is not the United States; Figure 1 plots the medians across each five-fold cross-validations, with error bars given by a single standard deviation. The experiments on the Adult dataset were conducted in a Kaggle kernel with a Tesla P100 GPU, and each predicate took roughly 40 minutes to run. In all cases, while the algorithms evaluated achieve very similar test accuracy, FILTERTRON correctly classifies a noticeably larger fraction of high-income members of the target group than logistic regression or gradient descent on LeakyReLU, and is comparable to random forest. We defer additional implementation details and experimental results to the supplement. All code for the experiments can be found at `https://github.com/secanth/massart`.

**Fairness**   There is by now a mature literature on algorithmic fairness [DHP$^+$12, HPS16, KMR17], with many well-defined notions of what it means to be fair coming from different normative considerations. There is no one notion that clearly dominates; rather it depends on the circumstances and sometimes they are even at odds with one other [Cho17, KMR17, MP19]. Our results are perhaps most closely related to the notion of *equality of opportunity* [HPS16], as our experiments show that some off-the-shelf algorithms can suffer from high false negative rates on various demographic groups when noise is added to the rest of the data. In contrast, we show that provably robust algorithms can

be a useful ingredient in both anticipating and mitigating certain patterns of unfairness that can arise from using off-the-shelf learning algorithms. Our specific techniques are built on top of new efficient algorithms to search for portions of the distribution where a classifier is performing poorly and can be improved.

While we stress that the appealing properties that these experiments suggest that our techniques possess are purely empirical observations, given that Massart noise is a model of varying noise across populations, it is plausible that algorithms designed to tolerate Massart noise will generally work better in situations when noise varies across target groups. We believe that these tools may find other compelling applications.

## Broader Impacts

In this work we design algorithms with provable robustness guarantees in the challenging setting where the level of noise is allowed to vary across the domain. This models several scenarios of interest, most notably situations where data provided by certain demographic groups is subject to more noise than others. In a natural experiment on the UCI Adult dataset, we show that coping with this type of noise can help mitigate some natural types of unfairness that arise with off-the-shelf algorithms. Moreover our algorithms have the additional benefit that they lead to more readily interpretable hypotheses. In many settings of interest, we are able to give proper learning algorithms (where previously only improper learning algorithms were known). This could potentially help practitioners better understand and diagnose complex machine learning systems they are designing, and troubleshoot ways that the algorithm might be amplifying biases in the data.

## Acknowledgments

S.C. was supported in part by a Paul and Daisy Soros Fellowship, NSF CAREER Award CCF-1453261, and NSF Large CCF-1565235. F.K. was supported in part by NSF CCF-1453261. A.M. was supported in part by a Microsoft Trustworthy AI Grant, NSF CAREER Award CCF-1453261, NSF Large CCF-1565235, a David and Lucile Packard Fellowship, an Alfred P. Sloan Fellowship and an ONR Young Investigator Award.

## Footnotes

[1]This equivalence is literally true only for an oblivious adversary, but in Appendix I we explain that all the algorithms in this paper succeed in the adaptive case as well.

[2]It is standard in such settings to assume $\mathcal{D}_{\mathbf{x}}$ has bounded support. We can reduce from this to the unit ball case by normalizing points in the support and scaling $L$ appropriately.

[3]We also conducted synthetic experiments where FILTERTRON outperformed some baselines, see Section G.1 of the supplement.

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
