[Supplementary Material 1]

## Supplementary Materials Roadmap

In Section A, we outline the architecture of our proofs in greater detail. In Section B, we precisely define the models we work with and include other preliminaries like notation and useful technical facts. In Section C, we prove our results on proper learning halfspaces in the presence of Massart noise, in particular Theorem 1.1, restated here for convenience:

**Theorem 1.1** (Informal, see Theorem C.18). *For any $0 \leq \eta < 1/2$, let $\mathcal{D}$ be a distribution over $(\mathbf{X}, Y)$ given by an $\eta$-Massart halfspace, and suppose $\mathbf{X}$ is supported on vectors of bit-complexity at most b. There is an algorithm which runs in time $\mathrm{poly}(d, b, 1/\epsilon)$ and sample complexity $\widetilde{O}(\mathrm{poly}(d, b)/\epsilon^3)$ and outputs a classifier $\mathbf{w}$ whose 0-1 error over $\mathcal{D}$ is at most $\eta + \epsilon$.*

In Section D, we prove our results on knowledge distillation, in particular Theorem 1.2, restated here for convenience:

**Theorem 1.2** (Informal, see Theorem D.3). *Let $\mathcal{D}, b$ be as in Theorem 1.1. There is an algorithm which, given query access to a possibly improper hypothesis $\mathbf{h}$ and $\widetilde{O}(\mathrm{poly}(d, b)/\epsilon^4)$ samples, runs in time $\mathrm{poly}(d, b, 1/\epsilon)$ and outputs a proper classifier $\mathbf{w}$ whose 0-1 error over $\mathcal{D}$ exceeds that of $\mathbf{h}$ by at most $\epsilon$. If the underlying halfspace has a margin $\gamma$, there is an algorithm that achieves this but only requires $\widetilde{O}(d/\gamma^2\epsilon^4)$ samples and runs in near-linear time.*

In Section E, we prove our statistical query lower bound for learning Massart halfsapces to error $\mathsf{OPT} + \epsilon$, restated here for convenience.

**Theorem 1.3** (Informal, see Theorem E.1). *Any SQ algorithm for distribution-independently learning halfspaces to error $\mathsf{OPT} + o(1)$ under Massart noise, requires a super-polynomial number of queries.*

In Section F, we prove our results on learning misspecified generalized linear models, in particular Theorem 1.4, restated here for convenience:

**Theorem 1.4** (Informal, see Theorem F.1). *Let $\sigma : \mathbb{R} \to [-1, 1]$ be any odd, monotone, L-Lipschitz function. For any $\epsilon > 0$ and $0 \leq \zeta < 1/2$, there is a polynomial-time algorithm which, given $\mathrm{poly}(L, \epsilon^{-1}, (\zeta \vee \epsilon)^{-1})$ samples from a $\zeta$-misspecified GLM with link function $\sigma$ and true direction $\mathbf{w}^*$, outputs an improper classifier $\mathbf{h}$ whose 0-1 error over $\mathcal{D}$ satisfies*

$$\underset{\mathcal{X}}{\mathrm{err}}(\mathbf{h}) \leq \frac{1 - \mathbb{E}_{\mathcal{D}}[\sigma(|\langle \mathbf{w}^*, \mathbf{x} \rangle|)]}{2} + \zeta + \epsilon.$$

Lastly, in Section G, we describe the experiments we conducted on synthetic data and the UCI Adult dataset. Finally, in Appendix H, we record miscellaneous deferred proofs, and in Appendix I we show that FILTERTRON, like [DGT19], works in slightly greater generality than the Massart noise model.

## A  Overview of Techniques

**The LeakyRelu loss.**   All of our algorithms use the LeakyRelu loss $\ell_\lambda$ in a central way, similarly to [DGT19]. See the definition in Section B.2. We briefly explain why this is natural, and in the process explain some important connections to the literature on learning generalized linear models [AHW96, KW98, KS09, KKSK11]. First, it is easy to see that the problem of learning a halfspace where the labels are flipped with probability $\eta$ (i.e. under RCN) is equivalent to learning a generalized linear model $\mathbb{E}[Y \mid \mathbf{X}] = (1 - 2\eta)\mathrm{sgn}(\langle \mathbf{w}^*, \mathbf{X} \rangle)$. Auer et al. [AHW96] defined the notion of a *matching loss* which constructs a loss function, assuming the link function is monotonically increasing, that is convex and so has no bad local minima; if the link function is also Lipschitz then minimizing this loss provably recovers the GLM. In particular, this works to learn halfspaces with margin under RCN [AHW96, KS09, KKSK11, Kan18]. We see that LeakyRelu$_\eta$ is a matching loss from the integral representation

$$\ell_\eta(\mathbf{w}, \mathbf{X}) = \frac{1}{2} \int_0^{\langle \mathbf{w}, \mathbf{X} \rangle} ((1 - 2\eta)\mathrm{sgn}(r) - Y)dr.$$

In fact, this integral representation is used implicitly in [DGT19] (i.e. Lemma F.5 below) and also in our proof of Lemma C.4 below.

## A.1  Separation Oracles and Proper-to-Improper Reduction

**An idealized zero-sum game.**  Our proper learning algorithms are all based on the framework of finding approximately optimal strategies in the following zero-sum game:

$$\min_{\|\mathbf{w}\| \leq 1} \max_c \mathbb{E}[c(\mathbf{X})\ell_\lambda(\mathbf{w}, \mathbf{X})] \tag{3}$$

where $\lambda$ is a fixed parameter chosen slightly larger than $\eta$, and $c(\mathbf{X})$ is any measurable function such that $c(\mathbf{X}) \geq 0$ and $\mathbb{E}[c(\mathbf{X})] = 1$. It is helpful to think of $c(\mathbf{X})$ as a *generalized filter*, and we will often consider functions of the form $c_S(\mathbf{X}) = \frac{\mathbb{1}[\mathbf{X} \in S]}{\Pr[\mathbf{X} \in S]}$, which implement conditioning on $\mathbf{X} \in S$. Because the LeakyRelu is homogenous ($\ell_\lambda(\mathbf{w}, c\mathbf{X}) = c \cdot \ell_\lambda(\mathbf{w}, \mathbf{X})$ for $c \geq 0$), we sometimes reinterpret $c(\mathbf{X})$ as a rescaling of $\mathbf{X}$.

In this game, we can think of the $\min$ player as the classifier, whose goal is to output a hyperplane $\mathbf{w}$ with loss almost as small as the ground truth hyperplane $\mathbf{w}^*$. On the other hand, the $\max$ player is a special kind of *discriminator* whose goal is to prove that $\mathbf{w}$ has inferior predictive power compared to $\mathbf{w}^*$, by finding a reweighting of the data such that $\mathbf{w}$ performs very poorly in the LeakyRelu loss. This is based on the fact that in the Massart model, for *any* reweighting $c(\mathbf{X})$ of the data, $\mathbf{w}^*$ performs well in the sense that $\mathbb{E}[c(\mathbf{X})\ell_\lambda(\mathbf{w}^*, \mathbf{X})] < 0$. This follows from Lemma C.2.

In fact, it turns out that this class of discriminators is so powerful that $\mathbf{w}$ having optimal zero-one loss is *equivalent* to having value less than zero in the minimax game. We will show this in Section D. Furthermore, the outer optimization problem is convex, so if we could find a good strategy for the $\max$ player, we could then optimize $\mathbf{w}$. Unfortunately, this is impossible because the $\max$ player's strategies in this game range over *all possible reweightings* of the distribution over $\mathbf{X}$, so it is both statistically and computationally intractable to compute the best response $c$ given $\mathbf{w}$. For example, if $c$ is supported on a set $S$ with extremely low probability, we may never even see a datapoint with $c(\mathbf{X}) > 0$, so it will be impossible to estimate the value of the expectation.

The key to our approach is to fix alternative strategies for the $\max$ player which are computationally and statistically efficient. Then we will analyze the resulting dynamics when the $\mathbf{w}$ player plays against this adversary and updates their strategy in a natural way (e.g. gradient descent). Thus our framework naturally yields simple and practical learning algorithms.

**Comparison to previous approach and no-go results.**  We briefly explain the approach of [DGT19] in the context of (3) and why their approach only yields an improper learner. In the first step of the algorithm, they minimize the LeakyRelu loss over the entire space (i.e. taking $c(\mathbf{X}) = 1$). They show this generates a $\mathbf{w}$ with good zero-one loss on a subset of space $S$. They fix this hypothesis on $S$, and then restrict to $\mathbb{R}^d \setminus S$ (i.e., take $c(\mathbf{X}) = \frac{\mathbb{1}[\mathbf{X} \notin S]}{\Pr[\mathbf{X} \notin S]}$) and restart their algorithm. Because they fix $c$ before minimizing over $\mathbf{w}$, their first step is minimizing a fixed convex surrogate loss. However, Theorem 3.1 of [DGT19] establishes that no proper learner based on minimizing a fixed surrogate loss will succeed in the Massart setting. In contrast, our algorithms choose $c$ adversarially based on $\mathbf{w}$. For this reason, we evade the lower bound of [DGT19] and successfully solve the proper learning problem.

**Proper learner for halfspaces with margin.**  Our proper learner for learning halfspaces with margin is based upon the following upper bound on (3):

$$\min_{\|\mathbf{w}\| \leq 1} \max_{r > 0} \mathbb{E}[\ell_\lambda(\mathbf{w}, \mathbf{X}) \mid |\langle \mathbf{w}, \mathbf{X} \rangle| \leq r], \tag{4}$$

where $r$ will be restricted so that $\Pr[|\langle \mathbf{w}, \mathbf{X} \rangle| \leq r] \geq \epsilon$ for some small $\epsilon > 0$. By (greatly) restricting the possible strategies for the discriminator to "slabs" along the direction of $\mathbf{w}$, we completely fix the problem of computational and statistical intractability for the $\max$-player. In particular, the optimization problem over $r > 0$ is one-dimensional, and the expectation can be accurately estimated from samples using rejection sampling.

However, by doing this we are faced with two new problems: First, computing the optimal $\mathbf{w}$ is a non-convex optimization problem, so it may be difficult to find its global minimum. Second, the value of (4) is only an upper bound on (3), so we need a new analysis to show the optimal $\mathbf{w}$ actually has good prediction accuracy. To solve the latter issue, we prove in Lemma C.4 that any $\mathbf{w}$ with value $< 0$ for the game (4) achieves prediction error at most $\lambda + O(\epsilon)$ and, since we can take $\lambda = \eta + O(\epsilon)$, we

get a proper predictor matching the guarantees for the improper learner of [DGT19]. Also knowing that it suffices to find a $\mathbf{w}$ with negative value for (4), we can resolve the issue of optimizing the non-convex objective over $\mathbf{w}$. If gradient descent fails to find a point $\mathbf{w}$ with negative LeakyRelu loss, this means the $\max$ player has been very successful in finding convex losses where the current iterate $\mathbf{w}_t$ performs poorly compared to $\mathbf{w}^*$, which achieves negative loss. This cannot go on for too long, because gradient descent is a provably low regret algorithm for online convex optimization [Zin03].

**Proper learner for general halfspaces.** The above argument based on low regret fails when the margin is allowed to be exponentially small in the dimension. The reason is that the optimal value of (4) can be too close to zero. In order to deal with this issue, we need an algorithm which can "zoom in" on points very close to the halfspace, as in earlier algorithms [BFKV98, Coh97, DV08] for learning halfspaces under RCN. This becomes somewhat technically involved, as concentration of measure can fail when random variables of different size are summed. Ultimately, we show how to build upon some of the techniques in [Coh97] to construct the needed rescaling $c$, giving us a separation oracle. By doing this, we give a variant of the algorithm of [Coh97] that is robust to Massart noise.

**Proper to improper reduction.** Suppose that we are given black-box access to a hypothesis $h$ (possibly improper) achieving good prediction error in the Massart halfspace setting. In the context of (3), this serves as valuable advice for the $\min$ player, because it enables efficient sampling from the *disagreement set* $\{\mathbf{x} : h(\mathbf{x}) \neq \operatorname{sgn}(\langle \mathbf{w}, \mathbf{X} \rangle)\}$. If $h$ has significantly better performance than $\mathbf{w}$ overall, this means that $\mathbf{w}$ has very poor performance on the disagreement set – otherwise, the zero-one loss of $h(\mathbf{X})$ and $\operatorname{sgn}(\langle \mathbf{w}, \mathbf{X} \rangle)$ would be close.

Using this idea, we can "boost" our proper learners to output a $\mathbf{w}$ whose performance almost matches $h$, by running the same algorithms as before but having the $\max$ player always restrict to the disagreement set before generating its generalized filter $c$. In light of the SQ lower bound we observe for achieving optimal error (discussed later in the overview), this has an appealing consequence – it means that in the Massart setting, the power of SQ algorithms with blackbox access to a good teacher can be much more powerful than ordinary SQ algorithms. The fact that blackbox access suffices is also surprising, since most popular approaches to knowledge distillation are not blackbox (see e.g. [HVD14]).

## A.2 Learning Misspecified GLMs

We now proceed to the more challenging problem of learning misspecified GLMs. In this case, when $\zeta > 0$, it could be that achieving negative value in (3) is actually impossible; to deal with this, we do not attempt to learn a proper classifier in the general misspecified setting except when $\zeta = 0$. Similar to [DGT19], our algorithm breaks the domain $\mathcal{X}$ into disjoint regions $\{\mathcal{X}^{(i)}\}$ and assigns a constant label $s^{(i)} \in \{\pm 1\}$ to each. This setting poses a host of new challenges, and the partitions induced by our algorithm will need to be far richer in structure than those of [DGT19].

**Key issue: target error varies across regions** As a thought experiment, consider what it takes for an improper hypothesis $\mathbf{h}$ to compete with $\operatorname{sgn}(\langle \mathbf{w}^*, \cdot \rangle)$. Let $\mathcal{D}$ be the distribution over $(\mathbf{X}, Y)$ arising from a misspecified GLM (see Definition B.2). Now take an improper classifier $\mathbf{h}$ which breaks $\mathcal{X}$ into regions $\{\mathcal{X}^{(i)}\}$. A natural way to ensure $\mathbf{h}$ competes with $\mathbf{w}^*$ over all of $\mathcal{X}$ is to ensure that it competes with it over every region $\mathcal{X}^{(i)}$. For any $i$, note that the zero-one error of $\operatorname{sgn}(\langle \mathbf{w}^*, \cdot \rangle)$ with respect to $\mathcal{D}$ restricted to $\mathcal{X}^{(i)}$ satisfies

$$\operatorname*{err}_{\mathcal{X}^{(i)}}(\mathbf{w}^*) \leq \frac{1 - \mathbb{E}_{\mathcal{D}}[\sigma(|\langle \mathbf{w}^*, \mathbf{X} \rangle|) \mid \mathbf{X} \in \mathcal{X}^{(i)}]}{2} \triangleq \lambda(\mathcal{X}^{(i)}).$$

Now we can see the issue that for learning Massart halfspaces, $\lambda(\mathcal{X}^{(i)})$ is always $\eta$. But for general misspecified GLMs, $\lambda(\mathcal{X}^{(i)})$ can vary wildly with $i$. For this reason, iteratively breaking off regions like in [DGT19] could be catastrophic if, for instance, one of these regions $\mathcal{R}$ has $\lambda(\mathcal{R})$ significantly less than $\lambda(\mathcal{X})$ and yet our error guarantee on $\mathcal{R}$ is only in terms of some global quantity like in [DGT19]. Without the ability to go back and revise our predictions on $\mathcal{R}$, we cannot hope to achieve the target error claimed in Theorem 1.4.

**Removing regions conservatively** In light of this, we should only ever try to recurse on the complement of a region $\mathcal{R}$ when we have certified that our classifier achieves zero-one error $\lambda(\mathcal{R}) + O(\epsilon)$

over that region. Here is one extreme way to achieve this. Consider running SGD on the LeakyRelu loss and finding some direction $\mathbf{w}$ and annulus $\mathcal{R}$ on which the zero-one error of $\text{sgn}(\langle\mathbf{w},\cdot\rangle)$ is $O(\epsilon)$-close to $\lambda(\mathcal{X})$ (henceforth we will refer to this as "running SGD plus filtering"). Firstly, it is possible to show that $\text{err}_{\mathcal{R}}(\mathbf{w}) \leq \lambda(\mathcal{X}) + O(\epsilon)$ (see Lemma F.4 and Lemma F.5). There are two possibilities. First, it could be that $\lambda(\mathcal{R}) \geq \lambda(\mathcal{X}) - \epsilon$, in which case we can safely recurse on the complement of $\mathcal{R}$. Alternatively we could have $\lambda(\mathcal{R}) < \lambda(\mathcal{X}) - \epsilon$.[4] But if this happens, we can recurse by running SGD plus filtering on $\mathcal{R}$. Moreover this recursion must terminate at depth $O(1/\epsilon)$ because $\lambda(\mathcal{R}') \geq 0$ for any region $\mathcal{R}'$, so we can only reach the second case $O(1/\epsilon)$ times in succession, at which point we have found a hypothesis and region on which the hypothesis is certified to be competitive with $\mathbf{w}^*$. The only issue is that after $O(1/\epsilon)$ levels of recursion, this region might only have mass $\exp(-\Omega(1/\epsilon))$, and we would need to take too many samples to get any samples from this region.

**Our algorithm** The workaround is to only ever run SGD plus filtering on regions with non-negligible mass. At a high level, our algorithm maintains a constantly updating partition $\{\mathcal{X}^{(i)}\}$ of the space into a *bounded* number of regions, each equipped with a $\pm 1$ label, and only ever runs SGD plus filtering on the largest region at the time. Every time SGD plus filtering is run, some region of the partition $\mathcal{X}'$ might get refined into two pieces $\tilde{\mathcal{X}}$ and $\mathcal{X}'\backslash\tilde{\mathcal{X}}$, and their new labels might be updated to differ. To ensure the number of regions in the partition remains bounded, the algorithm will occasionally merge regions of the partition on which their respective labels have zero-one error differing by at most some $\delta$. And if running SGD plus filtering ever puts us into the first case above for some region $\mathcal{R}$, we can safely remove $\mathcal{R}$ from future consideration.

The key difficulty is to prove that the above procedure does not go on forever, which we do via a careful potential argument. We remark that the partitions this algorithm produces are inherently richer in structure than those output by that of [DGT19]. Whereas their overall classifier can be thought of as a decision tree, the one we output is a general *threshold circuit* (see Remark F.2), whose structure records the history of splitting and merging regions over time.

### A.3 Statistical Query Lower Bounds

To prove Theorem 1.3, we establish a surprisingly missed connection between learning under Massart noise and Valiant's notion of evolvability [Val09]. Feldman [Fel08] showed that a concept $f$ is *evolvable* with respect to Boolean loss if and only if it can be efficiently learned by a *correlational SQ* (CSQ) algorithm, i.e. one that only gets access to the data in the following form. Rather than directly getting samples, it is allowed to make noisy queries to statistics of the form $\mathbb{E}_{(\mathbf{X},Y)\sim\mathcal{D}}[Y \cdot G(\mathbf{X})]$ for any $G : \mathcal{X} \rightarrow \{\pm 1\}$. See Section E.1 for the precise definitions. Note that unlike SQ algorithms, CSQ algorithms do not get access to statistics like $\mathbb{E}_{(\mathbf{X})\sim\mathcal{D}_{\mathbf{x}}}[G(\mathbf{X})]$, and when $\mathcal{D}_{\mathbf{x}}$ is unknown, this can be a significant disadvantage [Fel11].

At a high level, the connection between learning under Massart noise and learning with CSQs (without label noise) stems from the following simple observation. For any function $G : \mathcal{X} \rightarrow \{\pm 1\}$, concept $f$, and distribution $\mathcal{D}$ arising from $f$ with $\eta$-Massart noise

$$\mathbb{E}_{(\mathbf{X},Y)\sim\mathcal{D}}[Y \cdot G(\mathbf{X})] = \mathbb{E}_{(\mathbf{X},Y)\sim\mathcal{D}}[f(\mathbf{X})G(\mathbf{X})(1 - 2\eta(\mathbf{X}))].$$

One can think of the factor $1-2\eta(\mathbf{X})$ as, up to a normalization factor $Z$, *tilting* the original distribution $\mathcal{D}_{\mathbf{x}}$ to some other distribution $\mathcal{D}'_{\mathbf{x}}$. If we consider the noise-free distribution $\mathcal{D}'$ over $(\mathbf{X}, Y)$ where $\mathbf{X} \sim \mathcal{D}'_{\mathbf{x}}$ and $Y = f(\mathbf{X})$, then the statistic $\mathbb{E}_{(\mathbf{X},Y)\sim\mathcal{D}}[Y \cdot G(\mathbf{X})]$ is equal, up to a factor of $Z$, to the statistic $\mathbb{E}_{(\mathbf{X},Y)\sim\mathcal{D}'}[Y \cdot G(\mathbf{X})]$. See Fact E.5.

This key fact can be used to show that distribution-independent CSQ algorithms that learn without label noise yield distribution-independent algorithms that learn under Massart noise (see Theorem E.4). It turns out a partial converse holds, and we use this in conjunction with known CSQ lower bounds for learning halfspaces [Fel11] to establish Theorem 1.3.

# B Technical Preliminaries

## B.1 Generative Model

In this section we formally define the models we will work with. First recall the usual setting of classification under Massart noise.

**Definition B.1** (Classification Under Massart Noise). *Fix noise rate $0 \leq \eta < 1/2$ and domain $\mathcal{X}$. Let $\mathcal{D}_{\mathbf{x}}$ be an arbitrary distribution over $\mathcal{X}$. Let $\mathcal{D}$ be a distribution over pairs $(\mathbf{X}, Y) \in \mathcal{X} \times \{\pm 1\}$ given by the following generative model. Fix an unknown function $f : \mathcal{X} \to \{\pm 1\}$. Ahead of time, an adversary chooses a quantity $0 \leq \eta(\mathbf{x}) \leq \eta$ for every $\mathbf{x}$. Then to sample $(\mathbf{X}, Y)$ from $\mathcal{D}$, 1) $\mathbf{X}$ is drawn from $\mathcal{D}_{\mathbf{x}}$, 2) $Y = f(\mathbf{X})$ with probability $1 - \eta(\mathbf{X})$, and otherwise $Y = -f(\mathbf{X})$. We will refer to the distribution $\mathcal{D}$ as arising from concept $f$ with $\eta$-Massart noise.*

*In the special case where $\mathcal{X}$ is a Hilbert space and $f(\mathbf{x}) \triangleq \operatorname{sgn}(\langle \mathbf{w}^*, \mathbf{x} \rangle)$ for some unknown $\mathbf{w}^* \in \mathcal{X}$, we will refer to the distribution $\mathcal{D}$ as* arising from an $\eta$-Massart halfspace.

We will consider the following extension of Massart halfspaces.

**Definition B.2** (Misspecified Generalized Linear Models). *Fix misspecification parameter $0 \leq \zeta < 1/2$ and Hilbert space $\mathcal{X}$. Let $\sigma : \mathbb{R} \to [-1, 1]$ be any odd, monotone, $L$-Lipschitz function, not necessarily known to the learner. Let $\mathcal{D}_{\mathbf{x}}$ be any distribution over $\mathcal{X}$ supported on the unit ball.[5]*

*Let $\mathcal{D}$ be a distribution over pairs $(\mathbf{X}, Y) \in \mathcal{X} \times \{\pm 1\}$ given by the following generative model. Fix an unknown $\mathbf{w}^* \in \mathcal{X}$. Ahead of time, a $\zeta$-misspecification adversary chooses a function $\delta : \mathcal{X} \to \mathbb{R}$ for which*

$$-2\zeta \leq \delta(\mathbf{x})\operatorname{sgn}(\langle \mathbf{w}^*, \mathbf{x} \rangle) \leq 1 - |\sigma(\langle \mathbf{w}^*, \mathbf{x} \rangle)|$$

*for all $\mathbf{x} \in \mathcal{X}$. Then to sample $(\mathbf{X}, Y)$ from $\mathcal{D}$, 1) $\mathbf{X}$ is drawn from $\mathcal{D}_{\mathbf{x}}$, 2) $Y$ is sampled from $\{\pm 1\}$ so that $\mathbb{E}[Y \mid \mathbf{X}] = \sigma(\langle \mathbf{w}^*, \mathbf{X} \rangle) + \delta(\mathbf{X})$. We will refer to such a distribution $\mathcal{D}$ as* arising from an $\zeta$-misspecified GLM with link function $\sigma$.

**Remark B.3.** *We emphasize that in the setting of $\zeta$-misspecified GLMs, the case of $\zeta = 0$ is already nontrivial as the adversary can decrease the noise level arbitrarily at any point; in particular, the $\eta$-Massart adversary in the halfspace model can be equivalently viewed as a $0$-misspecification adversary for the link function $\sigma(z) = (1 - 2\eta)\operatorname{sgn}(z)$. While this is not Lipschitz, in the case that the halfspace has a $\gamma$ margin we can make it $O(1/\gamma)$-Lipschitz by making the function linear on $[-\gamma, \gamma]$, turning it into a "ramp" activation. This shows that (Massart) halfspaces with margin are a special case of (misspecified) generalized linear models [KS09].*

**Remark B.4.** *Misspecified GLMs can also capture a noisy version of agnostic learning. When $\sigma = 0$, we have that $\mathbb{E}[Y \mid \mathbf{X}] = \delta(\mathbf{X})$, so in particular the label for any point $x$ under a misspecified GLM is independent of the ground truth vector $w^*$. For an arbitrary distribution $\mathcal{D}$ over $\mathcal{X} \times \{\pm 1\}$, consider a noisy version $\mathcal{D}'$ given by sampling $(\mathbf{X}, Y)$ from $\mathcal{D}$ and outputting $(\mathbf{X}, Y)$ with probability $1/2 + \zeta$ and $(\mathbf{X}, -Y)$ otherwise. Then $-2\zeta \leq \mathbb{E}[Y \mid \mathbf{X}] \leq 2\zeta$, and $\mathcal{D}'$ therefore arises from a $\zeta$-misspecified GLM with link function equal to the zero function.*

A learning algorithm $\mathcal{A}$ is given i.i.d. samples from $\mathcal{D}$, and its goal is to output a hypothesis $\mathbf{h} : \mathcal{X} \to \{\pm 1\}$ for which $\Pr_{\mathcal{D}}[\mathbf{h}(\mathbf{X}) \neq Y]$ is as small as possible, with high probability. We say that $\mathcal{A}$ is *proper* if $h$ is given by $h(\mathbf{x}) \triangleq \langle \hat{w}, \mathbf{x} \rangle$. If $\mathcal{A}$ runs in polynomial time, we say that $\mathcal{A}$ *PAC learns* in the presence $\eta$-Massart noise.

**Margin.** In some cases, we give stronger guarantees under the additional assumption of a *margin*. This is a standard notion in classification which is used, for example, in the analysis of the perceptron algorithm [MP17]; one common motivation for considering the margin assumption is to consider the case of infinite-dimensional halfspaces (which otherwise would be information-theoretically impossible). Suppose that $\mathbf{X}$ is normalized so that $\|\mathbf{X}\| \leq 1$ almost surely (under the distribution $\mathcal{D}$) and $\|\mathbf{w}^*\| = 1$. We say that the halfspace $\mathbf{w}^*$ has margin $\gamma$ if $|\langle \mathbf{w}^*, \mathbf{X} \rangle| \geq \gamma$ almost surely.

## B.2 Notation

**LeakyRelu Loss** Given *leakage parameter* $\lambda \geq 0$, define the function

$$\underset{\lambda}{\text{LeakyRelu}}(z) = \frac{1}{2}z + \left(\frac{1}{2} - \lambda\right)|z| = \begin{cases} (1-\lambda)z & \text{if } z \geq 0 \\ \lambda z & \text{if } z < 0. \end{cases}$$

Given $\mathbf{w}, \mathbf{x}$, let $\ell_\lambda(\mathbf{w}, \mathbf{x}) \triangleq \mathbb{E}_\mathcal{D}[\text{LeakyRelu}_\lambda(-Y\langle \mathbf{w}, \mathbf{X}\rangle) \mid \mathbf{X} = \mathbf{x}]$ denote the LeakyRelu loss incurred at point $\mathbf{x}$ by hypothesis $\mathbf{w}$. Observe that the $\text{LeakyRelu}_\lambda$ function is convex for all $\lambda \leq 1/2$. Similar to [DGT19], we will work with the convex proxy for 0-1 error given by $L_\lambda(\mathbf{w}) \triangleq \mathbb{E}_{\mathcal{D}_\mathbf{x}}[\ell_\lambda(\mathbf{w}, \mathbf{X})]$.

We will frequently condition on the event $\mathbf{X} \in \mathcal{X}'$ for some subsets $\mathcal{X}' \subseteq \mathcal{X}$. Let $\ell_\lambda^{\mathcal{X}'}$ and $L_\lambda^{\mathcal{X}'}$ denote the corresponding losses under this conditioning.

**Zero-one error** Given $\mathbf{w}$ and $\mathbf{x} \in \mathcal{X}$, let $\text{err}_\mathbf{x}(\mathbf{w}) = \eta(\mathbf{x})$ denote the probability, over the Massart-corrupted response $Y$, that $\text{sgn}(\langle \mathbf{w}, \mathbf{x}\rangle) \neq Y$. Given $\mathcal{X}' \subseteq \mathcal{X}$, let $\text{err}_{\mathcal{X}'}(\mathbf{w}) \triangleq \text{Pr}_\mathcal{D}[\text{sgn}(\langle \mathbf{w}, \mathbf{X}\rangle) \neq Y \mid \mathbf{X} \in \mathcal{X}']$. For any $\mathbf{h}(\cdot) : \mathcal{X}' \rightarrow \{\pm 1\}$, we will also overload notation by defining $\text{err}_{\mathcal{X}'}(\mathbf{h}) \triangleq \text{Pr}_\mathcal{D}[\mathbf{h}(\mathbf{X}) \neq Y \mid \mathbf{X} \in \mathcal{X}']$. Furthermore, if $\mathbf{h}(\cdot)$ is a *constant* classifier which assigns the same label $s \in \{\pm 1\}$ to every element of $\mathcal{X}'$, then we refer to the misclassification error of $\mathbf{h}(\cdot)$ by $\text{err}_{\mathcal{X}'}(s) \triangleq \text{Pr}_\mathcal{D}[Y \neq s \mid \mathbf{X} \in \mathcal{X}']$. When working with a set of samples from $\mathcal{D}$, we will use $\widehat{\text{err}}$ to denote the empirical version of err.

**Probability Mass of Sub-Regions** Given regions $\mathcal{X}'' \subseteq \mathcal{X}' \subseteq \mathcal{X}$, it will also be convenient to let $\mu(\mathcal{X}'' \mid \mathcal{X}')$ denote $\text{Pr}_{\mathcal{D}_\mathbf{x}}[\mathbf{X} \in \mathcal{X}'' \mid \mathbf{X} \in \mathcal{X}']$; when $\mathcal{X}' = \mathcal{X}$, we will simply denote this by $\mu(\mathcal{X}'')$. When working with a set of samples from $\mathcal{D}$, we will use $\widehat{\mu}$ to denote the empirical version of $\mu$.

**Annuli, Slabs, and Affine Half-Spaces** We will often work with regions restricted to annuli, slabs, and affine halfspaces. Given direction $\mathbf{w}$ and threshold $\tau$, let $\mathcal{A}(\mathbf{w}, \tau)$, $\mathcal{S}(\mathbf{w}, \tau)$, $\mathcal{H}^+(\mathbf{w}, \tau)$ and $\mathcal{H}^-(\mathbf{w}, \tau)$ denote the set of points $\{\mathbf{x} \in \mathcal{X} : |\langle \mathbf{w}, \mathbf{x}\rangle| \geq \tau\}$, $\{\mathbf{x} \in \mathcal{X} : |\langle \mathbf{w}, \mathbf{x}\rangle| < \tau\}$, $\{\mathbf{x} \in \mathcal{X} : \langle \mathbf{w}, \mathbf{x}\rangle \geq \tau\}$, and $\{\mathbf{x} \in \mathcal{X} : \langle \mathbf{w}, \mathbf{x}\rangle \leq -\tau\}$ respectively.

## B.3 Miscellaneous Tools

**Lemma B.5** ([B+15, H+16]). *Given convex function $L$ and initialization $\mathbf{w}^{(0)} = 0$, consider projected SGD updates given by $\mathbf{w}^{(t+1)} = \Pi(\mathbf{w}^{(t)} - \nu \cdot \mathbf{v}^{(t)})$, where $\Pi$ is projection to the unit ball, $\mathbf{v}^{(t)}$ is a stochastic gradient such that $\mathbb{E}[\mathbf{v}^{(t)} \mid \mathbf{w}^{(t)}]$ is a subgradient of $L$ at $\mathbf{w}^{(t)}$ and $\|v^{(t)}\| \leq 1$ almost surely. For any $\epsilon, \delta > 0$, for $T = \Omega(\log(1/\delta)/\epsilon^2)$ and learning rate $\nu = 1/\sqrt{T}$, the average iterate $\overline{w} \triangleq \frac{1}{T}\sum_{t=1}^T \mathbf{w}^{(t)}$ satisfies $L(\overline{w}) \leq \min_{\mathbf{w}:\|\mathbf{w}\|\leq 1} L(\mathbf{w}) + \epsilon$ with probability at least $1 - \delta$.*

**Fact B.6.** *If $Z$ is a random variable that with probability $p$ takes on value $x$ and otherwise takes on value $y$, then $\mathbb{V}[Z] = (y - x)^2 \cdot p(1-p)$.*

**Theorem B.7** (Hoeffding's inequality, Theorem 2.2.2 of [Ver18]). *Suppose that $X_1, \ldots, X_n$ are independent mean-zero random variables and $|X_i| \leq K$ almost surely for all $i$. Then*

$$\text{Pr}\left[\left|\sum_{i=1}^n X_i\right| \geq t\right] \leq 2\exp\left(\frac{-t^2}{2nK^2}\right).$$

**Theorem B.8** (Bernstein's inequality, Theorem 2.8.4 of [Ver18]). *Suppose that $X_1, \ldots, X_n$ are independent mean-zero random variables and $|X_i| \leq K$ almost surely for all $i$. Then*

$$\text{Pr}\left[\left|\sum_{i=1}^n X_i\right| \geq t\right] \leq 2\exp\left(\frac{-t^2/2}{\sum_{i=1}^n \mathbb{V}[X_i] + Kt/3}\right).$$

**Outlier Removal** In the non-margin setting, and as in the previous works [BFKV98, Coh97, DGT19], we will rely upon outlier removal as a basic subroutine. We state the result we use here. Here $A \preceq B$ denotes the PSD ordering on matrices, i.e. $A \preceq B$ when $B - A$ is positive semidefinite.

**Theorem B.9** (Theorem 1 of [DV04]). *Suppose $\mathcal{D}$ is a distribution over $\mathbb{R}^d$ supported on b-bit integer vectors, and $\epsilon, \delta > 0$. There is an algorithm which requires $\tilde{O}(d^2 b/\epsilon)$ samples, runs in time $poly(d, b, 1/\epsilon, 1/\delta)$, and with probability at least $1 - \delta$, outputs an ellipsoid $\mathcal{E} = \{\mathbf{x} : \mathbf{x}^T A \mathbf{x} \leq 1\}$ where $A \succeq 0$ such that:*

1. *$\Pr_{\mathbf{X} \sim \mathcal{D}}[\mathbf{X} \in \mathcal{E}] \geq 1 - \epsilon$.*

2. *For any $\mathbf{w}$, $\sup_{\mathbf{x} \in \mathcal{E}} \langle \mathbf{w}, \mathbf{x} \rangle^2 \leq \beta \, \mathbb{E}_{\mathbf{X} \sim \mathcal{D}}[\langle \mathbf{w}, \mathbf{X} \rangle^2 \mid \mathbf{X} \in \mathcal{E}]$.*

*where $\beta = \tilde{O}(db/\epsilon)$.*

Since our algorithms rely on this result in the non-margin setting, they will have polynomial dependence on the bit-complexity of the input; this is expected, because even in the setting of no noise (linear programming) all known algorithms have runtime depending polynomially on the bit complexity of the input and improving this is a long-standing open problem.

## C  Properly Learning Halfspaces Under Massart Noise

In light of the game-theoretic perspective explained in the technical overview (i.e. (3)), the main problem we need to solve in order to (properly) learn halfspaces in the Massart model is to develop efficient strategies for the filtering player. While in the overview we viewed the output of the filtering player as being the actual filter distribution $c(\mathbf{X})$, to take into account finite sample issues we will need to combine this with the analysis of estimating the LeakyRelu gradient under this reweighted measure.

In the first two subsections below, we show how to formalize the plan outlined in the technical overview and show how they yield separating hyperplanes (sometimes with additional structure) for the $\mathbf{w}$ player to learn from; we then describe how to instantiate the max player with either gradient-descent or cutting-plane based optimization methods in order to solve the Massart halfspace learning problem.

Our most practical algorithm, FILTERTRON, uses gradient descent and works in the margin setting. In the non-margin setting, we instead use a cutting plane method and explain how our approach relates to the work of Cohen [Coh97]. We also, for sample complexity reasons, consider a cutting plane-based variant of FILTERTRON. This variant achieves a rate of $\epsilon = \tilde{O}(1/n^{1/3})$ in the $\eta + \epsilon$ guarantee, where $n$ is the number of samples and we suppress the dependence on other parameters. This significantly improves the $\tilde{O}(1/n^{1/5})$ rate of [DGT19], but does not yet match the optimal $\Theta(1/n^{1/2})$ rate achieved by inefficient algorithms [MN+06].

### C.1  Separation Oracle for Halfspaces with Margin

In this section, we show how to extract a separation oracle for any $\mathbf{w}$ with zero-one loss larger than $\eta$. In fact, the separating hyperplane is guaranteed to have a margin between $\mathbf{w}$ and $\mathbf{w}^*$, which means it can work as the descent direction in gradient descent.

**Theorem C.1.** *Suppose that $(\mathbf{X}, Y)$ are jointly distributed according to an $\eta$-Massart halfspace model with true halfspace $w^*$ and margin $\gamma$, and suppose that $\lambda \in [\eta + \epsilon, 1/2]$ for some $\epsilon > 0$. Suppose that $\mathbf{w}$ is a vector in the unit ball of $\mathbb{R}^d$ such that*

$$\text{err}(\mathbf{w}) \geq \lambda + 2\epsilon.$$

*Then Algorithm FINDDESCENTDIRECTION$(\mathbf{w}, \epsilon, \delta, \lambda)$ runs in sample complexity $m = O(\log(2/\delta)/\epsilon^3 \gamma^2)$, time complexity $O(md + m \log(m))$, and with probability at least $1 - \delta$ outputs a vector $\mathbf{g}$ such that $\|\mathbf{g}\| \leq 1$ and*

$$\langle \mathbf{w}^* - \mathbf{w}, -\mathbf{g} \rangle \geq \gamma \epsilon/8.$$

Recall from the preliminaries that $\mathcal{S}(\mathbf{w}, \tau) = \{\mathbf{x} : |\langle \mathbf{w}, \mathbf{x} \rangle| < \tau\}$ and that $L_\lambda^{\mathcal{X}'}(\mathbf{w})$ denotes the LeakyRelu loss over the distribution conditioned on $\mathbf{X} \in \mathcal{X}'$. In the algorithm we refer to the gradient of the LeakyRelu loss, which is convex but not differentiable everywhere – when the loss is not differentiable, the algorithm is free to choose any subgradient. On the other hand, while $\mathbf{g}$ is a

---
**Algorithm 3:** FINDDESCENTDIRECTION($\mathbf{w}, \epsilon, \delta, \lambda$)
---
1 Define empirical distribution $\hat{\mathcal{D}}$ from $m = O(\log(2/\delta)/\epsilon^3\gamma^2)$ samples, and let $\hat{L}_\lambda$ denote the LeakyRelu loss with respect to $\hat{\mathcal{D}}$.
2 Let $R = \{r > 0 : \text{Pr}_{\hat{\mathcal{D}}}[\mathbf{X} \in \mathcal{S}(\mathbf{w}, r)] \geq \epsilon\}$.
3 Let $r^* = \text{argmax}_{r \in R} \hat{L}_\lambda^{\mathcal{S}(\mathbf{w},r)}(\mathbf{w})$.
4 Return $\mathbf{g} = \nabla \hat{L}_\lambda^{\mathcal{S}(\mathbf{w},r^*)}(\mathbf{w})$.
---

subgradient of $\max_{r \in R} \hat{L}_\lambda^{\mathcal{S}(\mathbf{w},r)}(w)$ we are careful to ensure $\mathbf{g}$ is also the subgradient of a particular function $\hat{L}_\lambda^{\mathcal{S}(\mathbf{w},r^*)}(w)$, which may not be true for arbitrary subgradients of the maximum.

**Lemma C.2** (Lemma 2.3 of [DGT19]). *Suppose that* $(\mathbf{X}, Y)$ *are jointly distributed according to an* $\eta$-*Massart halfspace model with true halfspace* $\mathbf{w}^*$ *and margin* $\gamma$. *Then for any* $\lambda \geq \eta$,

$$L_\lambda(\mathbf{w}^*) \leq -\gamma(\lambda - \text{err}(\mathbf{w}^*)).$$

*Proof.* We include the proof from [DGT19] for completeness. From the definition,

$$
\begin{aligned}
L_\lambda(\mathbf{w}^*) &= \mathbb{E}\left[\left(\frac{1}{2}\text{sgn}(-\langle\mathbf{w}^*, \mathbf{X}\rangle Y) + \frac{1}{2} - \lambda\right)|\langle\mathbf{w}^*, \mathbf{X}\rangle|\right] \\
&= \mathbb{E}[(\mathbb{1}[\text{sgn}(\langle\mathbf{w}^*, \mathbf{X}\rangle) \neq Y] - \lambda)|\langle\mathbf{w}^*, \mathbf{X}\rangle|] \\
&= \mathbb{E}[(\text{Pr}[\text{sgn}(\langle\mathbf{w}^*, \mathbf{X}\rangle) \neq Y \mid \mathbf{X}] - \lambda)|\langle\mathbf{w}^*, \mathbf{X}\rangle|] \\
&\leq -\gamma\,\mathbb{E}[\lambda - \text{Pr}[\text{sgn}(\langle\mathbf{w}^*, \mathbf{X}\rangle) \neq Y \mid \mathbf{X}]] = -\gamma(\lambda - \text{err}(\mathbf{w}^*))
\end{aligned}
$$

where in the second equality we used the law of total expectation, and in the last inequality we used that $\text{Pr}[\text{sgn}(\langle\mathbf{w}^*, \mathbf{X}\rangle) \neq Y \mid \mathbf{X}] \leq \eta \leq \lambda$ by assumption and $|\langle\mathbf{w}^*, \mathbf{X}\rangle| \geq \gamma$ by the margin assumption. $\qquad\square$

Based on Lemma C.2, it will follow from Bernstein's inequality that for any particular $r$, $\hat{L}_\lambda^{\mathcal{S}(\mathbf{w},r)}(\mathbf{w}^*)$ will be negative with high probability as long as we take $\Omega(1/\epsilon\gamma^2(\lambda - \text{err}(w^*))^2)$ samples. To prove the algorithm works, we show that given this many samples, the bound actually holds uniformly over all $r \in R$. This requires a chaining argument; we leave the proof to Appendix H. Note that $\gamma$ in this Lemma plays the role of $\gamma(\lambda - \text{err}(w^*))$ in the previous Lemma.

**Lemma C.3.** *Let* $\delta > 0, \epsilon > 0$ *be arbitrary. Suppose that for all* $r \geq 0$ *such that* $\text{Pr}_{\mathcal{D}}[\mathbf{X} \in \mathcal{S}(\mathbf{w}, r)] \geq \epsilon/2$,

$$L_\lambda^{\mathcal{S}(\mathbf{w},r)}(\mathbf{w}^*) \leq -\gamma$$

*for some* $\gamma > 0$, *with respect to distribution* $\mathcal{D}$. *Suppose* $\hat{\mathcal{D}}$ *is the empirical distribution formed from* $n$ *i.i.d. samples from* $\mathcal{D}$. *Then with probability at least* $1 - \delta$, *for any* $r \in R$ *(where* $R = \{r > 0 : \text{Pr}_{\hat{\mathcal{D}}}[\mathbf{X} \in \mathcal{S}(\mathbf{w}, r)] \geq \epsilon\}$ *as in Algorithm* FINDDESCENTDIRECTION*),*

$$\sup_{r \in \mathcal{R}} \hat{L}_\lambda^{\mathcal{S}(\mathbf{w},r)}(\mathbf{w}^*) \leq -\gamma/4$$

*as long as* $n = \Omega\left(\frac{\log(2/\delta)}{\epsilon\gamma^2}\right)$.

The last thing we will need to establish is a lower bound on $\sup_{r \in R} \hat{L}_\lambda^{\mathcal{S}(\mathbf{w},r)}(\mathbf{w})$. First we establish such a bound for the population version $L_\lambda^{\mathcal{S}(\mathbf{w},r)}(\mathbf{w})$.

**Lemma C.4.** *Suppose that* $(\mathbf{X}, Y) \sim \mathcal{D}$ *with* $\mathbf{X}$ *valued in* $\mathbb{R}^d$ *and* $Y$ *valued in* $\{\pm 1\}$. *Suppose that* $\mathbf{w}$ *is a vector in the unit ball of* $\mathbb{R}^d$ *such that*

$$\text{err}(\mathbf{w}) \geq \lambda + 2\epsilon$$

*for some* $\lambda, \epsilon \geq 0$. *Then there exists* $r \geq 0$ *such that* $\text{Pr}_{\mathcal{D}}[\mathbf{X} \in \mathcal{S}(\mathbf{w}, r)] \geq 2\epsilon$ *and* $L_\lambda^{\mathcal{S}(\mathbf{w},r)}(\mathbf{w}) \geq 0$.

*Proof.* Proof by contradiction — suppose there is no such $r$. Define $r_0 = \min\{r : \Pr[\mathbf{X} \in \mathcal{S}(\mathbf{w}, r)] \geq 2\epsilon\}$, which exists because the CDF is always right-continuous. Then for all $r \geq r_0$, we have

$$0 > L_\lambda^{\mathcal{S}(\mathbf{w},r)}(w) \Pr[\mathbf{X} \in \mathcal{S}(\mathbf{w}, r)] = \mathbb{E}[\ell_\lambda(\mathbf{w}, r) \cdot \mathbb{1}[|\langle w, \mathbf{X}\rangle| \leq r]]$$
$$= \mathbb{E}[(\mathbb{1}[\mathrm{sgn}(\langle \mathbf{w}, \mathbf{X}\rangle) \neq Y] - \lambda)|\langle w, \mathbf{X}\rangle| \cdot \mathbb{1}[|\langle w, \mathbf{X}\rangle| \leq r]]$$
$$= \int_0^r \mathbb{E}[(\Pr[\mathrm{sgn}(\langle \mathbf{w}, \mathbf{X}\rangle) \neq Y \mid X] - \lambda) \cdot \mathbb{1}[|\langle w, \mathbf{X}\rangle| \in (s, r]]]ds$$

where in the second equality we used the same rewrite of the LeakyRelu function as in the proof of Lemma C.2, the third equality uses $x = \int_0^\infty \mathbb{1}[y < x]dy$ for $x > 0$ as well as the law of total expectation. Therefore, for every $r \geq r_0$ there exists $s(r) < r$ such that

$$0 > \mathbb{E}[(\Pr[\mathrm{sgn}(\langle \mathbf{w}, \mathbf{X}\rangle) \neq Y \mid X] - \lambda) \cdot \mathbb{1}[|\langle w, \mathbf{X}\rangle| \in (s(r), r]]].$$

Rearranging and dividing by $\Pr[|\langle w, X\rangle| \in (s(r), r]] > 0$ gives

$$\lambda > \Pr[\mathrm{sgn}(\langle \mathbf{w}, \mathbf{X}\rangle) \neq Y \mid |\langle w, X\rangle| \in (s(r), r]]. \tag{5}$$

Define $R' = \{r \geq 0 : \lambda > \Pr[\mathrm{sgn}(\langle \mathbf{w}, \mathbf{X}\rangle) \neq Y \mid |\langle w, X\rangle| > r]\}$ and define $r' = \inf R'$. We claim that $r' < r_0$; otherwise, applying (5) with $r = r'$ shows that $s(r') \in R'$ but $s(r') < r'$, which contradicts the definition of $r'$. Let $r_1 \in (r', r_0)$ so $r_1 \in R^*$. Then

$$\Pr[\mathrm{sgn}(\langle w, X\rangle) \neq Y] = \Pr[\mathrm{sgn}(\langle w, X\rangle) \neq Y \mid |\langle w, X\rangle| \leq r_1] \Pr[|\langle w, \mathbf{X}\rangle| \leq r_1]$$
$$+ \Pr[\mathrm{sgn}(\langle w, X\rangle) \neq Y \mid |\langle w, X\rangle| > r_1] \Pr[|\langle w, \mathbf{X}\rangle| > r_1]$$
$$< 2\epsilon + \lambda.$$

by the fact $r_1 \in (r', r_0)$ and the definition of $r'$ and $r_0$. This contradicts the assumption that $\Pr[\mathrm{sgn}(\langle w, X\rangle) \neq Y] \geq \lambda + 2\epsilon$. $\qquad\square$

Given these Lemmas, we can proceed to the proof of the Theorem.

*Proof of Theorem C.1.* By convexity,

$$\hat{L}_\lambda^{\mathcal{S}(\mathbf{w},r^*)}(\mathbf{w}) + \langle \nabla \hat{L}_\lambda^{\mathcal{S}(\mathbf{w},r^*)}(\mathbf{w}), \mathbf{w}^* - \mathbf{w}\rangle \leq \hat{L}_\lambda^{\mathcal{S}(\mathbf{w},r^*)}(\mathbf{w}^*)$$

so rearranging and using the definition of $\mathbf{g}$ gives

$$\hat{L}_\lambda^{\mathcal{S}(\mathbf{w},r^*)}(\mathbf{w}) - \hat{L}_\lambda^{\mathcal{S}(\mathbf{w},r^*)}(\mathbf{w}^*) \leq \langle -\mathbf{g}, \mathbf{w}^* - \mathbf{w}\rangle. \tag{6}$$

It remains to lower bound the left hand side. By Lemma C.2 we know that for all $r$ such that $\Pr[\mathbf{X} \in \mathcal{S}(\mathbf{w}, r^*)] > 0$, we have $L_\lambda^{\mathcal{S}(\mathbf{w},r^*)}(\mathbf{w}^*) \leq -\gamma(\lambda - \eta) \leq -\gamma\epsilon$ hence by Lemma C.3 we have

$$\inf_{r \in R} -\hat{L}_\lambda^{\mathcal{S}(\mathbf{w},r)}(\mathbf{w}^*) \geq \gamma\epsilon/4 \tag{7}$$

with probability at least $1 - \delta/2$ provides that the number of samples is $m = \Omega(\log(2/\delta)/\epsilon^3\gamma^2)$.

From Lemma C.4 we know that there exists $r$ such that $\Pr_\mathcal{D}[\mathbf{X} \in \mathcal{S}(\mathbf{w}, r)] \geq 2\epsilon$ and $L_\lambda^{\mathcal{S}(\mathbf{w},r)}(\mathbf{w}) \geq 0$. From two applications of Bernstein's inequality (as in the proof of Lemma C.3, see Appendix H), it follows that as long as $m = \Omega(\log(2/\delta)/\epsilon^3\gamma^2)$ then with probability at least $1 - \delta/2$ we have $\Pr_\mathcal{D}[\mathbf{X} \in \mathcal{S}(\mathbf{w}, r)] \geq \epsilon$ and $L_\lambda^{\mathcal{S}(\mathbf{w},r)}(\mathbf{w}) \geq -\gamma\epsilon/8$. Under this event we have that $r \in R$ and so by the definition of $r^*$,

$$\hat{L}_\lambda^{\mathcal{S}(\mathbf{w},r^*)}(\mathbf{w}) \geq L_\lambda^{\mathcal{S}(\mathbf{w},r)}(\mathbf{w}) \geq -\gamma\epsilon/8. \tag{8}$$

Combining (6),(7), and (8) proves the result. The fact that $\mathbf{g}$ is bounded norm follows from the assumptions that $\|\mathbf{X}\| \leq 1$ and $|\mathbf{w}| \leq 1$; the runtime guarantee follows since all steps can be efficiently implemented if we first sort the samples by their inner product with $\mathbf{w}$. $\qquad\square$

## C.2 Separation Oracle for General Halfspaces

In this section, we show how to extract a separation oracle for any $w$ with zero-one loss larger than $\eta$ without any dependence on the margin parameter; instead, our guarantee has polynomial dependence on the bit-complexity of the input. We first give a high-level overview of the separation oracle and how it fits into the minimax scheme 3.

The generalized filter we use does the following: (1) it rescales $\mathbf{X}$ by $\frac{1}{|\langle \mathbf{w}, \mathbf{X} \rangle|}$ (to emphasize points near the current hyperplane) and (2) applies outlier removal [BFKV98, DV04] to the rescaled $\mathbf{X}$ to produce the final filter. Given $c$, the max player uses the gradient of the LeakyRelu loss as a cutting plane to eliminate bad strategies $\mathbf{w}$ and produce its next iterate.

**Remark C.5.** *The generalized filter used in this section is also implicitly used in the algorithm of [Coh97] in the construction of a separation oracle in the RCN setting. Remarkably, the separating hyperplane output by the algorithm of [Coh97] is also very similar to the gradient of the LeakyRelu loss,* except *that it has an additional rescaling factor between the terms with positive and negative signs for $\langle \mathbf{w}, \mathbf{X} \rangle Y$, which apparently causes difficulties for analyzing the algorithm in the Massart setting [Coh97]. Our analysis shows that there is no issue if we delete this factor.*

In our algorithm we assume that $\langle \mathbf{w}, \mathbf{X} \rangle \neq 0$ almost surely; this assumption is w.l.o.g. if we slightly perturb either $\mathbf{w}$ or $\mathbf{X}$, as argued in [Coh97]. Thus it remains to exhibit the separation oracle $\mathcal{O}(\cdot)$. The following theorem provides the formal guarantees for the algorithm GENERALHALFSPACEORACLE($\mathbf{w}, \epsilon, \delta, \lambda$).

**Theorem C.6.** *Suppose that $(\mathbf{X}, Y)$ are jointly distributed according to an $\eta$-Massart halfspace model with true halfspace $\mathbf{w}^*$, and suppose that $\lambda \in [\eta + \epsilon, 1/2]$ for some $\epsilon > 0$. Suppose that $\mathbf{w}$ is a vector in the unit ball of $\mathbb{R}^d$ such that*

$$\mathrm{err}(\mathbf{w}) \geq \lambda + \epsilon.$$

*Then Algorithm GENERALHALFSPACEORACLE($\mathbf{w}, \epsilon, \delta, \lambda$) runs in sample complexity $n = \tilde{O}\left(\frac{db}{\epsilon^3}\right)$, time complexity $\mathrm{poly}(d, b, \frac{1}{\epsilon}, \frac{1}{\delta})$, and with probability at least $1 - \delta$ outputs a vector $\mathbf{g}$ such that $\langle -\mathbf{g}, \mathbf{w}^* - \mathbf{w} \rangle > 0$. More generally, $\langle -\mathbf{g}, \mathbf{w}' - \mathbf{w} \rangle > 0$ with probability at least $1 - \delta$ for any fixed $\mathbf{w}'$ such that either $\mathbf{w}' = 0$ or $\mathrm{sgn}(\langle \mathbf{w}', \mathbf{X} \rangle) = \mathrm{sgn}(\langle \mathbf{w}^*, \mathbf{X} \rangle)$ almost surely.*

*Proof.* First, we note that the joint law of the rescaled distribution $\overline{\mathcal{D}}$ still follows the $\eta$-Massart halfspace model. This is because $\mathrm{sgn}(\langle \mathbf{w}^*, \mathbf{x} \rangle) = \mathrm{sgn}(\langle \mathbf{w}, \frac{\mathbf{x}}{|\langle \mathbf{w}, \mathbf{x} \rangle|} \rangle)$ for any $\mathbf{x}$, i.e. rescaling does not affect the law of $Y \mid \mathbf{X}$.

By convexity,

$$\hat{L}_\lambda(\mathbf{w}^*) \geq \hat{L}_\lambda(\mathbf{w}) + \langle \nabla \hat{L}(\mathbf{w}), \mathbf{w}^* - \mathbf{w} \rangle = \hat{L}_\lambda(\mathbf{w}) + \langle \mathbf{g}, \mathbf{w}^* - \mathbf{w} \rangle$$

so by rearranging, we see that $\langle -\mathbf{g}, \mathbf{w}^* - \mathbf{w} \rangle \geq \hat{L}_\lambda(\mathbf{w}) - \hat{L}_\lambda(\mathbf{w}^*)$, so it suffices to show the difference of the losses is nonnegative. We prove this by showing that $\hat{L}_\lambda(\mathbf{w}) > 0$ and $\hat{L}_\lambda(\mathbf{w}^*) \leq 0$ with high probability.

First, we condition on the success case in Theorem B.9, which occurs with probability at least $1 - \delta/2$. Next, we prove that $\hat{L}_\lambda(\mathbf{w}) \geq 0$ with probability at least $1 - \delta/4$. Let $\bar{L}_\lambda$ denote the LeakyRelu loss over $\mathcal{D}$. Note that by the same rewrite of the LeakyRelu loss as in Lemma C.2, we have

$$\bar{L}_\lambda^\mathcal{E}(\mathbf{w}) \geq (\mathrm{err}(\mathbf{w}) - \epsilon/2 - \lambda) \geq \epsilon/2.$$

The first inequality follows from $|\langle \mathbf{w}, \mathbf{X} \rangle| = 1$ under $\overline{\mathcal{D}}$ and that the outlier removal step (Theorem B.9) removes no more than $\epsilon/2$ fraction of $\hat{\mathcal{D}}$. The last inequality follows by assumption $\mathrm{err}(\mathbf{w}) \geq \lambda + \epsilon$. Then by Hoeffding's inequality (Theorem B.7), it follows that $\hat{L}_\lambda(\mathbf{w}) \geq \epsilon/4$ with probability $1 - \delta/4$ as long as $n = \Omega\left(\frac{\log(2/\delta)}{\epsilon^2}\right)$.

Finally, we prove $\hat{L}_\lambda(\mathbf{w}) \leq 0$ with probability $1 - \delta/4$. Define $\overline{\mathcal{D}}_\mathcal{E}$ to be $\mathcal{D}$ conditioned on $X \in \mathcal{E}$. By the same reasoning as Lemma C.3, we have

$$\overline{L}_\lambda^\mathcal{E}(\mathbf{w}^*) = \mathbb{E}_{(\mathbf{X}, Y) \sim \overline{\mathcal{D}}}[(\mathrm{Pr}[\mathrm{sgn}(\langle \mathbf{w}^*, \mathbf{X} \rangle \neq Y] - \lambda)|\langle \mathbf{w}^*, \mathbf{X} \rangle|] \leq -\epsilon \mathbb{E}_{\mathbf{X} \sim \overline{\mathcal{D}}}[|\langle \mathbf{w}^*, \mathbf{X} \rangle|].$$

Observe that from the guarantee of Theorem B.9,

$$\mathbb{E}_{\mathbf{X}\sim\bar{\mathcal{D}}_{\mathcal{E}}}[|\langle\mathbf{w}^*,\mathbf{X}\rangle|^2] \leq \mathbb{E}_{\mathbf{X}\sim\bar{\mathcal{D}}_{\mathcal{E}}}[|\langle\mathbf{w}^*,\mathbf{X}\rangle|]\sup_{\mathbf{x}\in\mathcal{E}}|\langle\mathbf{w}^*,\mathbf{x}\rangle| \leq \mathbb{E}_{\mathbf{X}\sim\bar{\mathcal{D}}}[|\langle\mathbf{w}^*,\mathbf{X}\rangle|]\sqrt{\beta\mathbb{E}_{\bar{\mathcal{D}}_{\mathcal{E}}}|\langle\mathbf{w}^*,\mathbf{X}\rangle|^2}$$

so $\mathbb{E}_{\mathbf{X}\sim\bar{\mathcal{D}}_{\mathcal{E}}}[|\langle\mathbf{w}^*,\mathbf{X}\rangle|^2] \leq \beta\mathbb{E}_{\mathbf{X}\sim\bar{\mathcal{D}}}[|\langle\mathbf{w}^*,\mathbf{X}\rangle|]^2$. It follows that

$$\overline{L}_\lambda^{\mathcal{E}}(\mathbf{w}^*) \leq (-\epsilon/\sqrt{\beta})\sqrt{\mathbb{E}_{\bar{\mathcal{D}}_{\mathcal{E}}}[|\langle\mathbf{w}^*,\mathbf{X}\rangle|^2]}.$$

Define $\sigma = \sqrt{\mathbb{E}_{\bar{\mathcal{D}}_{\mathcal{E}}}[|\langle\mathbf{w}^*,\mathbf{X}\rangle|^2]}$. By Bernstein's inequality (Theorem B.8), if we condition on at least $m$ samples falling into $\mathcal{E}$ then

$$\Pr[\hat{L}_\lambda^{\mathcal{E}}(\mathbf{w}^*) \geq -\frac{\epsilon\sigma}{2\sqrt{\beta}}] \leq 2\exp\left(-\frac{cm^2(\epsilon\sigma)^2/\beta}{m\sigma^2 + m\epsilon\sigma^2}\right) = 2\exp\left(-c'm\epsilon^2/\beta\right)$$

where $c,c' > 0$ are absolute constants. Therefore if $m = \Omega\left(\frac{\beta}{\epsilon^2}\log(2/\delta)\right)$, we have with probability $1 - \delta/16$ that $\hat{L}_\lambda^{\mathcal{E}}(\mathbf{w}^*) \leq -\frac{\epsilon\sigma}{2\sqrt{\beta}}$. Using Hoeffding's inequality to ensure $m$ is sufficiently large, the same conclusion holds with probability at least $1 - \delta/4$ assuming $n = \Omega\left(\frac{\beta}{\epsilon^2}\log(2/\delta)\right)$.

Using the union bound and combining the estimates, we see that $\langle-\mathbf{g},\mathbf{w}^*-\mathbf{w}\rangle \geq \hat{L}_\lambda(\mathbf{w})-\hat{L}_\lambda(\mathbf{w}^*) > \epsilon/4$ with probability at least $1 - \delta$, as long as $n = \Omega\left(\frac{\beta}{\epsilon^2}\log(2/\delta)\right)$; recalling that $\beta = \tilde{O}(db/\epsilon)$ gives the result. The same argument applies to $\mathbf{w}' = 0$ since $\hat{L}_\lambda(0) = 0$, and also to any fixed $\mathbf{w}'$ such that $\text{sgn}(\langle\mathbf{w}',\mathbf{X}\rangle) = \text{sgn}(\langle\mathbf{w}^*,\mathbf{X}\rangle)$.

$\square$

---

**Algorithm 4:** GENERALHALFSPACEORACLE($\mathbf{w},\epsilon,\delta,\lambda$)

---

1 Define $\overline{\mathcal{D}}$ to be the joint distribution of $\left(\frac{\mathbf{X}}{|\langle\mathbf{w},\mathbf{X}\rangle|}, Y\right)$ when $(\mathbf{X}, Y) \sim \mathcal{D}$.

2 Define $\mathcal{E}$ by applying Theorem B.9 with distribution $\overline{\mathcal{D}}_{\mathbf{X}}$, taking $\epsilon' = \epsilon/2$ and $\delta' = \delta/2$.

3 Form empirical distribution $\hat{\mathcal{D}}$ from $O\left(\frac{\beta^2}{\epsilon^2}\log(2/\delta)\right)$ samples of $\overline{\mathcal{D}}$, and let $\hat{L}_\lambda$ be the LeakyRelu loss over this empirical distribution.

4 Return $\mathbf{g} = \nabla\hat{L}_\lambda^{\mathcal{E}}(\mathbf{w})$

---

### C.3 Gradient Descent Learner

We first develop the gradient-based learner for the margin setting. The learner is not doing gradient descent on a fixed convex function; in fact, it is essentially running gradient descent on the nonconvex loss $\max_{r\in R} L_\lambda(\mathbf{w})$ where $R$ depends on $w$ (and is defined in FINDDESCENTDIRECTION). Nevertheless, we can prove our algorithm works using the low regret guarantee for projected gradient descent. Recall that projected gradient descent over closed convex set $\mathcal{K}$ with step size sequence $\beta_1,\ldots,\beta_T$ and vectors $\mathbf{g}_1,\ldots,\mathbf{g}_T$ is defined by the iteration

$$\mathbf{x}_{t+1} = \Pi_\mathcal{K}(\mathbf{x}_t - \beta_t\mathbf{g}_t)$$

where $\Pi_\mathcal{K}(y) = \text{argmin}_{x\in\mathcal{K}}\|x - y\|$ is the Euclidean projection onto $\mathcal{K}$, and $\mathbf{x}_1$ is an arbitrary point in $\mathcal{K}$. In the case $\mathcal{K}$ is the unit ball, $\Pi_\mathcal{K}(y) = \frac{y}{\max(1,\|y\|)}$.

**Theorem C.7** (Theorem 3.1 of [H+16], [Zin03]). *Suppose that $\mathcal{K}$ is a closed convex set such that $\max_{x,y\in\mathcal{K}}\|x - y\| \leq D$. Suppose that the sequence of vectors $\mathbf{g}_1,\ldots,\mathbf{g}_T$ satisfy $\|\mathbf{g}_t\| \leq G$. Then projected online gradient descent with step size $\beta_t = \frac{D}{G\sqrt{t}}$ outputs a sequence $\mathbf{x}_1,\ldots,\mathbf{x}_T \in \mathcal{K}$ satisfying*

$$\sum_{t=1}^T\langle\mathbf{g}_t,\mathbf{x}_t-\mathbf{x}^*\rangle \leq \frac{3}{2}GD\sqrt{T}$$

*for any $\mathbf{x}^* \in \mathcal{K}$.*

---

**Algorithm 5:** FILTERTRON($\epsilon, \eta, \delta, \lambda, T$)

---

**1** Let $\mathbf{w}_1$ be an arbitrary vector in the unit ball.

**2** Build an empirical distribution $\hat{\mathcal{H}}$ from $m = \Omega(\log(T/\delta)/\epsilon^2)$ samples (to use as a test set).

**3** **for** $t = 1$ *to* $T$ **do**

**4**     **if** $\hat{\text{err}}(\mathbf{w}_t) < \eta + \epsilon/2$ **then**

**5**        Return $\mathbf{w}_t$.

**6**     **else**

**7**        Let $\beta_t = \frac{1}{\sqrt{t}}$.

**8**        Let $\mathbf{g}_t = \text{FINDDESCENTDIRECTION}(\mathbf{w}_t, \epsilon/6, \delta/2T, \lambda)$.

**9**        Let $\mathbf{w}_{t+1} = \frac{\mathbf{w}_t - \beta_t \mathbf{g}_t}{\max(1, \|\mathbf{w}_t - \beta_t \mathbf{g}_t\|)}$.

---

**Theorem C.8.** *Suppose that* $(\mathbf{X}, Y)$ *is an* $\eta$-*Massart halfspace model with margin* $\gamma$. *With probability at least* $1 - \delta$, *Algorithm* FILTERTRON *returns* $\mathbf{w}$ *such that*

$$\text{err}(\mathbf{w}) \leq \eta + \epsilon$$

*when* $T = \frac{145}{\gamma^2 \epsilon^2}$ *and* $\lambda = \eta + \epsilon/6$. *The algorithm runs in total sample complexity* $n = O\left(\frac{\log(2/\delta\gamma\epsilon)}{\gamma^4 \epsilon^5}\right)$ *and runtime* $O(nd + n \log n)$.

*Proof.* First we note that the algorithm as written always returns the same $\mathbf{w}_t$ as a modified algorithm which: (1) first produces iterates $\mathbf{w}_1, \ldots, \mathbf{w}_T$ without looking at the test set, i.e. ignores the if-statement on line 4 and (2) then returns the first $\mathbf{w}_t$ which achieves error at most $\eta + \epsilon$ on the test set $\hat{\mathcal{H}}$. So we analyze this variant of the algorithm, which makes it clear that the $\mathbf{w}_t$ will be independent of the test set $\hat{\mathcal{H}}$. Requiring that $m = \Omega(\log(T/\delta)/\epsilon^2)$, we then see by Bernstein's inequality (Theorem B.8) and the union bound that if $\mathbf{w}_t$ has test error at most $\eta + \epsilon/2$, then $\text{err}(\mathbf{w}_t) \leq \eta + \epsilon$ with probability at least $1 - \delta/2$.

It remains to handle the failure case where the algorithm does not return any $\mathbf{w}_t$. In this case, applying Theorem C.1 and the union bound, it holds with probability at least $1 - \delta/2$ that

$$\langle \mathbf{w}^* - \mathbf{w}_t, -\mathbf{g}_t \rangle \geq \gamma \epsilon/8.$$

In this case, applying the regret inequality from Theorem C.7 shows that

$$\gamma \epsilon T/8 \leq \sum_{t=1}^{T} \langle \mathbf{w}^* - \mathbf{w}_t, -\mathbf{g}_t \rangle \leq \frac{3}{2} \sqrt{T}$$

which gives a contradiction plugging in $T = \frac{145}{\gamma^2 \epsilon^2}$. Therefore, the algorithm fails only when one of the events above does not occur, which by the union bound happens with probability at most $\delta$. $\quad\square$

**Remark C.9.** *It is possible to prove that Algorithm* FILTERTRON *succeeds with constant step size and similar guarantees using additional facts about the LeakyRelu loss. We omit the details.*

**Remark C.10.** *Algorithm* FILTERTRON *is compatible with the kernel trick and straightforward to "kernelize", because the algorithm depends only on inner products between training datapoints. See [KS09] for kernelization of a similarly structured algorithm.*

The FILTERTRON algorithm can easily be modified to work in general normed spaces, by replacing online gradient descent with online mirror descent, i.e. modifying the update step for $\mathbf{w}_{t+1}$ in line 9 of the algorithm. This generalization is often useful when working with sparsity — for example, [KM17] used the analogous generalization of the GLMTRON [KKSK11] to efficiently learn sparse graphical models. In the case of learning (Massart) halfspaces, there has been a lot of interest in the sparse setting for the purpose of performing *1-bit compressed sensing*: see [ABHZ16] and references within.

We state the result below; for details about mirror descent see the textbooks [B+15, H+16].

**Definition C.11.** *Let $\| \cdot \|$ be an arbitrary norm on $\mathbb{R}^d$, and let $\| \cdot \|_*$ denote its dual norm. We say that $\mathbf{w}^*$ has a $\gamma$-margin with respect to the norm $\| \cdot \|$ and random vector $\mathbf{X}$ if $\|\mathbf{w}^*\| \leq 1$, $\|\mathbf{X}\|_* \leq 1$ almost surely, and $|\langle \mathbf{w}^*, \mathbf{X}\rangle| \geq \gamma$ almost surely.*

**Theorem C.12.** *Suppose that $(\mathbf{X}, Y)$ is an $\eta$-Massart halfspace model with margin $\gamma$ with respect to (general) norm $\| \cdot \|$. Suppose that $\Phi$ is 1-strongly convex with respect to $\| \cdot \|$ over convex set $\mathcal{K}$ and $\sup_{x,y \in \mathcal{K}}(\Phi(x) - \Phi(y)) \leq D^2$. Then there exists an algorithm (FILTERTRON modified to use mirror descent steps with regularizer $\Phi$) which, with probability at least $1 - \delta$, returns $\mathbf{w}$ such that $\mathrm{err}(\mathbf{w}) \leq \eta + \epsilon$ with total sample complexity $O(\frac{D^2 \log(2/\delta\gamma\epsilon)}{\gamma^4\epsilon^5})$ and runtime $poly(1/\gamma, 1/\epsilon, d, D, \log(1/\delta))$.*

*Proof.* Under these assumptions, we can check that the gradients output by the separation oracle satisfy $\|\mathbf{g}\|_* = O(\|\mathbf{X}\|_*) = O(1)$, so mirror descent [B$^+$15, H$^+$16] guarantees a regret bound of the form $O(D\sqrt{T})$. Then the result follows in the same way as the proof of Theorem C.8. $\qquad\square$

For a concrete application, suppose we want to learn a Massart conjunction of $k$ variables over an arbitrary distribution on the hypercube $\{0,1\}^d$, up to error $\eta + \epsilon$. This corresponds an halfspace $\mathbf{w}^*$ with $\gamma = O(1/k)$ with respect to the $\ell_1$ norm. Choosing entropy as the regularizer specializes mirror descent to a simple multiplicative weights update [H$^+$16], and the resulting algorithm has sample complexity $O(\frac{k^4 \log(d) \log(2/\delta\gamma\epsilon)}{\epsilon^5})$; in particular, it only has a logarithmic dependence on the dimension $d$. In contrast, had we simply applied Theorem C.8, we would pick up an unecessary polynomial dependence on the dimension.

### C.4 Cutting-Plane Learners

**Margin Case.** In the analysis of Algorithm FILTERTRON, we let the algorithm use fresh samples in every iteration of the loop. This makes the analysis clean, however it is not ideal as far as the sample complexity is concerned. One option to improve it is to combine the analysis FILTERTRON with the separation oracle, using the same samples at every iteration, and bound the resulting sample complexity. A more modular approach, which we pursue here, is to replace gradient descent with a different optimization routine. This approach will also illustrate that the real sample complexity bottleneck (e.g. the rate in $\epsilon$) is the cost of running the separation oracle a single time. The algorithm we will use is due to Vaidya [Vai89], because its rate is optimal for convex optimization in fixed dimension, but any optimization method achieving this rate will work, see for example [JLSW20].

**Theorem C.13** ([Vai89], Section 2.3 of [B$^+$15]). *Suppose that $\mathcal{K}$ is an (unknown) convex body in $\mathbb{R}^d$ which contains a Euclidean ball of radius $r > 0$ and contained in a Euclidean ball centered at the origin of radius $R > 0$. There exists an algorithm which, given access to a separation oracle for $\mathcal{K}$, finds a point $\mathbf{x} \in \mathcal{K}$, runs in time $poly(\log(R/r), d)$, and makes $O(d\log(Rd/r))$ calls to the separation oracle.*

For this purpose we need a lower bound on $r$, which is straightforward to prove using the triangle inequality.

**Lemma C.14.** *Suppose that $\|\mathbf{w}^*\| = 1$ and $|\langle \mathbf{w}^*, \mathbf{X}\rangle| \geq \gamma$. Then for any $\mathbf{u}$ with $\|\mathbf{u}\| \leq c\gamma/4$, $|\langle \mathbf{w}^* + \mathbf{u}, \mathbf{X}\rangle| \geq (1-c)\gamma$.*

Taking $c = O(\epsilon\gamma)$ in the above Lemma and using that the cutting plane generated by Theorem C.1 always has a margin $\Omega(\epsilon\gamma)$, we can guarantee that the resulting halfspace always separates $\mathbf{w}$ and the ball of radius $c$ around $\mathbf{w}^*$. Therefore if we replace the gradient steps in Algorithm FILTERTRON by the update of Vaidya's algorithm (setting $R = 1 + c$), we can prove the following Theorem. In place of the regret inequality, we use that if the algorithm does not achieve small test error in the first $T - 1$ steps, then in step $T$ it must find a point in the ball of radius $c$ around $\mathbf{w}^*$ and this is an optimal predictor.

**Theorem C.15.** *Suppose that $(\mathbf{X}, Y)$ is an $\eta$-Massart halfspace model with margin $\gamma$. There exists an algorithm (FILTERTRON modified to use Vaidya's algorithm) such that with probability at least $1 - \delta$, it returns $\mathbf{w}$ such that*

$$\mathrm{err}(\mathbf{w}) \leq \eta + \epsilon$$

*when $T = O(d \log(d/\gamma\epsilon))$ and $\lambda = \eta + \epsilon/6$. The algorithm runs in total sample complexity $n = O\left(\frac{d\log(2/\delta\gamma)}{\gamma^2\epsilon^3}\right)$ and runtime $poly(n, d)$.*

**Remark C.16.** *We may always assume* $d = \tilde{O}(1/\gamma^2)$ *by preprocessing with the Johnson-Lindenstrauss Lemma [AV99]. Focusing on the dependence on* $\epsilon$*, the above guarantee of* $\tilde{O}(1/\epsilon^3)$ *significantly improves the* $\tilde{O}(1/\epsilon^5)$ *dependence of [DGT19]; in comparison the minimax optimal rate is* $\Theta(1/\epsilon^2)$ *[MN$^+$06]. (The lower bound holds even for RCN in one dimension and with margin 1, by considering testing between* $\mathbf{w} = \pm 1$ *when* $\mathbf{X} = 1$ *almost surely and* $\eta = 1/2 - \epsilon$*.) The remaining gap is unavoidable with this separation oracle, as we need to restrict to slabs of probability mass* $\epsilon$*, and this necessarily comes at the loss of the factor of* $O(1/\epsilon)$*. Interestingly, our algorithm for general* $d$*-dimensional halfspaces (Theorem C.18) also has a* $\tilde{O}(1/\epsilon^3)$ *rate, even though it uses a totally different separation oracle. It remains open whether this gap can be closed for efficient algorithms.*

**General case.** In the case of halfspaces without margin, we only have access to a separation oracle so we cannot use gradient-based methods. Since our algorithm already depends on the bit complexity of the input, we give an algorithm with a polynomial runtime guarantee in terms of the bit complexity (i.e. we do not have to assume access to exact real arithmetic). The ellipsoid algorithm [GLS12] is guaranteed to output a $\hat{\mathbf{w}} \in \mathbb{S}^{d-1}$ with misclassification error $err(\hat{\mathbf{w}}) \leq \eta + \epsilon$ using $poly(d, b)$ oracle queries and in time $\text{poly}(d, b, \frac{1}{\epsilon}, \frac{1}{\delta})$ with probability $1 - \delta$ provided two conditions hold:

1. First, the volume of the set of vectors $\mathbf{w}' \in \mathbb{S}^{d-1}$ achieving optimal prediction error (i.e. $\text{sgn}(\langle \mathbf{w}', \mathbf{X} \rangle) = \text{sgn}(\langle \mathbf{w}^*, \mathbf{X} \rangle)$ almost surely) is greater than $2^{-\text{poly}(b,d)}$

2. Second, for any $\mathbf{w}$ with misclassification error $err(\mathbf{w}) \geq \eta$ there exists a separation oracle $\mathcal{O}(\mathbf{w})$ that outputs a hyperplane $\mathbf{g} \in \mathbb{R}^d$ satisfying $\langle -\mathbf{g}, \mathbf{w}' - \mathbf{w} \rangle \geq 0$ for any $\mathbf{w}'$ achieving optimal prediction error. Furthermore $\mathcal{O}(\cdot)$ is computable in $\text{poly}(d, b, \frac{1}{\epsilon})$ time.

We use the following result to show that the set of vectors achieving optimal prediction error has non-negligible volume. The precise polynomial dependence can be found in [GLS12]; this is the full-dimensional case of the more general results used to show that linear programs are solvable in polynomial time.

**Lemma C.17** (Proposition 2.4 of [Coh97]). *Let* $\mathcal{D}$ *be any distribution supported on* $b$*-bit integer vectors in* $\mathbb{R}^d$*. There exists a compact, convex subset* $F$ *of the unit ball in* $\mathbb{R}^d$ *such that:*

1. $F$ *is the intersection of a convex cone, generated by* $d$ *vectors* $\mathbf{h}_1, \dots, \mathbf{h}_d$*, with the closed unit ball.*

2. $\text{vol}(F) \geq 2^{-\text{poly}(b,d)}$*.*

3. *For any* $w \in F \setminus \{0\}$*,* $\text{sgn}(\langle w, \mathbf{X} \rangle) = \text{sgn}(\langle w^*, \mathbf{X} \rangle)$ $\mathcal{D}$*-almost surely.*

**Theorem C.18.** *Suppose that* $(\mathbf{X}, Y)$ *is an* $\eta$*-Massart halfspace model and* $\mathbf{X}$ *is supported on vectors of bit-complexity at most* $b$*. There exists an algorithm (the ellipsoid method combined with* GENERALHALFSPACEORACLE*) which runs in time* $poly(d, b, 1/\epsilon, 1/\delta)$ *and sample complexity* $\tilde{O}\left(\frac{\text{poly}(d,b)}{\epsilon^3}\right)$ *which outputs* $\mathbf{w}$ *such that*

$$\text{err}(\mathbf{w}) \leq \eta + \epsilon$$

*with probability at least* $1 - \delta$*.*

*Proof.* Given the above stated guarantee for the ellipsoid method, Lemma C.17, and Theorem C.6, all we need to check is the sample complexity of the algorithm. To guarantee that a single call to GENERALHALFSPACEORACLE produces a separation oracle separating $F$ from the current iterate $\mathbf{w}$, we apply Theorem C.6 with $\mathbf{w}'$ equal to each of $0, \mathbf{h}_1, \dots, \mathbf{h}_d$ and apply the union bound. Since all of these vectors will be on the correct side of the separating hyperplane, the convex set $F$ wil be as well. Applying the union bound over all of the iterations of the algorithm gives the result. $\quad\square$

# D  Reduction from Proper to Improper Learning

In this section, we develop blackbox reductions from proper to improper learning under Massart noise. This means that we show, given query access to any classifier $h$ which achieves 0-1 errer upper bounded by $L$, how to generate a *proper* classifier $\mathbf{x} \mapsto \text{sgn}(\langle \mathbf{w}, \mathbf{x} \rangle)$ which achieves equally good 0-1 loss in polynomial time and using a polynomial number of samples from the model. The significance

of this is that it allows us to use a much more powerful class to initially learn the classifier, which may be easier to make robust to the Massart noise corruption, and still extract at the end a simple and interpretable halfspace predictor. Schemes along these lines, which use a more powerful learner to train a weaker one, are referred to as knowledge distillation or model compression procedures (see e.g. [HVD14]), and sometimes involve the use of "soft targets" (i.e. mildly non-blackbox access to the teacher). Our results show that the in the Massart setting, there is in fact a simple way to perform blackbox knowledge distillation which is extremely effective.

The key observation we make is that such a reduction is actually *implied* by the existence of an $\eta + \epsilon$ Massart proper learner based upon implementation of a gradient or separating hyperplane oracle, as developed in Section C. The reason is that if we view an improper hypothesis as a "teacher" and a halfspace $\text{sgn}(\langle \mathbf{w}, \mathbf{X} \rangle)$ parameterized by $\mathbf{w}$ in the unit ball as a "student", as long as the student is inferior to the teacher we can make progress by focusing on the region where the teacher and student disagree. On this region, the student has accuracy lower than 50%, so by appealing to the previously developed separation oracle for the proper learner, we can move $w$ towards $w^*$ (in the case of the gradient-based learner) or otherwise zoom in on $w^*$ (for the cutting-plane based learner).

In other words, in context of the $\mathbf{w}$ versus $S$ game introduced in the previous section, blackbox access to an improper learner massively increases the strength of the $S$ player. We make the preceeding arguments formal in the remainder of this section.

## D.1 Boosting Separation Oracles using a Good Teacher

Given blackbox access to a "teacher" hypothesis $h$ with good 01 loss, in order to produce an equally good proper learner we simply run one of our proper learning algorithms replacing the original separation oracle $\mathcal{O}(\mathbf{w})$ with the following "boosted" separation oracle BOOSTSEPARATIONORACLE$(h, \mathbf{w}, \mathcal{O})$. Remarkably, even if $\eta = 0.49$ and the original separation oracle can only separate $\mathbf{w}^*$ from $\mathbf{w}$ with 01 loss greater than 49%, if we have black-box query access to hypothesis $h$ achieving error rate 1%, the resulting boosted oracle can suddenly distinguish between $\mathbf{w}^*$ and all $\mathbf{w}$ achieving error greater than 1%.

---

**Algorithm 6:** BOOSTSEPARATIONORACLE$(h, \mathbf{w}, \mathcal{O})$

---

1 Let $\mathcal{R} = \{\mathbf{x} : h(\mathbf{x}) \neq \text{sgn}(\langle \mathbf{w}, \mathbf{X} \rangle)\}$.
2 Define $\mathcal{D}_\mathcal{R}$ to be $\mathcal{D}$ conditional on $\mathbf{X} \in \mathcal{R}$. Note that given sampling access to $\mathcal{D}$, distribution $\mathcal{D}_\mathcal{R}$ is efficiently sampleable using rejection sampling.
3 Return the output of $\mathcal{O}(\mathbf{w})$ on distribution $\mathcal{D}_\mathcal{R}$.

---

**Lemma D.1.** *Suppose that the joint distribution* $(\mathbf{X}, Y)$ *follows the $\eta$-Massart halfspace model,* $h : \mathbb{R}^d \to \{\pm 1\}$ *and* $\mathbf{w} \in \mathbb{R}^d$ *a vector such that*

$$\text{err}(h) - \text{err}(\mathbf{w}) \geq \epsilon$$

*for $\epsilon > 0$. Suppose $\mathcal{O}$ is an oracle which with probability $1 - \delta/2$, access to $m$ samples from any Massart distribution $\mathcal{D}'$, and input $\mathbf{w}'$ such that $\Pr_{\mathcal{D}'}[\text{sgn}(\langle \mathbf{w}', \mathbf{X} \rangle) = Y] < 1/2$ outputs a separating hyperplane $\mathbf{g}'$ with $\|\mathbf{g}'\| \leq 1$ such that $\langle \mathbf{w}^* - \mathbf{w}', -\mathbf{g}' \rangle \geq \gamma$ for $\gamma \geq 0$. Then* BOOSTSEPARATIONORACLE$(h, \mathbf{w}, \mathcal{O})$ *returns with probability at least $1 - \delta$ a vector $\mathbf{g}$ with $\|\mathbf{g}\| \leq 1$ such that*

$$\langle \mathbf{w}^* - \mathbf{w}, -\mathbf{g} \rangle \geq \gamma$$

*and has sample complexity $O\left(\frac{m + \log(2/\delta)}{\epsilon}\right)$.*

*Proof.* Observe that

$$\epsilon \leq \text{err}(h) - \text{err}(\mathbf{w}) \leq \Pr[\mathbf{X} \in \mathcal{R}]$$

since $h$ and $\text{sgn}(\langle \mathbf{w}, \mathbf{X} \rangle)$ agree perfectly outside of $\mathcal{R}$. Furthermore, restricted to $\mathcal{R}$ it must be the case that $\Pr[\text{sgn}(\langle \mathbf{w}, \mathbf{X} \rangle) = Y \mid \mathbf{X} \in \mathcal{R}] < 1/2$, otherwise it would be the case that $\text{err}(h) > \text{err}(w)$ which contradicts the assumption. Therefore, as long as sampling from $\mathcal{D}$ produces at least $m$ samples landing in $\mathcal{R}$ the result follows from the guarantee for $\mathcal{O}$; by Bernstein's inequality (Theorem B.8), $O(\frac{m + \log(2/\delta)}{\epsilon})$ samples will suffice with probability at least $1 - \delta/2$, so taking the union bound gives the result. $\square$

## D.2  Proper to Improper Reductions

By combining BOOSTSEPARATIONORACLE with any of our proper learners, we get an algorithm for converting any improper learner into a proper one. We state the formal results here: in all of these results we assume that $h$ is a hypothesis which we are given oracle access to.

---

**Algorithm 7:** FILTERTRONDISTILLER$(\epsilon, \eta, \delta, \lambda, T)$

---

1  Let $\mathbf{w}_1$ be an arbitrary vector in the unit ball.
2  Build a empirical distribution $\hat{\mathcal{H}}$ from $m = \Omega(\log(T/\delta)/\epsilon^2)$ samples (to use as a test set).
3  **for** $t = 1$ *to* $T$ **do**
4      **if** $\hat{\text{err}}(\mathbf{w}_t) < \eta + \epsilon/2$ **then**
5          Return $\mathbf{w}_t$.
6      **else**
7          Let $\beta_t = \frac{1}{\sqrt{t}}$.
8          Let $\mathbf{g}_t = $
        BOOSTSEPARATIONORACLE$(h, \mathbf{w}_t, $FINDDESCENTDIRECTION$(\cdot, \epsilon/6, \delta/2T, \lambda))$.
9          Let $\mathbf{w}_{t+1} = \frac{\mathbf{w}_t - \eta_t \mathbf{g}_t}{\max(1, \|\mathbf{w}_t - \eta_t \mathbf{g}_t\|)}$.

---

**Theorem D.2.** *Suppose that* $(\mathbf{X}, Y)$ *is an* $\eta$-*Massart halfspace model with margin* $\gamma$ *and* $\eta < 1/2$. *Suppose that* $\epsilon > 0$ *such that* $\eta + \epsilon/6 \leq 1/2$. *With probability at least* $1 - \delta$, *Algorithm* FILTERTRON *combined with* BOOSTSEPARATIONORACLE *and oracle hypothesis* $h$, *returns* $\mathbf{w}$ *such that*

$$\text{err}(\mathbf{w}) \leq \text{err}(h) + \epsilon$$

*when* $T = O\left(\frac{1}{\gamma^2 \epsilon^2}\right)$ *and* $\lambda = \eta + \epsilon/6$. *The algorithm runs in total sample complexity* $n = O\left(\frac{\log(2/\delta\gamma\epsilon)}{\gamma^4 \epsilon^6}\right)$ *and runtime* $O(nd + n \log n)$.

Using the cutting plane variant, as in Theorem C.15, gives a matching result with improved sample complexity $O\left(\frac{d \log(2/\delta\gamma)}{\gamma^2 \epsilon^4}\right)$. Finally, we state the result for halfspaces without a margin assumption.

**Theorem D.3.** *Suppose that* $(\mathbf{X}, Y)$ *is an* $\eta$-*Massart halfspace model and* $\mathbf{X}$ *is supported on vectors of bit-complexity at most* $b$. *There exists an algorithm (the ellipsoid method combined with* BOOSTSEPARATIONORACLE *applied to* GENERALHALFSPACEORACLE*) which runs in time* $poly(d, b, 1/\epsilon, 1/\delta)$ *and sample complexity* $\tilde{O}\left(\frac{\text{poly}(d,b)}{\epsilon^4}\right)$ *which outputs* $\mathbf{w}$ *such that*

$$\text{err}(\mathbf{w}) \leq \text{err}(h) + \epsilon$$

*with probability at least* $1 - \delta$.

## D.3  Applications of the Proper-to-Improper Reduction

An interesting feature of the proper to improper reduction for Massart noise is that it allows us to use our toolbox of very general learners (e.g. boosting, kernel regression, neural networks) to fit an initial classifier, while nevertheless allowing us to output a simple halfspace in the end. In this section, we describe a number of interesting applications of this idea.

To start, we give the first proper learner achieving error better than $\eta + \epsilon$ for Massart halfspaces over the hypercube, i.e. $\mathbf{X} \sim Uni(\{\pm 1\}^d)$; in fact, we show that error $\mathsf{OPT} + \epsilon$ is achievable for any fixed $\epsilon > 0$ in polynomial time. (This is a stronger guarantee except when $\eta$ is already quite small.) We start with the following improper learner for halfspaces in the *agnostic* model [KKMS08], which is based on $L_1$ linear regression over a large monomial basis:

**Theorem D.4** (Theorem 1 of [KKMS08]). *Suppose that* $(\mathbf{X}, Y)$ *are jointly distributed random variables, where the marginal law of* $\mathbf{X}$ *follows* $Uni(\{\pm 1\}^d)$ *and* $Y$ *is valued in* $\{\pm 1\}$. *Define*

$$\mathsf{OPT} = \min_{\mathbf{w}^* \neq 0} \Pr[\text{sgn}(\langle \mathbf{w}^*, \mathbf{X} \rangle) \neq Y].$$

*For any $\epsilon > 0$, there exists an algorithm with runtime and sample complexity* $\mathrm{poly}(d^{1/\epsilon^4}, \log(1/\delta))$ *which with probability at least $1 - \delta$, outputs a hypothesis $h$ such that*

$$\Pr[h(\mathbf{X}) \neq Y] \leq \textsf{OPT} + \epsilon.$$

No matching proper learner is known in the agnostic setting. Based on results in computational complexity theory [ABX08], there should not be a general reduction from proper to improper learning of halfspaces. This means that there is no apparent way to convert $h(\mathbf{X})$ into a halfspace with similar performance; it's also possible that $h$ performs significantly better than $\textsf{OPT}$, since it can fit patterns that a halfspace cannot.

If we make the additional assumption that $(\mathbf{X}, Y)$ follows the Massart halfspace model, it turns out that all of these issues go away. By combining Theorem D.4 with our Theorem D.3, we immediately obtain the following new result:

**Theorem D.5.** *Suppose that $(\mathbf{X}, Y) \sim \mathcal{D}$ follows an $\eta$-Massart halfspace model with true halfspace $\mathbf{w}^*$, and the marginal law of $\mathbf{X}$ is $Uni(\{\pm 1\}^d)$. For any $\epsilon > 0$, there exists an algorithm (given by combining the algorithms of Theorem D.4 and Theorem D.3) with runtime and sample complexity* $\mathrm{poly}(d^{1/\epsilon^4}, \log(1/\delta))$ *which with probability at least $1 - \delta$, outputs $\mathbf{w}$ such that*

$$\Pr[\mathrm{sgn}(\langle \mathbf{w}, \mathbf{X} \rangle) \neq Y] \leq \textsf{OPT} + \epsilon.$$

*(Recall that $\textsf{OPT} = \Pr[\mathrm{sgn}(\langle \mathbf{w}^*, \mathbf{X} \rangle) \neq Y]$.)*

As applications of the same distillation-based meta-algorithm, we note a few other interesting results for learning halfspaces in the Massart model which follow by combining an improper agnostic learner with our reduction:

1. Theorem D.5 remains true with the weaker assumption that $\mathbf{X}$ is drawn from an arbitrary permutation-invariant distribution over the hypercube [Wim10].

2. For learning Massart conjunctions (or disjunctions) over an arbitrary permutation-invariant distribution over the hypercube, there is an algorithm with runtime and sample complexity can be improved to $d^{O(\log(1/\epsilon))}$ by combining the improper learner of [FK15] with our reduction.

3. Suppose that $\mathbf{Z}$ is a $d$-dimensional random vector with independent coordinates. Suppose that for each $i$ from 1 to $d$, $\mathbf{Z}_i$ is valued in set $\mathcal{Z}_i$ and $|\mathcal{Z}_i| = O(\mathrm{poly}(d))$. A *linear threshold function* is a halfspace in the one-hot encoding $\mathbf{X}$ of $\mathbf{Z}$, i.e. a function of the form

$$\mathbf{z} \mapsto \mathrm{sgn}\left(\sum_{i=1}^{n} w_i(\mathbf{z}_i)\right)$$

   where each $w_i$ is an arbitrary function. Then combining our reduction with the result of [BOW10] shows that we can properly learn Massart linear threshold functions in runtime and sample complexity $\mathrm{poly}(d^{O(1/\epsilon^4)}, \log(1/\delta))$.

4. If we furthermore assume $|\mathcal{Z}_i| = O(1)$ for all $i$, then the same result as 3. holds for $\mathbf{Z}$ drawn from a mixture of $O(1)$ product measures [BOW10].

All of these algorithms can be implemented in the SQ framework. Interestingly, the result 2. above (for learning Massart conjunctions) matches the SQ lower bound over $Uni(\{\pm 1\}^d)$ observed in Theorem E.1, so this result is optimal up to constants in the exponent.

## E   Statistical Query Lower Bound for Getting $\textsf{OPT} + \epsilon$

In this section we make an intriguing connection between evolvability as defined by Valiant [Val09] and PAC learning under Massart noise. To the best of our knowledge, it appears that this connection has been overlooked in the literature. As our main application, we use an existing lower bound from the evolvability literature [Fel11] to deduce a super-polynomial statistical query lower bound for learning halfspaces under Massart noise to error $\textsf{OPT} + \epsilon$ for sub-constant $\epsilon$. This answers an open question from [DGT19].

Formally, we show the following:

**Theorem E.1.** *There is an absolute constant $0 < c < 1$ such that for any sufficiently large $k \in \mathcal{N}$, the following holds. Any SQ algorithm that can distribution-independently PAC learn any halfspace to error $\mathsf{OPT} + 2^{-k}/12$ in the presence of $2^{-ck}$ Massart noise must make $(n/8k)^{k/9}$ statistical queries of tolerance $(n/8k)^{-k/9}$.*

In Section E.1, we review basic notions of evolvability and statistical query learning. In Section E.2 we show that if a concept class is distribution-independently evolvable, then it can be distribution-independently PAC-learned to error $\mathsf{OPT} + \epsilon$. In Section E.3 we show a partial reverse implication, namely that certain kinds of lower bounds against evolvability of concept classes imply lower bounds even against PAC-learning those concepts under *specific distributions* to error $\mathsf{OPT} + \epsilon$ in the presence of Massart noise. In Section E.4, we describe one such lower bound against evolvability of halfspaces due to [Fel11] and then apply the preceding machinery to conclude Theorem E.1.

## E.1 Statistical Query Learning Preliminaries

**Definition E.2.** *Let $\tau > 0$. In the* statistical query model *[Kea98], the learner can make queries to a* statistical query oracle *for target concept $f : \mathbb{R}^n \to \{\pm 1\}$ with respect to distribution $\mathcal{D}_{\mathbf{x}}$ over $\mathbb{R}^n$ and with* tolerance $\tau$. *A query takes the form of a function $F : \mathbb{R}^n \times \{\pm 1\} \to [-1, 1]$, to which the oracle may respond with any real number $z$ for which $|\mathbb{E}_{\mathcal{D}_{\mathbf{x}}}[F(\mathbf{x}, f(\mathbf{x}))] - z| \le \tau$. We say that a concept class $\mathcal{C}$ is SQ learnable over a distribution $\mathcal{D}_{\mathbf{x}}$ if there exists an algorithm which, given $f \in \mathcal{C}$, PAC learns $f$ to error $\epsilon$ using $\mathrm{poly}(1/\epsilon, n)$ SQ queries with respect to $\mathcal{D}_{\mathbf{x}}$ with tolerance $\mathrm{poly}(\epsilon, 1/n)$.*

*A* correlational statistical query *(CSQ) is a statistical query $F$ of the form $F(\mathbf{x}, y) \triangleq G(\mathbf{x}) \cdot y$ for some function $G : \mathbb{R}^n \to [-1, 1]$. We say that a concept class $\mathcal{C}$ is CSQ learnable if it is SQ learnable with only correlational statistical queries.*

Valiant [Val09] introduced a notion of learnability by random mutations that he called *evolvability*. A formal definition of this notion would take us too far afield, so the following characterization due to [Fel08] will suffice:

**Theorem E.3** ([Fel08], Theorems 1.1 and 4.1). *A concept class is CSQ learnable over a class of distributions $\mathcal{D}_{\mathbf{x}}$ if and only if it is evolvable with Boolean loss over that class of distributions.*

Henceforth, we will use the terms *evolvable* and *CSQ learnable* interchangeably.

## E.2 Evolvability Implies Massart Learnability

We now show the following black-box reduction which may be of independent interest, though we emphasize that this result will not be needed in our proof of Theorem E.1.

**Theorem E.4.** *Let $\mathcal{C}$ be a concept class for which there exists a CSQ algorithm $\mathcal{A}$ which can learn any $f \in \mathcal{C}$ to error $\epsilon$ over any distribution $\mathcal{D}_{\mathbf{x}}$ running in time $T$ and using at most $N$ correlational statistical queries with tolerance $\tau$. Then for any $\eta \in [0, 1/2)$, given any distribution $\mathcal{D}$ arising from $f$ with $\eta$ Massart noise, $\mathrm{MASSARTLEARNFROMCSQS}(\mathcal{D}, \mathcal{A}, \epsilon, \delta)$ outputs $\mathbf{h}$ satisfying $\Pr_{\mathcal{D}}[\mathbf{h}(\mathbf{x}) \ne y] \le \mathsf{OPT} + 2\epsilon$ with probability at least $1 - \delta$. Furthermore it draws at most $M = N \cdot \mathrm{poly}(1/\epsilon, 1/(1 - 2\eta), 1/\tau) \cdot \log(N/\delta)$ samples and runs in time $T + M$.*

By Theorem E.3, one consequence of Theorem E.4 is that functions which are distribution-independently evolvable with Boolean loss are distribution-independently PAC learnable in the presence of Massart noise.

The main ingredient in the proof of Theorem E.4 is the following basic observation.

**Fact E.5.** *Take any distribution $\mathcal{D}_{\mathbf{x}}$ over $\mathbb{R}^n$ which has density $\mathcal{D}_{\mathbf{x}}(\cdot)$, any function $f : \mathbb{R}^n \to \{\pm 1\}$, and any correlational statistical query $F(\mathbf{x}, y) \triangleq G(\mathbf{x}) \cdot y$, and let $\mathcal{D}$ be the distribution arising from $f$ with $\eta$ Massart noise. If $\mathcal{D}'_{\mathbf{x}}$ is the distribution with density given by*

$$\mathcal{D}'_{\mathbf{x}}(\mathbf{x}) \triangleq \frac{1}{Z}(1 - 2\eta(\mathbf{x}))\mathcal{D}_{\mathbf{x}}(\mathbf{x}) \; \forall \, \mathbf{x} \in \mathbb{R}^n, \qquad Z \triangleq \int_{\mathbb{R}^n} (1 - 2\eta(\mathbf{x}))\mathcal{D}_{\mathbf{x}}(\mathbf{x})d\mathbf{x}, \qquad (9)$$

*and $\mathcal{D}'$ is the distribution over $(\mathbf{x}, y)$ where $\mathbf{x} \sim \mathcal{D}'_{\mathbf{x}}$ and $y = f(\mathbf{x})$, then*

$$\mathop{\mathbb{E}}_{(\mathbf{x}, y) \sim \mathcal{D}}[F(\mathbf{x}, y)] = Z \cdot \mathop{\mathbb{E}}_{(\mathbf{x}, y) \sim \mathcal{D}'}[F(\mathbf{x}, y)]. \qquad (10)$$

*Proof.* This simply follows from the fact that $\mathbb{E}_{(\mathbf{x},y)\sim\mathcal{D}}[F(\mathbf{x},y)] = \mathbb{E}_{\mathbf{x}\sim\mathcal{D}_{\mathbf{x}}}[f(\mathbf{x})G(\mathbf{x})(1-2\eta(\mathbf{x}))].$ $\square$

This suggests that given a CSQ algorithm for learning a concept $f$ over arbitrary distributions, one can learn $f$ over arbitrary distributions in the presence of Massart noise by simply brute-force searching for the correct normalization constant $Z$ in (9) and simulating CSQ access to $\mathcal{D}'$ using (10). Formally, this is given in MASSARTLEARNFROMCSQS.

---

**Algorithm 8:** MASSARTLEARNFROMCSQS$(\mathcal{D}, \mathcal{A}, \epsilon, \delta)$

**Input:** Sample access to distribution $\mathcal{D}$ arising from $f \in \mathcal{C}$ with $\eta$ Massart noise, algorithm $\mathcal{A}$
      which distribution-independently learns any concept in $\mathcal{C}$ to error $\epsilon$ using at most $N$
      CSQs with tolerance $\tau$, error parameter $\epsilon > 0$, failure probability $\delta$
**Output:** Hypothesis $\mathbf{h}$ for which $\Pr_{(\mathbf{x},y)\sim\mathcal{D}}[\mathbf{h}(\mathbf{x}) \neq y] \leq \mathsf{OPT} + 2\epsilon$

1 Set $\tau' \leftarrow \tau(1-2\eta)^2/2$.
2 **for** $\tilde{Z} \in [0, \tau', 2\tau', ..., 1]$ **do**
3     Simulate $\mathcal{A}$: answer every correlational statistical query $F(\mathbf{x},y) \triangleq G(\mathbf{x}) \cdot y$ that it makes
      with an empirical estimate of $\frac{1}{\tilde{Z}}\mathbb{E}_{(\mathbf{x},y)\sim\mathcal{D}}[F(\mathbf{x},y)]$ formed from $O(\frac{\log(N/\delta)}{(1-2\eta)^2\tau^2})$ samples.
4     Let $\mathbf{h}$ be the output of $\mathcal{A}$.
5     Empirically estimate $\Pr_{(\mathbf{x},y)\sim\mathcal{D}}[\mathbf{h}(\mathbf{x}) \neq y]$ using $O(\log(1/\delta\tau(1-2\eta)^2/\epsilon^2)$ samples. If it is
      at most $\mathsf{OPT} + 3\epsilon/2$, output $\mathbf{h}$.
6 Return FAIL.

---

*Proof.* Let $\mathcal{D}'_{\mathbf{x}}, \mathcal{D}', Z$ be as defined in Fact E.5, and let $\tau' = \tau(1-2\eta)^2/2$. First note that $Z$ defined in (9) is some quantity in $[1-2\eta, 1]$. Let $\tilde{Z}$ be any number for which $Z \leq \tilde{Z} \leq Z + \tau'$. Note that this implies that $|1/Z - 1/\tilde{Z}| \leq \tau'/Z^2 \leq \tau/2$. In particular, for any correlational statistical query $F(\mathbf{x},y)$, Fact E.5 implies that

$$\left| \frac{1}{\tilde{Z}}\mathbb{E}_{\mathcal{D}}[F(\mathbf{x},y)] - \mathbb{E}_{\mathcal{D}'}[F(\mathbf{x},y)] \right| \leq \left| \frac{1}{Z} - \frac{1}{\tilde{Z}} \right| \cdot |\mathbb{E}_{\mathcal{D}}[F(\mathbf{x},y)]| \leq \tau/2.$$

With $O\left(\frac{\log(2N/\delta)}{(1-2\eta)^2\tau^2}\right)$ samples, we can ensure that for this choice of $\tilde{Z}$ and this query $F$, the empirical estimate, call it $\widehat{\mathbb{E}}^{\mathcal{D}}[F(\mathbf{x},y)]$, formed in Step 3 satisfies

$$\frac{1}{\tilde{Z}}\left| \mathbb{E}_{\mathcal{D}}[F(\mathbf{x},y)] - \widehat{\mathbb{E}}^{\mathcal{D}}[F(\mathbf{x},y)] \right| \leq \tau/2,$$

with probability $1 - \delta/2N$. In particular, this implies that for this choice of $\tilde{Z}$ in the main loop of MASSARTLEARNFROMCSQS, all of our answers to the CSQs made by $\mathcal{A}$ are correct to within error $\tau$ with probability $1 - \delta/2$, in which case the simulation is correct and the hypothesis $\mathbf{h}$ output by $\mathcal{A}$ in that iteration of the loop satisfies $\Pr_{\mathcal{D}'_{\mathbf{x}}}[\mathbf{h}(\mathbf{x}) \neq f(\mathbf{x})] \leq \epsilon$.

So we have that

$$\epsilon \geq \Pr_{\mathcal{D}'_{\mathbf{x}}}[\mathbf{h}(\mathbf{x}) \neq f(\mathbf{x})] = \frac{1}{Z}\mathbb{E}_{\mathcal{D}_{\mathbf{x}}}[(1-2\eta(\mathbf{x})) \cdot \mathbb{1}[\mathbf{h}(\mathbf{x}) \neq f(\mathbf{x})]] \geq \mathbb{E}_{\mathcal{D}_{\mathbf{x}}}[(1-2\eta(\mathbf{x})) \cdot \mathbb{1}[\mathbf{h}(\mathbf{x}) \neq f(\mathbf{x})]],$$

or equivalently, $\Pr_{(\mathbf{x},y)\sim\mathcal{D}}[\mathbf{h}(\mathbf{x}) \neq y] \leq \mathsf{OPT} + \epsilon$. $O(\log(1/\delta\tau(1-2\eta)^2)/\epsilon^2)$ samples are sufficient to ensure that the empirical estimates in Step 5 are accurate in every iteration of the loop with probability $1 - \delta/2$, so we conclude that the hypothesis output by MASSARTLEARNFROMCSQS has error at most $\mathsf{OPT} + 2\epsilon$ with probability at least $1 - \delta$, as desired. $\square$

**Remark E.6.** *The reduction from Theorem E.4 can be applied to study distribution-dependent learning of Massart halfspaces. If $\mathcal{D}_{\mathbf{x}}$ is the distribution we want to learn Massart halfspaces over, it suffices to have a CSQ algorithm which succeeds over any distribution with a density over $\mathcal{D}_{\mathbf{x}}$ valued in $[1-2\eta, \frac{1}{1-2\eta}]$.*

*As an example application of this reduction, we can combine it with the results of [KVV10] to get a learning algorithm with error $O(\sqrt{\eta})$ for Massart conjunctions over the unit hypercube; this follows*

*from the fact that their analysis is robust to a small amount of misspecification (related to the notion of drift in their work). This is somewhat weaker than our proper learner which can achieve $\eta + \epsilon$ error in the same setting.*

### E.3 A Partial Converse to Theorem E.4

Next, we show a partial converse to Theorem E.4, namely that in some cases, correlational statistical query lower bounds can imply distribution-specific SQ hardness of learning to error $\mathsf{OPT} + \epsilon$ in the presence of Massart noise.

**Theorem E.7.** *Let $0 \leq \eta < 1/2$, and let $\mathcal{D}_{\mathbf{x}}^*$ be a known distribution over $\mathbb{R}^n$ with density $\mathcal{D}_{\mathbf{x}}^*(\cdot)$ and support $\mathcal{X}$. Let $\mathcal{C}$ be a concept class consisting of functions $\mathbb{R}^n \to \{\pm 1\}$, and let $\mathcal{F}$ be a family of distributions supported on $\mathcal{X}$ for which the following holds:*

1. *For every $\mathbf{x} \in \mathcal{X}$ and $\mathcal{D}_{\mathbf{x}} \in \mathcal{F}$, $1 - \eta \leq \frac{\mathcal{D}_{\mathbf{x}}(\mathbf{x})}{\mathcal{D}_{\mathbf{x}}^*(\mathbf{x})} \leq 1 + \eta$.*

2. *Any CSQ algorithm that can learn concepts in $\mathcal{C}$ to error $\epsilon$ over any distribution in $\mathcal{F}$ must make more than $N$ correlational statistical queries with tolerance $\tau$.*

*Then any SQ algorithm that can learn concepts in $\mathcal{C}$ over $\mathcal{D}_{\mathbf{x}}^*$ to error $\mathsf{OPT} + \epsilon$ in the presence of $\eta$ Massart noise requires more than $N$ statistical queries in with tolerance $\tau$.*

*Proof.* We will show the contrapositive. Let $\mathcal{A}$ be an SQ algorithm for learning $\mathcal{C}$ over $\mathcal{D}_{\mathbf{x}}^*$ to error $\mathsf{OPT} + \epsilon$ in the presence of $\eta$ Massart noise.

Take any $f \in \mathcal{C}$ and any $\mathcal{D}_{\mathbf{x}} \in \mathcal{F}$ and consider the following distribution $\mathcal{D}$ over $\mathcal{X} \times \{\pm 1\}$ arising from $f$ with $\eta$ Massart noise. For every $\mathbf{x} \in \mathcal{X}$, let $\eta(\mathbf{x})$ satisfy

$$1 - 2\eta(\mathbf{x}) = \frac{\mathcal{D}_{\mathbf{x}}(\mathbf{x})}{(1 + \eta)\mathcal{D}_{\mathbf{x}}^*(\mathbf{x})}. \tag{11}$$

By assumption, the right-hand side is at most 1 and at least $\frac{1-\eta}{1+\eta} \geq 1 - 2\eta$, so $0 \leq \eta(\mathbf{x}) \leq \eta$. Let $\mathcal{D}$ be the distribution over pairs $(\mathbf{x}, y)$ where $\mathbf{x} \sim \mathcal{D}_{\mathbf{x}}^*$, and $y = f(\mathbf{x})$ with probability $1 - \eta(\mathbf{x})$ and $y = -f(\mathbf{x})$ otherwise.

We will now show how to use $\mathcal{A}$ to produce a CSQ algorithm $\mathcal{A}'$ for learning any $f \in \mathcal{C}$ over distribution $\mathcal{D}_{\mathbf{x}}$. Suppose $\mathcal{A}$ makes $N$ statistical queries $F_1(\mathbf{x}, y), ..., F_N(\mathbf{x}, y)$ with tolerance $\tau$ and outputs $\mathbf{h} : \mathcal{X} \to \{\pm 1\}$ for which $\Pr_{\mathcal{D}}[\mathbf{h}(\mathbf{x}) \neq y] \leq \mathsf{OPT} + \epsilon$. Equivalently, we have that

$$\mathop{\mathbb{E}}_{\mathbf{x} \sim \mathcal{D}_{\mathbf{x}}^*} [(1 - 2\eta(\mathbf{x})) \cdot \mathbb{1}[f(\mathbf{x}) \neq \mathbf{h}(\mathbf{x})]] \leq \epsilon.$$

Note that this implies that

$$\mathop{\Pr}_{\mathbf{x} \sim \mathcal{D}_{\mathbf{x}}} [f(\mathbf{x}) \neq \mathbf{h}(\mathbf{x})] \leq (1 + \eta)\epsilon. \tag{12}$$

The CSQ algorithm $\mathcal{A}'$ proceeds as follows. Say $\mathcal{A}$ makes query $F_i(\mathbf{x}, y)$ with tolerance $\tau$. If $F_i$ only depends on $\mathbf{x}$, then because $\mathcal{D}_{\mathbf{x}}^*$ is a known distribution, this is just an average over $\mathbb{R}^n \times \{1\}$ that $\mathcal{A}'$ can compute on its own, without making any correlational statistical queries. If $F_i$ depends on both $\mathbf{x}$ and $y$, it can be decomposed as $F_i(\mathbf{x}, y) = G_i(\mathbf{x}) \cdot y + G_i'(\mathbf{x})$ for some functions $G_i, G_i'$. Again, $\mathcal{A}'$ can compute $\mathbb{E}_{\mathcal{D}_{\mathbf{x}}}[G_i'(\mathbf{x})]$ on its own without making any correlational statistical queries, and it can make a query $\frac{1}{1+\eta}G_i(\mathbf{x}) \cdot y$ to the CSQ oracle over $\mathcal{D}_{\mathbf{x}}$ with tolerance $\tau$ to get a $\tau$-close estimate of

$$\frac{1}{1 + \eta} \mathop{\mathbb{E}}_{\mathcal{D}_{\mathbf{x}}} [G_i(\mathbf{x})f(\mathbf{x})] = \mathop{\mathbb{E}}_{\mathcal{D}_{\mathbf{x}}^*} [G_i(\mathbf{x})f(\mathbf{x})]$$

By the guarantees on $\mathcal{A}$ and by (12), we know $\mathcal{A}'$ makes at most $N$ correlational statistical queries and outputs a hypothesis $\mathbf{h}$ with error $(1 + \eta)\epsilon$ relative to $f$, with respect to the distribution $\mathcal{D}_{\mathbf{x}}$. $\square$

**Remark E.8.** *By definition $\mathsf{OPT}$ in Theorem E.7 is given by $\mathsf{OPT} = \mathbb{E}_{\mathcal{D}_{\mathbf{x}}^*}[\eta(\mathbf{x})]$, so by (11) we conclude that $1 - 2\mathsf{OPT} = \frac{1}{1+\eta}$, so in particular, $\mathsf{OPT} = \frac{\eta}{2(1+\eta)}$.*

**Remark E.9.** *The following converse to Remark E.6 holds as a consequence of Theorem E.7. Namely, known results for learning Massart halfspaces to error $\mathsf{OPT} + \epsilon$ over a known base distribution can be implemented in SQ and therefore imply CSQ algorithms for tiltings of the base distribution in the absence of noise. Equivalently, by Theorem E.3, this yields evolutionary algorithms with some natural resistance to "drift" [KVV10].*

*This is the case for the algorithm of [ZLC17] for learning Massart halfspaces under the uniform measure on the sphere, which runs a Langevin gradient method directly on the (smoothed) zero-one loss; the zero-one loss is easily estimated by a CSQ query. In fact, since their algorithm is based on searching for local improvements to the zero-one loss, it can be more directly fit into the evolvability framework without using the reduction of [Fel08], and can be viewed as a natural improvement to the drift-resistant halfspace evolvability results over the sphere established in [KVV10].*

## E.4  Instantiating Theorem E.7 for Halfspaces

In this section we show that for $\mathcal{C}$ the class of conjunctions of $k$ variables, there exists a family of distributions $\mathcal{F}$ satisfying the hypotheses of Theorem E.7 for $\mathcal{D}_{\mathbf{x}}^*$ the uniform distribution over $\{0,1\}^n \times \{1\}$.

The construction is based on [Fel11] which showed a super-polynomial lower bound against distribution-independently CSQ learning conjunctions, so we follow their notation closely. Fix parameter $k \in \mathcal{N}$. Let $U$ denote the uniform distribution over $\{0,1\}^n$. For every $S \subset [n]$ of size $k$, define the function $\theta_S$ as follows. First, let $t_S : \{0,1\}^n \to \{\pm 1\}$ denote the conjunction on the bits indexed by $S$. This has Fourier expansion $t_S(\mathbf{x}) = -1 + 2^{-k+1} \sum_{I \subseteq S} \chi_I(\mathbf{x})$, where $\chi_I(\mathbf{x}) \triangleq \prod_{i \in I} \mathbf{x}_i$. Define the function $\phi_S$ by zeroing out all Fourier coefficients of $t_S$ of size $1, ..., k/3$:

$$\phi_S(\mathbf{x}) = -1 + 2^{-k+1} + 2^{-k+1} \sum_{I \subseteq S : |I| > k/3} \chi_I(\mathbf{x}).$$

Define $Z \triangleq \sum_{x \in \{0,1\}^n} |\phi_S(\mathbf{x})|$ and let $\mathcal{D}_{\mathbf{x}}^S$ be the distribution given by $\mathcal{D}_{\mathbf{x}}^S(\mathbf{x}) = |\phi_S(\mathbf{x})|/Z$. Define the "tilted" conjunction $\theta_S(\mathbf{x}) = 2^n \cdot \phi_S(\mathbf{x})/Z$.

**Fact E.10.** *There is an absolute constant $0 < c < 1$ such that for every $\mathbf{x} \in [n]$, $|t_S(\mathbf{x}) - \phi_S(\mathbf{x})| \le 2^{-ck}$. In particular, for $k$ sufficiently large, $t_S(\mathbf{x}) = \mathrm{sgn}(\phi_S(\mathbf{x}))$ for all $\mathbf{x} \in \{0,1\}^n$, and therefore $\mathcal{D}_{\mathbf{x}}^S(\mathbf{x}) \cdot t_S(\mathbf{x}) = U(\mathbf{x}) \cdot \theta_S(\mathbf{x})$.*

*Proof.*

$$2^{-k+1} \sum_{I \subset S : |I| \in [k/3]} \chi_I(\mathbf{x}) \le 2^{-k+1} \sum_{j \in [k/3]} \binom{k}{j} \le 2 \cdot \Pr[\mathrm{Bin}(k, 1/2) \le k/3] \le 2^{-ck}$$

for some absolute constant $0 < c < 1$, from which the the first part of the claim follows. The latter parts follow immediately by triangle inequality. $\square$

Fact E.10 implies that the likelihood ratio between $\mathcal{D}_{\mathbf{x}}^S$ and $U$ is bounded:

**Fact E.11.** *There is an absolute constant $0 < c < 1$ such that for every $\mathbf{x}$, $|\mathcal{D}_{\mathbf{x}}^S(\mathbf{x})/U(\mathbf{x}) - 1| \le 2^{-ck}$.*

*Proof.* By definition $\mathcal{D}_{\mathbf{x}}^S(\mathbf{x})/U(\mathbf{x}) = |\phi_S(\mathbf{x})|/\mathbb{E}_x[|\phi_S(\mathbf{x})|]$. By Fact E.10, $\phi_S(\mathbf{x}) \in [1 - 2^{-ck}, 1 + 2^{-ck}]$, from which the claim follows. $\square$

The following is implicit in [Fel11]:

**Theorem E.12** ([Fel11], Theorem 8 and Remark 9). *Let $\mathcal{C}$ be the class of all conjunctions $t_S$ for $S \subset [n]$ of size $k$. Let $\mathcal{F}$ be the family of distributions $\{\mathcal{D}_{\mathbf{x}}^S\}_{S \subseteq [n]:|S|=k}$ defined above. Any CSQ algorithm for learning concepts in $\mathcal{C}$ over any distribution from $\mathcal{F}$ needs at least $(n/8k)^{k/9}$ queries with tolerance $(n/8k)^{-k/9}$.*

For completeness, we give a self-contained proof below.

*Proof.* Let $\alpha \triangleq 1/\mathbb{E}_U[|\phi_S(\mathbf{x})|]$, and define $\psi(\mathbf{x}) \triangleq \alpha(2^{-1+k} - 1)$. Let $A$ be any correlational SQ algorithm for the problem in the lemma statement. Simulate $A$ and let $F_1(\mathbf{x}), ..., F_q(\mathbf{x})$ be the sequence of correlational queries it makes given the oracle responds with $\langle F_i, \psi \rangle_U \triangleq \mathbb{E}_U[F_i(\mathbf{x}) \cdot \psi(\mathbf{x})]$ for each $i \in [q]$. Let $h_\psi$ be the hypothesis the algorithm outputs at the end.

For any $S \subseteq [n]$ of size $k$, at least one of the following must happen:

1. There exists $i \in [q]$ for which $|\langle F_i, \psi \rangle_U - \langle F_i, t_S \rangle_{\mathcal{D}^S_\mathbf{x}}| \geq \tau$.

2. $\Pr_{\mathcal{D}^S_\mathbf{x}}[h_\psi(\mathbf{x}) \neq t_S(\mathbf{x})] \leq \epsilon$ for $\epsilon \triangleq 2^{-k}/6$.

Indeed, if (1) does not hold, then from the perspective of the algorithm in the simulation, the responses of the CSQ oracle are consistent with the underlying distribution being $\mathcal{D}^S_\mathbf{x}$ and the underlying concept being $t_S$, so it will correctly output a hypothesis $h_\psi(\mathbf{x})$ which is $\epsilon$-close to $t_S$, so (2) will hold.

We now devise a large packing over subsets $S$ of size $k$ for which (2) holds for at most one subset, and for which the number of subsets of the packing for which (1) can hold is upper bounded by $O(q/\tau^2)$ in terms of $q$, yielding the desired lower bound on the number of queries any CSQ algorithm must make to solve the learning problem.

The packing is simply the maximal family $\mathcal{F}$ of subsets $S \subset [n]$ of size $k$ whose pairwise intersections are of size at most $k/3$. By a greedy construction, such a family will have size at least $2^{-k}(n/k)^{k/3} = (n/8k)^{k/3}$.

To show that (2) holds for exactly one subset in $\mathcal{F}$, suppose there are two such subsets $S, T$. We have that
$$\Pr_U[h_\psi(\mathbf{x}) \neq t_S(\mathbf{x})] \leq (1 + 2^{-ck}) \Pr_{\mathcal{D}^S_\mathbf{x}}[h_\psi(\mathbf{x}) \neq t_S(\mathbf{x})] \leq 2\epsilon \leq 2^{-k}/3,$$
where the first step follows by Fact E.11. Similarly, $\Pr_U[h_\psi(\mathbf{x}) \neq t_T(\mathbf{x})] \leq 2^{-k}/3$. But $S \neq T$, so $\Pr_U[t_S(\mathbf{x}) \neq t_T(\mathbf{x})] \geq 2^{-k}$, a contradiction.

Finally, we upper bound the number of subsets of the packing for which (1) can hold. Recalling that $\mathcal{D}^S_\mathbf{x}(\mathbf{x})t_S(\mathbf{x}) = U(\mathbf{x})\theta_S(\mathbf{x})$ for every $\mathbf{x}$, (1) is equivalent to the condition that there exists some $i \in [q]$ for which $|\langle F_i, \theta_S - \psi \rangle_U| \geq \tau$. But by construction
$$\psi(\mathbf{x}) - \theta_S(\mathbf{x}) = \frac{2^{-k+1}}{\mathbb{E}_U[|\phi_S(\mathbf{x})|]} \sum_{I \subseteq S: |I| > k/3} \chi_I(\mathbf{x}),$$
from which it follows that
$$\tau \leq \frac{2^{-k+1}}{\mathbb{E}_U[|\phi_S(\mathbf{x})|]} \sum_{I \subseteq S: |I| > k/3} |\langle \chi_I, F_i \rangle_U|.$$

By averaging and recalling that $\mathbb{E}_U[|\phi_S(\mathbf{x})|] \in [1 - 2^{-ck}, 1 + 2^{-ck}]$, we conclude that there exists some Fourier coefficient $I_S \subseteq S$ of size greater than $k/3$ for which $|\langle \chi_{I_S}, F_i \rangle_U| \geq \tau/4$. Furthermore, for $S \neq T$ in $\mathcal{F}$ for which (1) holds, clearly $I_S \neq I_T$ because $I_S \subseteq S, I_T \subseteq T$ are of size greater than $|S \cap T|$. On the other hand, for any $i \in [q]$, there are at most $16/\tau^2$ sets $I$ for which $|\langle \chi_I, F_i \rangle_U| \geq \tau/4$, as $\|F_i\|_2^2 \leq 1$. We conclude that there are most $16q/\tau^2$ sets $S \in \mathcal{F}$ for which (1) holds. The proof of the theorem upon taking $\tau = (n/8k)^{-k/9}$. $\qquad\square$

We can now apply Theorem E.12 and Theorem E.7 to prove the main result of this section:

*Proof of Theorem E.1.* Let $\mathcal{D}^*_\mathbf{x}$ be the uniform distribution over $\{0,1\}^n \times \{1\}$. We show the following stronger claim, namely that the theorem holds even if we restrict to instances where $\mathbf{x} \sim \mathcal{D}^*_\mathbf{x}$, and the unknown halfspace $\mathbf{w} \in \mathbb{R}^{n+1}$ is promised to consist, in the first $n$ coordinates, of some bitstring of Hamming weight $k$ (corresponding to a conjunction of size $k$) and, in the last coordinate, the number $k$. By design, if $f$ is the conjunction corresponding to $\mathbf{w}$, then $f(\mathbf{x}) = \text{sgn}(\mathbf{w}, (\mathbf{x}, 1))$ for all $\mathbf{x} \in \{0,1\}^n$.

Let $\mathcal{F}$ be as defined in Theorem E.12. By Theorem E.7 and Theorem E.12, it is enough to show that for any $\mathcal{D}^S_\mathbf{x} \in \mathcal{F}$ and any $\mathbf{x} \in \{0,1\}^n \times \{1\}$, $1 - \eta \leq \frac{\mathcal{D}^S_\mathbf{x}(\mathbf{x})}{\mathcal{D}^*_\mathbf{x}} \leq 1 + \eta$ for $\eta = 2^{-ck}$ for some $0 < c < 1$. But this was already shown in Fact E.11, completing the proof of the Theorem. $\qquad\square$

Note that by Remark E.8, OPT is within a factor of 2 of $\eta$. On the one hand, this implies that our Theorem E.1 is not strong enough to rule out efficient SQ algorithms for achieving $O(\mathsf{OPT}) + \epsilon$ for Massart halfspaces. On the other hand, one can also interpret Theorem E.1 as saying that improving the accuracy guarantees of [DGT19] by even a constant factor will require super-polynomial statistical query complexity.

## F Improperly Learning Misspecified GLMs

In this section we will prove our main result about GLMs:

**Theorem F.1.** *Fix any $\epsilon > 0, \delta > 0$. Let $\mathcal{D}$ be a distribution arising from an $\zeta$-misspecified GLM with odd link function $\sigma(\cdot)$. With probability at least $1 - \delta$, algorithm* LEARNMISSPECGLM*($\mathcal{D}, \epsilon, \delta$) outputs an improper classifier $\mathbf{h}$ for which*

$$\operatorname*{err}_{\mathcal{X}}(\mathbf{h}) \leq \frac{1 - \mathbb{E}_{\mathcal{D}}[\sigma(|\langle \mathbf{w}^*, \mathbf{X}\rangle|)]}{2} + \zeta + O(\epsilon).$$

*Moreover, the algorithm has sample complexity $N = \operatorname{poly}(L, \epsilon^{-1}, (\zeta \vee \epsilon)^{-1}) \cdot \log(1/\delta)$ and runs in time $d \cdot \operatorname{poly}(N)$, where $d$ is the ambient dimension.*[6]

Our lemmas will reference subsets $\mathcal{X}' \subseteq \mathcal{X}$; when the definition of $\mathcal{X}'$ is clear, we will use $\mathcal{D}'$ and $\mathcal{D}'_{\mathbf{x}}$ to denote the distributions $\mathcal{D}$ and $\mathcal{D}_{\mathbf{x}}$ conditioned on $\mathbf{X} \in \mathcal{X}'$.

For any such $\mathcal{X}'$, define

$$\Delta(\mathcal{X}') \triangleq \mathbb{E}[\sigma(|\langle \mathbf{w}^*, \mathbf{X}\rangle|) \mid \mathbf{X} \in \mathcal{X}'] - 2\zeta.$$

This should be interpreted as the accuracy of the optimal classifier on the clean data, minus a conservative upper bound on the loss in performance when $\zeta$-misspecification is introduced; our goal is to produce a classifier with accuracy at least $\Delta(\mathcal{X}) - O(\epsilon)$. Also define the "optimal leakage parameter"

$$\lambda(\mathcal{X}') \triangleq \frac{1 - \Delta(\mathcal{X}')}{2}. \tag{13}$$

Equivalently, our goal is to produce a classifier whose misclassification error is at most $\lambda(\mathcal{X}) + O(\epsilon)$. Note that in the special case of halfspaces under $\eta$-Massart noise, which can be viewed as a 0-misspecified GLM with link function $\sigma(z) = (1 - 2\eta)\operatorname{sgn}(z)$, $\Delta(\mathcal{X}')$ would be equal to $1 - 2\eta$, and $\lambda(\mathcal{X}')$ would be $\eta$, i.e. the ideal choice of leakage parameter from [DGT19], regardless of the choice of $\mathcal{X}'$.

We give an improper, piecewise-constant classifier $\mathbf{h}(\cdot)$ which is specified by a partition $\mathcal{X} = \sqcup_i \mathcal{X}^{(i)}$ and a choice of sign $s^{(i)} \in \{\pm 1\}$ for each region of the partition, corresponding to the classifier's label for all points in that region. As our goal is for $\operatorname{err}_{\mathcal{X}}(\mathbf{h}) \leq \lambda(\mathcal{X}) + O(\epsilon)$, it will suffice for every index $i$ of the partition of our improper classifier to satisfy $\operatorname{err}_{\mathcal{X}^{(i)}}(s^{(i)}) \leq \lambda(\mathcal{X}^{(i)}) + O(\epsilon)$.

At a high level, our algorithm starts with a trivial partition and labeling of $\mathcal{X}$ and iteratively refines it. At any point, each region $\mathcal{X}^{(i)}$ of the current partition is in one of two possible states:

- *Live*: the algorithm may continue refining the classifier over points in $\mathcal{X}^{(i)}$ in future iterations.
- *Frozen*: the current constant classifier $s^{(i)}$ on $\mathcal{X}^{(i)}$ is sufficiently competitive that we do not need to update it in future iterations.

To update a given partition and labeling, the algorithm takes one of two actions at every step as long as the mass of the live regions is non-negligible:

a) *Splitting*: for the largest live region $\mathcal{X}^{(i)}$ of the partition, try to break off a non-negligible subregion $\tilde{\mathcal{X}} \subset \mathcal{X}^{(i)}$ for which we can update the label we assign to points in $\tilde{\mathcal{X}}$ and make progress. We will show that if this is impossible, then this *certifies* that the performance of the current constant classifier $s^{(i)}$ on $\mathcal{X}^{(i)}$ is already sufficiently close to $\Delta(\mathcal{X}^{(i)})$ that we can mark $\mathcal{X}^{(i)}$ as frozen.

b) *Merging*: after a splitting step, try to merge any two regions which the current classifier assigns the same label and on which it has comparable performance. This ensures that the number of live regions in the partition is bounded and, in particular, that as long as the mass of the live regions is non-negligible, the mass of the region chosen in the next splitting step is non-negligible.

Formally our algorithm, LEARNMISSPECGLM, is given in Algorithm 9 below.

---

**Algorithm 9:** LEARNMISSPECGLM$(\mathcal{D}, \epsilon, \delta)$

---

**Input:** Sample access to $\mathcal{D}$, error parameter $\epsilon$, failure probability $\delta$
**Output:** $O(\zeta)$-competitive improper classifier $\mathbf{h}$ given by a partition of $\mathcal{X}$ into regions $\{\mathcal{X}^{(i)}\}$
       with labels $\{s^{(i)}\}$

1   $\delta' \leftarrow O(\delta\epsilon^6/L),\ \upsilon \leftarrow O(\epsilon^{3/2})$.
2   Draw $O(\log(1/\delta')/\epsilon^2)$ samples from $\mathcal{D}$ and pick $s \in \{\pm 1\}$ minimizing $\widehat{\mathrm{err}}_{\mathcal{X}}(s)$.
3   Initialize with trivial classifier $\mathbf{h} = (\{\mathcal{X}\}\{s\})$ and set live region to be $\mathcal{X}_{\text{live}} = \mathcal{X}$.
4   **while** *True* **do**
5      Draw $O(\log(1/\delta')/\epsilon^2)$ samples and form estimate $\widehat{\mu}(\mathcal{X}_{\text{live}})$.
6      **if** $\widehat{\mu}(\mathcal{X}_{live}) \leq \epsilon$ **then**
7         **return h**
8      Draw $O(\log(1/\delta'\upsilon)/\epsilon\upsilon^2)$ samples from $\mathcal{D}$ and form estimate $\widehat{\mu}(\mathcal{X}^{(i)} \mid \mathcal{X}_{\text{live}})$ for every
        region $\mathcal{X}^{(i)}$ in current classifer.
9      $\mathcal{X}' \leftarrow \mathrm{argmax}_{\mathcal{X}^{(i)}}\ \widehat{\mu}(\mathcal{X}^{(i)} \mid \mathcal{X}_{\text{live}})$. Let $s_0$ be its label under $\mathbf{h}$.
10     **if** SPLIT$(\mathcal{X}', s_0, \epsilon, \delta')$ outputs $\tilde{\mathcal{X}}, \tilde{s}, s$ **then**
11        Update $\mathbf{h}$ by splitting $\mathcal{X}$ into $\tilde{\mathcal{X}}$ and $\mathcal{X}\backslash\tilde{\mathcal{X}}$ and assigning these regions labels $\tilde{s}$ and $s$
         respectively.
12        **if** MERGE$(\mathbf{h}, \upsilon, \delta')$ outputs a hypothesis $\mathbf{h}'$ **then**
13          $\mathbf{h} \leftarrow \mathbf{h}'$.
14     **else**
15        Freeze region $\mathcal{X}'$ with the label $s$ output by SPLIT. $\mathcal{X}_{\text{live}} \leftarrow \mathcal{X}_{\text{live}}\backslash\mathcal{X}'$.

---

**Remark F.2.** *To sample from a particular region of the partition in a given iteration of the main loop of* LEARNMISSPECGLMS, *we will need an efficient membership oracle in order to do rejection sampling. Fortunately, the function sending any $\mathbf{x} \in \mathcal{X}$ to the index of the region in the current partition to which it belongs has a compact representation as a circuit given by wiring together the outputs of certain linear threshold functions. To see this inductively, suppose that some region $\mathcal{X}'$ from the previous time step gets split into $\tilde{\mathcal{X}} = \mathcal{X}' \cap \mathcal{A}(\mathbf{w}, \tau)$ and $\mathcal{X}'\backslash\tilde{\mathcal{X}}$. Then if $\mathcal{C}$ is the circuit computing the previous partition, then the new partition after this split can be computed by taking the gate in $\mathcal{C}$ which originally output the index of region $\mathcal{X}'$ and replacing it with a gate corresponding to membership in $\mathcal{A}(\mathbf{w}, \tau)$, which is simply an AND of two (affine) linear threshold functions. Similarly, suppose some regions $\mathcal{X}_1, ..., \mathcal{X}_m$ from the previous time step get merged to form a region $\mathcal{X}'$. Then $\mathcal{C}$ gets updated by connecting the output gates for those regions with an additional OR gate.*

*In particular, because the main loop of* LEARNMISSPECGLMS *terminates after polynomially many iterations, and the size of the circuit increases by an additive constant with every iteration, the circuits that arise in this way can always be evaluated efficiently.*

### F.1 Splitting Step

The pseudocode for the splitting procedure, SPLIT, is given in Algorithm 10. At a high level, when SPLIT is invoked on a region $\mathcal{X}'$ of the current partition, for every possible choice of leakage parameter $\lambda$, it runs SGD on $L_\lambda^{\mathcal{X}'}$ and, similar to [DGT19], finds a non-negligible annulus over which the conditional misclassification error is minimized. It picks the $\lambda$ for which this conditional error is smallest and splits off the half of the annulus with smaller conditional error[7] as long as the error of

**Algorithm 10:** SPLIT$(\mathcal{X}', s_0, \epsilon, \delta)$

---

**Input:** Region $\mathcal{X}' \subseteq \mathcal{X}$ with label $s_0 \in \{\pm 1\}$, error parameter $\epsilon > 0$, failure probability $\delta > 0$

**Output:** Region $\tilde{\mathcal{X}} \subset \mathcal{X}'$ and labels $s, \tilde{s} \in \{\pm 1\}$ for $\mathcal{X}'\backslash\mathcal{X}$ and $\tilde{\mathcal{X}}$ respectively; or FREEZE and label $s \in \{\pm 1\}$ (see Lemma F.3)

**1** $\delta' \leftarrow O(\delta\epsilon)$.

**2** $\zeta' \leftarrow 2\zeta + \epsilon$.

**3** Draw $O(\log(1/\delta')/\epsilon^2)$ samples from $\mathcal{D}'$ and if $\widehat{\text{err}}_{\mathcal{X}'}(s_0) > 1/2$, $s \leftarrow -s_0$. Otherwise, $s \leftarrow s_0$. Use $\widehat{\text{err}}_{\mathcal{X}'}(s_0)$ to define $\widehat{\text{err}}_{\mathcal{X}'}(s)$ accordingly.

**4** Draw $O\left(\frac{L^2}{\epsilon^3 \zeta'^2}\right)$ fresh samples from $\mathcal{D}'$ to form an empirical distribution $\hat{\mathcal{D}}'$.

**5 for** $\lambda \in \{0, \epsilon/4, 2\epsilon/4, 3\epsilon/4, ..., 1/2\}$ **do**

**6** $\quad$ Run SGD on $L_\lambda^{\mathcal{X}'}(\cdot)$ for $O(\frac{L^2}{\epsilon^2 \zeta'^2} \cdot \log(1/\delta'))$ iterations to get $\mathbf{w}_\lambda$ for which $\|\mathbf{w}_\lambda\| = 1$ and
$\quad L_\lambda^{\mathcal{X}'}(\mathbf{w}_\lambda) \leq \min_{\mathbf{w}':\|\mathbf{w}'\| \leq 1} L_\lambda^{\mathcal{X}'}(\mathbf{w}') + \frac{\epsilon\zeta'}{4L}$.

**7** $\quad$ Find a threshold $\tau_\lambda$ such that $\widehat{\mu}(\mathcal{X}' \cap \mathcal{A}(\mathbf{w}_\lambda, \tau_\lambda) \mid \mathcal{X}') \geq \frac{\epsilon\zeta'}{9L\lambda}$ and misclassification error
$\quad \widehat{\text{err}}_{\mathcal{X}' \cap \mathcal{A}(\mathbf{w}_\lambda, \tau_\lambda)}(\mathbf{w}_\lambda)$ is minimized, where the empirical estimates $\widehat{\mu}$ and $\widehat{\text{err}}$ are with respect to $\hat{\mathcal{D}}'$.

**8** Take $\lambda$ for which empirical misclassification error $\widehat{\text{err}}_{\mathcal{X}' \cap \mathcal{A}(\mathbf{w}_\lambda, \tau_\lambda)}(\mathbf{w}_\lambda)$ is minimized.

**9** For each $s' \in \{\pm 1\}$, $\tilde{\mathcal{X}}_{s'} \leftarrow \mathcal{X}' \cap \mathcal{H}^{s'}(\mathbf{w}_\lambda, \tau_\lambda)$.

**10** Draw $O(\frac{L\lambda}{\epsilon^4 \zeta'} \cdot \log(1/\delta'))$ samples from $\mathcal{D}'$ and, for each $s' \in \{\pm 1\}$, form empirical estimates $\widehat{\mu}(\tilde{\mathcal{X}}_{s'} \mid \mathcal{X}')$ and $\widehat{\text{err}}_{\tilde{\mathcal{X}}_{s'}}(s')$.

**11 if** $\min_{s'} \widehat{\mu}(\tilde{\mathcal{X}}_{s'} \mid \mathcal{X}') \geq \Omega(\frac{\epsilon^2 \zeta'}{L\lambda})$ **then**

**12** $\quad \tilde{s} \leftarrow \text{argmin}_{s'} \widehat{\text{err}}_{\tilde{\mathcal{X}}_{s'}}(s')$.

**13 else**

**14** $\quad \tilde{s} \leftarrow \text{argmax}_{s'} \widehat{\mu}(\tilde{\mathcal{X}}_{s'} \mid \mathcal{X}')$

**15** $\tilde{\mathcal{X}} \leftarrow \tilde{\mathcal{X}}_{\tilde{s}}$.

**16 if** $\widehat{\text{err}}_{\tilde{\mathcal{X}}}(\tilde{s}) \leq \widehat{\text{err}}_{\mathcal{X}'}(s) - \epsilon$ **then**

**17** $\quad$ **return** $\tilde{\mathcal{X}}$ *and labels* $\tilde{s}, s$

**18 else**

**19** $\quad$ **return** FREEZE *and label* $s$

---

**Algorithm 11:** MERGE$(\mathbf{h}, \upsilon, \delta)$

---

**Input:** Improper classifier $\mathbf{h}$ given by a partition $\mathcal{X} = \sqcup_i \mathcal{X}^{(i)}$ and labels $s^{(i)} \in \{\pm 1\}$, coarseness parameter $\upsilon > 0$, failure probability $\delta > 0$

**Output:** Either an improper classifier $\mathbf{h}'$ given by partition $\mathcal{X} = \sqcup_i \mathcal{X}'^{(i)}$ and labels $s'^{(i)} \in \{\pm 1\}$, or None (see Lemma F.8)

**1** Let $M$ be the size of the partition $\sqcup_i \mathcal{X}^{(i)}$.

**2** Draw $O(\log(M/\delta)/\upsilon^2)$ samples; for every region $\mathcal{X}^{(i)}$, form empirical estimate $\widehat{\text{err}}_{\mathcal{X}^{(i)}}(s^{(i)})$.

**3 if** *exist indices* $i \neq j$ *for which* $|\widehat{\text{err}}_{\mathcal{X}^{(i)}}(s^{(i)}) - \widehat{\text{err}}_{\mathcal{X}^{(j)}}(s^{(j)})| \geq \upsilon$ **then**

**4** $\quad$ **return** *partition given by merging* $\mathcal{X}^{(i)}$ *and* $\mathcal{X}^{(j)}$ *and retaining the other regions in* $\{\mathcal{X}^{(i)}\}$, *and labels defined in the obvious way*

**5 else**

**6** $\quad$ **return** None

---

that half is non-negligibly better than the error of the previous constant classifier $s$ over all of $\mathcal{X}'$. If this is not the case, $\mathcal{X}'$ gets frozen.

Below, we prove the following main guarantee about SPLIT, namely that if it does not make progress by splitting off a non-negligible sub-region $\tilde{\mathcal{X}}$ of $\mathcal{X}'$, then this certifies that the classifier was already sufficiently competitive on $\mathcal{X}'$ that we can freeze $\mathcal{X}'$.

**Lemma F.3.** *Take any $\mathcal{X}' \subseteq \mathcal{X}$ with label $s_0$. With probability at least $1 - \delta$ over the entire execution of* SPLIT$(\mathcal{X}', s_0, \epsilon, \delta)$, $O((1/\epsilon^2 \vee \frac{L^2}{\epsilon^4 \cdot (\zeta \vee \epsilon)^2}) \cdot \log(1/\delta\epsilon))$ *samples are drawn from $\mathcal{D}'$, the $s$ computed in Step 3 satisfies* $\mathrm{err}_{\mathcal{X}'}(s) \leq 1/2 - \Omega(\epsilon)$ *if* $\mathrm{err}_{\mathcal{X}'}(s_0) \geq 1/2 + \Omega(\epsilon)$ *and $s = s_0$ otherwise, and at least one of the following holds:*

i) SPLIT *outputs $\tilde{\mathcal{X}}, s, \tilde{s}$ for which* $\mathrm{err}_{\tilde{\mathcal{X}}}(\tilde{s}) \leq \mathrm{err}_{\mathcal{X}'}(s) - \epsilon$ *and* $\mu(\tilde{\mathcal{X}} \mid \mathcal{X}') \geq \Omega(\frac{\epsilon^2 \cdot (\zeta \vee \epsilon)}{L(\lambda(\mathcal{X}') + O(\epsilon))})$

ii) SPLIT *outputs* FREEZE *and $s$ for which* $\mathrm{err}_{\mathcal{X}'}(s) \leq \lambda(\mathcal{X}') + 5\epsilon$.

We first show that the LeakyRelu loss incurred by the true classifier $\mathbf{w}^*$ is small. The following lemma is the only place where we make use of our assumptions about $\sigma(\cdot)$ and $\zeta$-misspecification.

**Lemma F.4.** *Let $\epsilon \geq 0$ be arbitrary, and let $\mathcal{X}'$ be any subset of $\mathcal{X}$. Then for $\lambda = \lambda(\mathcal{X}') + \epsilon$,*

$$L_\lambda^{\mathcal{X}'}(\mathbf{w}^*) \leq -\frac{2\epsilon}{L} \cdot \mathop{\mathbb{E}}_{\mathcal{D}'}[|\sigma(\langle \mathbf{w}^*, \mathbf{X} \rangle)|]$$

*Proof.* In this proof, all expectations will be with respect to $\mathcal{D}'$. By definition we have

$$\frac{\mathbb{E}[y\langle \mathbf{w}^*, \mathbf{X} \rangle]}{\mathbb{E}[|\langle \mathbf{w}^*, \mathbf{X} \rangle|]} = \frac{\mathbb{E}[|\langle \mathbf{w}^*, \mathbf{X} \rangle| \cdot \mathrm{sgn}(\langle \mathbf{w}^*, \mathbf{X} \rangle) \cdot (\sigma(\langle \mathbf{w}^*, \mathbf{X} \rangle) + \delta(\mathbf{X}))]}{\mathbb{E}[|\langle \mathbf{w}^*, \mathbf{X} \rangle|]}$$

$$\geq \frac{\mathbb{E}[|\langle \mathbf{w}^*, \mathbf{X} \rangle| \cdot |\sigma(\langle \mathbf{w}^*, \mathbf{X} \rangle)|]}{\mathbb{E}[|\langle \mathbf{w}^*, \mathbf{X} \rangle|]} - 2\zeta$$

$$= \Delta(\mathcal{X}') + \mathrm{Cov}\left(\frac{|\langle \mathbf{w}^*, \mathbf{X} \rangle|}{\mathbb{E}[|\langle \mathbf{w}^*, \mathbf{X} \rangle|]}, \sigma(|\langle \mathbf{w}^*, \mathbf{X} \rangle|)\right)$$

$$\geq \Delta(\mathcal{X}') + \frac{1}{L} \cdot \frac{\mathbb{V}[\sigma(|\langle \mathbf{w}^*, \mathbf{X} \rangle|)]}{\mathbb{E}[|\langle \mathbf{w}^*, \mathbf{X} \rangle|]},$$

where the second step we uses the fact that $\mathrm{sgn}(\sigma(\langle \mathbf{w}^*, \mathbf{X} \rangle)) = \mathrm{sgn}(\langle \mathbf{w}^*, \mathbf{X} \rangle)$ together with the fact that $\delta(\mathbf{X}) \geq -2\zeta$, the third step uses the definition of $\Delta(\mathcal{X}')$, and the fourth step uses $L$-Lipschitz-ness of $\sigma$.

Rearranging, we conclude that

$$(\Delta(\mathcal{X}') - \epsilon)\mathbb{E}[|\langle \mathbf{w}^*, \mathbf{X} \rangle|] - \mathbb{E}[y\langle \mathbf{w}^*, \mathbf{X} \rangle] \leq -\epsilon \cdot \mathbb{E}[|\langle \mathbf{w}^*, \mathbf{X} \rangle|] + \frac{1}{L} \cdot \mathbb{V}[\sigma(|\langle \mathbf{w}^*, \mathbf{X} \rangle|)],$$

from which we get

$$(\Delta(\mathcal{X}') - 2\epsilon)\mathbb{E}[|\langle \mathbf{w}^*, \mathbf{X} \rangle|] - \mathbb{E}[y\langle \mathbf{w}^*, \mathbf{X} \rangle] \leq -\frac{1}{L}\left(2\epsilon \mathbb{E}[|\sigma(\langle \mathbf{w}^*, \mathbf{X} \rangle)|] + \mathbb{V}[\sigma(|\langle \mathbf{w}^*, \mathbf{X} \rangle|)]\right) \quad (14)$$

by using $L$-Lipschitz-ness of $\sigma$ again to upper bound $\mathbb{E}[|\langle \mathbf{w}^*, \mathbf{X} \rangle|]$ by $\frac{1}{L}\mathbb{E}[\sigma(|\langle \mathbf{w}^*, \mathbf{X} \rangle|)]$.

Noting that $\mathbb{E}[y \mid \mathbf{X}] \cdot \langle \mathbf{w}^*, \mathbf{X} \rangle = (1 - 2\,\mathrm{err}_{\mathbf{X}}(\mathbf{w}^*)) \cdot |\langle \mathbf{w}^*, \mathbf{X} \rangle|$, we can rewrite the right-hand side of (14) as

$$\mathbb{E}\left[2|\langle \mathbf{w}^*, \mathbf{X} \rangle|\left(\mathop{\mathrm{err}}_{\mathbf{X}}(\mathbf{w}^*) - \frac{1 - \Delta(\mathcal{X}')}{2} - \epsilon\right)\right] = L_\lambda^{\mathcal{X}'}(\mathbf{w}^*)$$

for $\lambda = \lambda(\mathcal{X}')$ as defined in (13). $\qquad \square$

We will also need the following averaging trick from [DGT19], a proof of which we include for completeness.

**Lemma F.5.** *Take any $\mathcal{X}' \subseteq \mathcal{X}$. For any $\mathbf{w}, \lambda$ for which $L_\lambda^{\mathcal{X}'}(\mathbf{w}) < 0$ and $\|\mathbf{w}\| \leq 1$, there is a threshold $\tau \geq 0$ for which 1) $\mu(\mathcal{X}' \cap \mathcal{A}(\mathbf{w}, \tau) \mid \mathcal{X}') \geq \frac{|L_\lambda^{\mathcal{X}'}(\mathbf{w})|}{2\lambda}$, 2) $\mathrm{err}_{\mathcal{X}' \cap \mathcal{A}(\mathbf{w}, \tau)}(\mathbf{w}) \leq \lambda$.*

*Proof.* Without loss of generality we may assume $\mathcal{X}' = \mathcal{X}$. For simplicity, denote $L_\lambda^{\mathcal{X}}$ by $L$. Let $\xi \triangleq -L(\mathbf{w})/2$. We have that

$$
\begin{aligned}
0 > L(\mathbf{w}) + \xi &\leq L(w) + \xi \cdot \mathop{\mathbb{E}}_{\mathcal{D}_\mathbf{x}}[|\langle \mathbf{w}, \mathbf{X}\rangle|] \\
&= \mathop{\mathbb{E}}_{\mathcal{D}_\mathbf{x}, \mathbf{X}}[(\mathrm{err}(\mathbf{w}) - \lambda + \xi)|\langle \mathbf{w}, \mathbf{X}\rangle|] \\
&= \int_0^1 \mathop{\mathbb{E}}_{\mathcal{D}_\mathbf{x}, \mathbf{X}}[(\mathrm{err}(\mathbf{w}) - \lambda + \xi) \cdot \mathbb{1}[\mathbf{X} \in \mathcal{A}(\mathbf{w}, \tau)]]d\tau,
\end{aligned}
$$

where the second inequality follows by our assumption that $\|\mathbf{w}\| \leq 1$ and $\mathcal{D}$ is supported over the unit ball, and where the integration in the last step is over the Lebesgue measure on $[0, 1]$. So by averaging, there exists a threshold $\tau$ for which 2) holds. Pick $\tau$ to be the minimal such threshold. We conclude that

$$
L(\mathbf{w}) + \xi \geq \int_\tau^1 \mathop{\mathbb{E}}_{\mathcal{D}_\mathbf{x}, \mathbf{X}}[(\mathrm{err}(\mathbf{w}) - \lambda + \xi)\mathbb{1}[\mathbf{X} \in \mathcal{A}(\mathbf{w}, \tau)]]d\tau \geq -\lambda \cdot \mathop{\Pr}_{\mathcal{D}_\mathbf{x}}[\mathbf{X} \in \mathcal{A}(\mathbf{w}, \tau)],
$$

where the first step follows by minimality of $\tau$ and the second follows by the fact that $\mathrm{err}_\mathbf{x}(\mathbf{w}) - \lambda + \xi \geq -\lambda$ for all $\mathbf{x}$. Condition 1) then follows. □

Next we show that, provided $\Delta(\mathcal{X}')$ is not too small, some iteration of the main loop of SPLIT finds a direction $\mathbf{w}_\lambda$ and a threshold $\tau_\lambda$ such that the misclassification error of $\mathbf{w}_\lambda$ over $\mathcal{X}' \cap \mathcal{A}(\mathbf{w}_\lambda, \tau_\lambda)$ is small.

**Lemma F.6.** *Take any $\mathcal{X}' \subseteq \mathcal{X}$. If $\mathbb{E}_{\mathcal{D}'}[|\sigma(\langle \mathbf{w}^*, \mathbf{X}\rangle)|] \geq 2\zeta + \epsilon$, then with probability at least $1 - \delta'$ there is at least one iteration $\lambda$ of the main loop of SPLIT for which the threshold $\tau_\lambda$ found in Step 7 satisfies $\mathrm{err}_{\mathcal{X}' \cap \mathcal{A}(\mathbf{w}_\lambda, \tau_\lambda)}(\mathbf{w}_\lambda) \leq \lambda(\mathcal{X}') + \epsilon$.*

*Proof.* Let $\zeta' \triangleq 2\zeta + \epsilon$. The hypothesis implies $\lambda(\mathcal{X}') < 1/2 - \epsilon/2$. So by searching for $\lambda$ over an $\epsilon/4$-grid of $[0, 1/2]$, we ensure that for some $\lambda$ in the iteration of the main loop of SPLIT, $\epsilon/4 \leq \lambda - \lambda(\mathcal{X}') < \epsilon/2$. Denote this $\lambda$ by $\tilde{\lambda}$.

By Lemma F.4 we have that $L_{\tilde{\lambda}}^{\mathcal{X}'}(\mathbf{w}^*) \leq -\frac{\epsilon\zeta'}{2L}$, so by Lemma B.5, SGD finds a unit vector $\mathbf{w}_{\tilde{\lambda}}$ for which $L_{\tilde{\lambda}}^{\mathcal{X}'}(\mathbf{w}_{\tilde{\lambda}}) \leq -\frac{\epsilon\zeta'}{4L}$ with probability $1 - \delta'$ in $O(\frac{L^2}{\epsilon^2\zeta'^2} \cdot \log(1/\delta'))$ steps. By Lemma F.5 there is a threshold $\tau$ for which 1) $\mu(\mathcal{X}' \cap \mathcal{A}(\mathbf{w}_{\tilde{\lambda}}, \tau) \mid \mathcal{X}') \geq \frac{\epsilon\zeta'}{8L\tilde{\lambda}}$ and 2) $\mathrm{err}_{\mathcal{X}' \cap \mathcal{A}(\mathbf{w}_{\tilde{\lambda}}, \tau)}(\mathbf{w}_{\tilde{\lambda}}) \leq \tilde{\lambda}$.

It remains to verify that a threshold with comparable guarantees can be determined from the empirical distribution $\hat{\mathcal{D}}'$. By the DKW inequality, if the empirical distribution $\hat{\mathcal{D}}'$ is formed from at least $\Omega\left(\frac{L^2\tilde{\lambda}^2}{\epsilon^2\zeta'^2} \cdot \log(1/\delta')\right)$ samples from $\mathcal{D}'$, then with probability at least $1 - \delta'$ we can estimate the CDF of the projection of $\mathcal{D}'$ along direction $\mathbf{w}_{\tilde{\lambda}}$ to within additive $O(\frac{\epsilon\zeta'}{L\tilde{\lambda}})$. On the other hand, if $\hat{\mathcal{D}}'$ is formed from at least $\Omega(\frac{L\tilde{\lambda}}{\epsilon\zeta'} \cdot \frac{1}{\epsilon^2} \cdot \log(1/\delta'))$ samples from $\mathcal{D}'$, then we can estimate the misclassification error on the annulus within $\mathcal{X}'$ to error $O(\epsilon)$. The threshold $\tau_\lambda$ found in Step 7 will therefore satisfy the claimed bound. □

We now apply Lemma F.6 to argue that provided $\Delta(\mathcal{X}')$ is not too small, the constant classifier $\tilde{s}$ defined in Steps 12/14 of SPLIT has good performance on $\tilde{X}_{\tilde{s}}$, and furthermore $\tilde{X}_{\tilde{s}}$ has non-negligible mass.

**Lemma F.7.** *Take any $\mathcal{X}' \subseteq \mathcal{X}$. If $\mathbb{E}_{\mathcal{D}'}[|\sigma(\langle \mathbf{w}^*, \mathbf{X}\rangle)|] \geq 2\zeta + \epsilon$, then with probability at least $1 - O(\delta'/\epsilon)$, we have that $\mu(\tilde{\mathcal{X}}_{\tilde{s}} \mid \mathcal{X}') \geq \Omega(\frac{\epsilon^2 \cdot (\zeta \vee \epsilon)}{L\lambda})$ and $\Pr_{\mathcal{D}'}[y \neq \tilde{s} \mid \mathbf{X} \in \tilde{\mathcal{X}}_{\tilde{s}}] \leq \lambda(\mathcal{X}') + 4\epsilon$.*

*Proof.* Let $\zeta' \triangleq 2\zeta + \epsilon$. With probability at least $1 - O(\delta'/\epsilon)$, for every $\lambda$ in the main loop of SPLIT, the CDF of $\hat{\mathcal{D}}'$ projected along the direction $\mathbf{w}_\lambda$ is pointwise $O\left(\frac{\epsilon\zeta'}{L\lambda}\right)$-accurate, and the empirical misclassification error $\widehat{\mathrm{err}}_{\mathcal{X}' \cap \mathcal{A}(\mathbf{w}_\lambda, \tau_\lambda)}(\mathbf{w}_\lambda)$ is $O(\epsilon)$-accurate. In this case, for the $\lambda$ minimizing empirical misclassification error in Step 8, we have that 1) $\mu(\mathcal{X}' \cap \mathcal{A}(\mathbf{w}_\lambda, \tau_\lambda) \mid \mathcal{X}') \geq \Omega\left(\frac{\epsilon\zeta'}{L\lambda}\right)$, and by Lemma F.6, 2) $\mathrm{err}_{\mathcal{X}' \cap \mathcal{A}(\mathbf{w}_\lambda, \tau_\lambda)}(\mathbf{w}_\lambda) \leq \lambda(\mathcal{X}') + 2\epsilon$.

With $O(\frac{L\lambda}{\epsilon^4 \zeta'} \cdot \log(1/\delta'))$ samples from $\mathcal{D}'$, with probability at least $1 - \delta'$ we can estimate each $\widehat{\mu}(\tilde{\mathcal{X}}_{s'} \mid \mathcal{X}')$ to within $O(\frac{\epsilon^2 \zeta'}{L\lambda})$ error and, if $\min_{s'} \widehat{\mu}(\tilde{\mathcal{X}}_{s'} \mid \mathcal{X}') \geq \Omega(\frac{\epsilon^2 \zeta'}{L\lambda})$, each $\widehat{\mathrm{err}}_{\tilde{\mathcal{X}}_{s'}}(s')$ to within $O(\epsilon)$ error.

Without loss of generality suppose that $+1 = \mathrm{argmin}_{s'} \widehat{\mathrm{err}}_{\tilde{\mathcal{X}}_{s'}}(s')$. By averaging, $\mathrm{err}_{\tilde{\mathcal{X}}_+}(+1) \leq \lambda(\mathcal{X}') + 2\epsilon$. If $\min_{s'} \widehat{\mu}(\tilde{\mathcal{X}}_{s'} \mid \mathcal{X}' \cap \mathcal{A}(\mathbf{w}_\lambda, \tau_\lambda)) \geq \epsilon$, then $\mu(\tilde{\mathcal{X}}_+ \mid \mathcal{X}') \geq \Omega(\epsilon \cdot \frac{\epsilon \zeta'}{L\lambda})$ and we are done.

Otherwise $\min_{s'} \mu(\tilde{\mathcal{X}}_{s'} \mid \mathcal{X} \cap \mathcal{A}(\mathbf{w}_\lambda, \tau_\lambda)) \leq 2\epsilon$, so for $\tilde{s} = \mathrm{argmax}_{s'} \widehat{\mu}(\tilde{\mathcal{X}}_{s'} \mid \mathcal{X}')$ we have that $\mathrm{err}_{\tilde{\mathcal{X}}_{\tilde{s}}}(\tilde{s}) \leq \lambda(\mathcal{X}') + 4\epsilon$ and we are again done. $\square$

We are now ready to complete the proof of Lemma F.3.

*Proof of Lemma F.3.* With probability $1 - \delta'$, $\widehat{\mathrm{err}}_{\mathcal{X}'}(s_0)$ computed in Step 3 is accurate to within error $O(\epsilon)$, so the first part of the Lemma F.3 about $s$ and $s_0$ is immediate. Note that it also implies $\mathrm{err}_{\mathcal{X}'}(s) \leq 1/2 + O(\epsilon)$ unconditionally.

Now suppose outcome i) does not happen. By the contrapositive of Lemma F.7, this implies at least one of two things holds: either $\mathrm{err}_{\mathcal{X}'}(s)$ is already sufficiently low, so that $\mathrm{err}_{\mathcal{X}'}(s) \leq \lambda(\mathcal{X}') + 5\epsilon$, or the hypothesis of Lemma F.7 that $\mathbb{E}_{\mathcal{D}'}[|\sigma(\langle \mathbf{w}^*, \mathbf{X} \rangle)|] \geq 2\zeta + \epsilon$ does not hold.

But if the latter is the case, we would be able to approximate the optimal 0/1 accuracy on $\mathcal{X}'$ just by choosing random signs for the labels on $\mathcal{X}'$. Formally, we have that $\lambda(\mathcal{X}') \geq 1/2 - \epsilon/2$, so because $\mathrm{err}_{\mathcal{X}'}(s) \leq 1/2 + O(\epsilon)$, it certainly holds that $\mathrm{err}_{\mathcal{X}'}(s) \leq \lambda(\mathcal{X}') + 5\epsilon$. $\square$

## F.2 Merging Step

The pseudocode for the merging step is given in Algorithm 11. The following guarantee for MERGE is straightforward.

**Lemma F.8.** *Given partition $\{\mathcal{X}^{(i)}\}$, signs $\{s^{(i)}\}$, and any $\upsilon, \delta > 0$, the following holds with probability $1 - \delta$. If $\mathrm{MERGE}(\{\mathcal{X}^{(i)}\}, \{s^{(i)}\}, \upsilon, \delta)$ outputs None, then for any $i \neq j$ for which $s^{(i)} = s^{(j)}$, we have $|\mathrm{err}_{\mathcal{X}^{(i)}}(s^{(i)}) - \mathrm{err}_{\mathcal{X}^{(j)}}(s^{(j)})| \geq \upsilon/2$. Otherwise, if the output of $\mathrm{MERGE}(\{\mathcal{X}^{(i)}\}, \{s^{(i)}\}, \upsilon, \delta)$ is given by merging some pair of regions $\mathcal{X}^{(i)}, \mathcal{X}^{(j)}$, then $|\mathrm{err}_{\mathcal{X}^{(i)}}(s^{(i)}) - \mathrm{err}_{\mathcal{X}^{(j)}}(s^{(j)})| \leq 3\upsilon/2$.*

*Proof.* With probability $1 - \delta$ we have that every empirical estimate $\widehat{\mathrm{err}}_{\mathcal{X}^{(i)}}(s^{(i)})$ is accurate to within error $\upsilon/4$, so the claim follows by triangle inequality. $\square$

## F.3 Potential Argument

We need to upper bound the number of iterations that LEARNMISSPECGLM runs for; we do this by arguing that we make progress in one of three possible ways:

1. After a SPLIT, a region $\mathcal{X}'$ is frozen. This decreases the total mass of live regions.
2. After a SPLIT, a region $\mathcal{X}'$ is split into $\tilde{\mathcal{X}}$ and its complement, and the regions are assigned opposite labels. This increases the overall accuracy of the classifier.
3. After a SPLIT, a region $\mathcal{X}'$ is split into $\tilde{\mathcal{X}}$ and its complement, and the regions are assigned the same label. This, even with a possible subsequent MERGE, decreases the proportion of the variance of $\sigma(\langle \mathbf{w}^*, \mathbf{X} \rangle)$ unexplained by the partition.

Henceforth, condition on all of the $(1 - \delta)$-probability events of Lemmas F.3 and F.8 holding for every SPLIT and MERGE.

**Lemma F.9.** *There are at most $O(1/\upsilon)$ live regions at any point in the execution of LEARNMISSPECGLM.*

*Proof.* If the number of live regions in the current partition exceeds $2/\upsilon$ after some invocation of SPLIT, then by the contrapositive of the first part of Lemma F.8, the subsequent MERGE will successfully produce a coarser partition. $\square$

**Lemma F.10.** *Event 1 can happen at most $O(\log(1/\epsilon)/\upsilon)$ times.*

*Proof.* Suppose Event 1 happens after SPLIT is called on some region $\mathcal{X}'$. Let $\mathcal{X}_{\text{live}}$ denote the union of all live regions immediately prior to this. Because $\mathcal{X}'$ is, by Step 9 of LEARNMISSPECGLM, the region with largest empirical mass, and its empirical mass is $O(\mu(\mathcal{X}_{\text{live}}) \cdot \upsilon)$-close to its true mass, by Lemma F.9 we must have that $\mu(\mathcal{X}') \geq \mu(\mathcal{X}_{\text{live}}) \cdot \Omega(\upsilon)$, and once $\mathcal{X}'$ is frozen, the mass of the live region decreases by a factor of $1 - \Omega(\upsilon)$.

The total mass of the live regions is non-increasing over the course of LEARNMISSPECGLM, so the claim follows. $\square$

**Lemma F.11.** *Event 2 can happen at most $O(\frac{L}{\epsilon^3 \cdot (\zeta \vee \epsilon)})$ times.*

*Proof.* If $\mathcal{X}'$ was originally assigned some label $s_0$ prior to the invocation of SPLIT, then by design, the label $s$ computed in Step 3 performs at least as well as $s_0$ over $\mathcal{X}'$.

Additionally, if after this SPLIT $\mathcal{X}'$ is broken into $\tilde{\mathcal{X}}$ and $\mathcal{X}'\backslash\tilde{\mathcal{X}}$, and these two sub-regions are assigned opposite labels, we know $\tilde{\mathcal{X}}$ must have been assigned $-s$.

These two facts imply that the overall error of the classifier must decrease by at least $\mu(\tilde{\mathcal{X}}) \cdot (1 - 2\,\text{err}_{\tilde{\mathcal{X}}}(\tilde{s}))$ when Event 2 happens. By Lemma F.3, $\text{err}_{\tilde{\mathcal{X}}}(\tilde{s}) \leq \text{err}_{\mathcal{X}'}(s_0) - \epsilon \leq 1/2 - \epsilon$ and $\mu(\tilde{\mathcal{X}} \mid \mathcal{X}') \geq \Omega(\frac{\epsilon^2 \cdot (\zeta \vee \epsilon)}{L\lambda})$, so it follows that the overall error decreases by at least $\Omega(\frac{\epsilon^3 \cdot (\zeta \vee \epsilon)}{L\lambda})$ when Event 2 happens.

Note that the overall accuracy of the classifier is non-increasing, from which the claim follows. Indeed, the overall accuracy of the classifier is not affected by MERGE or freezes. And when Event 3 happens, the accuracy can only improve, again by the fact that $s$ performs at least as well as $s_0$ over $\mathcal{X}'$ by design. $\square$

**Lemma F.12.** *Event 3 can happen at most $O(L/\epsilon^6)$ times.*

*Proof.* Given partition and labeling, consider the potential function $\Phi(\{\mathcal{X}^{(i)}\}, \{s^{(i)}\}) \triangleq \mathbb{V}_i[\text{err}_{\mathcal{X}^{(i)}}(s^{(i)})]$, where $i$ is chosen from among the live indices of the partition with probability proportional to $\mu(\mathcal{X}^{(i)})$. We will show that Event 3 always increases this quantity and then bound how much the other iterations of LEARNMISSPECGLM decrease this quantity.

First consider how $\Phi$ changes under a SPLIT in which a region $\mathcal{X}'$, previously assigned some label $s^*$, is split into $\tilde{\mathcal{X}}$ and $\mathcal{X}'\backslash\tilde{\mathcal{X}}$ that get assigned the same label $s$. If $\mathcal{X}_{\text{live}}$ is the union of all live regions at that time, then $\Phi$ increases by precisely $\mu(\mathcal{X}' \mid \mathcal{X}_{\text{live}})$ times the variance of the random variable $Z$ which takes value $\text{err}_{\tilde{\mathcal{X}}}(s)$ with probability $\mu(\tilde{\mathcal{X}} \mid \mathcal{X}')$ and $\text{err}_{\mathcal{X}'\backslash\tilde{\mathcal{X}}}(s)$ otherwise. By Lemma F.3, $\mu(\mathcal{X}' \mid \mathcal{X}_{\text{live}}) \geq \Omega(\frac{\epsilon^2 \cdot (\zeta \vee \epsilon)}{L\lambda})$, and moreover

$$\text{err}_{\tilde{\mathcal{X}}}(s) \leq \text{err}_{\mathcal{X}'}(s_0) - \epsilon \leq \text{err}_{\mathcal{X}'\backslash\tilde{\mathcal{X}}}(s) - \epsilon,$$

where $s_0$ is the sign for which $\text{err}_{\mathcal{X}'}(s_0)$ is minimized and the second step follows by an averaging argument. By Fact B.6, we conclude that $\mathbb{V}[Z] \geq \Omega(\epsilon^2 \cdot \frac{\epsilon^2 \cdot (\zeta \vee \epsilon)}{L\lambda(\mathcal{X}')})$.

If MERGE is not called immediately after this, then this occurrence of Event 3 increases $\Phi$ by at least $\mu(\mathcal{X}') \cdot \Omega(\epsilon^2 \cdot \frac{\epsilon^2 \cdot (\zeta \vee \epsilon)}{L\lambda(\mathcal{X}')})$.

If MERGE is called however, $\Phi$ will decrease as follows. Let $\mathcal{X}_1$ and $\mathcal{X}_2$ be the two live regions, both with label $s' \in \{\pm 1\}$, that get merged. Because $\mathcal{X}'$ was chosen to be the largest live region at the time, $\mu(\mathcal{X}_1), \mu(\mathcal{X}_2) \leq \mu(\mathcal{X}') + O(\epsilon \upsilon^2) \leq O(\mu(\mathcal{X}'))$. So MERGE decreases $\Phi$ by at most $\mu(\mathcal{X}_1 \cup \mathcal{X}_2) \leq O(\mu(\mathcal{X}'))$ times the variance of the random variable $Z'$ which takes value $\text{err}_{\mathcal{X}_1}(s')$ with probability $\mu(\mathcal{X}_1 \mid \mathcal{X}_1 \cup \mathcal{X}_2)$ and $\text{err}_{\mathcal{X}_2}(s')$ otherwise. By Lemma F.8, we know $|\text{err}_{\mathcal{X}_1}(s') - \text{err}_{\mathcal{X}_2}(s')| \geq 3\upsilon/2$. So

$$\mathbb{V}[Z'] = O\left(\upsilon^2 \cdot \mu(\mathcal{X}_1 \mid \mathcal{X}_1 \cup \mathcal{X}_2) \cdot \mu(\mathcal{X}_2 \mid \mathcal{X}_1 \cup \mathcal{X}_2)\right) \leq O(\upsilon^2).$$

By taking $\upsilon = \Theta(\epsilon^{3/2})$, we ensure that if Event 3 consists of a SPLIT followed by a MERGE, then $\Phi$ increases by at least $\mu(\mathcal{X}') \cdot \Omega(\epsilon^3) \geq \Omega(\epsilon^{9/2})$, where we used Lemma F.9 and our choice of $\upsilon$ in the last step.

It remains to verify that the other iterations of LEARNMISSPECGLM do not decrease $\Phi$ by too much. Clearly any SPLIT that does not freeze a region will only ever increase $\Phi$, so we just need to account for A) invocations of MERGE immediately following a SPLIT under Event 2, and B) freezes.

For A), note that by the above calculations, SPLIT under Event 2 followed by a MERGE will decrease $\Phi$ by at most $\mathbb{V}[Z'] \leq O(\epsilon^3)$, and by Lemma F.11 this will happen at most $O(\frac{L}{\epsilon^3 \cdot (\zeta \vee \epsilon)})$ times. So all invocations of MERGE following an occurrence of Event 2 will collectively decrease $\Phi$ by at most $O(\frac{L}{\zeta \vee \epsilon} \leq O(L/\epsilon)$.

For B), we will naively upper bound the decrease in variance from a freeze by 1/4. By Lemma F.10 this happens at most $O(\log(1/\epsilon)/\upsilon) \leq O(\log(1/\epsilon)/\epsilon^{3/2})$ times. The claim follows. $\qquad\square$

**Corollary F.13.** LEARNMISSPECGLM *makes at most* $O(L/\epsilon^6)$ *calls to either* SPLIT *or* MERGE.

*Proof.* The number of invocations of MERGE is clearly upper bounded by the number of invocations of SPLIT, which is upper bounded by Lemmas F.10, F.11, and F.12. $\qquad\square$

### F.4 Putting Everything Together

We can now complete the proof of Theorem F.1.

*Proof of Theorem F.1.* By Corollary F.13 and a union bound over the conclusions of Lemmas F.3 and F.8 holding for the first $O(L/\epsilon^6)$ iterations, LEARNMISSPECGLM outputs a hypothesis $\mathbf{h}$ after $O(L/\epsilon^6)$ iterations. Furthermore, each iteration of LEARNMISSPECGLM calls SPLIT and Merge at most once each and draws $O(\log(1/\delta')/\mathrm{poly}(\epsilon))$ additional samples, so by the sample complexity bounds in Lemma F.3 and F.8, LEARNMISSPECGLM indeed has the claimed sample complexity. The $\mathrm{poly}(N) \cdot d$ runtime is also immediate.

It remains to verify that the output $\mathbf{h}$ is $O(\epsilon)$-competitive with $\lambda(\mathcal{X})$. For any frozen region $\mathcal{X}'$ of $\mathbf{h}$ with label $s$, the last part of Lemma F.3 tells us that $\mathrm{err}_{\mathcal{X}'}(s) \leq \lambda(\mathcal{X}') + 5\epsilon$. On the other hand, at the end of the execution of LEARNMISSPECGLM, $\mu(\mathcal{X}_{\mathrm{live}}) \leq 2\epsilon$, while

$$|\lambda(\mathcal{X}\backslash\mathcal{X}_{\mathrm{live}}) - \lambda(\mathcal{X})| = \frac{1}{2}|\Delta(\mathcal{X}\backslash\mathcal{X}_{\mathrm{live}}) - \Delta(\mathcal{X})| = \frac{\mu(\mathcal{X}_{\mathrm{live}})}{2} \cdot \left|\mathrm{err}_{\mathcal{X}_{\mathrm{live}}}(\mathbf{h}) - \mathrm{err}_{\mathcal{X}\backslash\mathcal{X}_{\mathrm{live}}}(\mathbf{h})\right| \leq \epsilon$$

by a union bound. So

$$\mathrm{err}_{\mathcal{X}}(\mathbf{h}) \leq \mu(\mathcal{X}\backslash\mathcal{X}_{\mathrm{live}}) \cdot (\lambda(\mathcal{X}\backslash\mathcal{X}_{\mathrm{live}}) + 5\epsilon) + \epsilon \leq \lambda(\mathcal{X}) + 7\epsilon,$$

as claimed. $\qquad\square$

### F.5 Making it Proper in the Massart Setting

When $\zeta = 0$, the Generalized Linear Model is always an instance of the $\eta = 1/2$ Massart halfspace model. This means we can use the proper-to-improper reductions we developed in Section D to convert our improper learner to a proper learner. The proper-to-improper reduction will require either a margin or a bit complexity bound; below we give the result under the natural margin assumption for GLMs.

**Theorem F.14.** *Fix any $\epsilon > 0$. Let $\mathcal{D}$ be a distribution arising from an 0-misspecified GLM with odd link function $\sigma(\cdot)$ and true weight vector $\mathbf{w}^*$. Suppose we have the following $\gamma$-margin assumption: that $|\sigma(\langle\mathbf{w}^*, \mathbf{X}\rangle)| \geq \gamma$ almost surely. With probability $1 - \delta$, algorithm LEARNMISSPECGLM($\mathcal{D}, \epsilon, \delta/2$) combined with Algorithm FILTERTRONDISTILL with $\eta = (1 - \gamma)/2$ outputs a $\mathbf{w}$ for which*

$$\mathrm{err}_{\mathcal{X}}(\mathbf{w}) \leq \frac{1 - \mathbb{E}_{\mathcal{D}}[\sigma(|\langle\mathbf{w}^*, \mathbf{X}\rangle|)]}{2} + O(\epsilon).$$

*Moreover, the algorithm has sample complexity $N = \mathrm{poly}(L, 1/\gamma, \epsilon^{-1}, (\zeta \vee \epsilon)^{-1}) \cdot \log(1/\delta)$ and runs in time $d \cdot \mathrm{poly}(N)$, where $d$ is the ambient dimension.*

It is also possible to modify the GLM learning algorithm directly in this case and extract a separation oracle, giving a slightly different proof of the above result: whenever the algorithm finds a region on which the current $\mathbf{w}$ is performing significantly worse than the best constant function, that region can be fed into the separation oracle for the proper halfspace learning algorithm.

In the more general setting $\zeta > 0$, we do not expect a proper to improper reduction to be possible because agnostic learning is embeddable (by adding a lot of noise of $Y$), and in the agnostic setting there are complexity-theoretic barriers [ABX08].

## G   Numerical Experiments

Figure 2: **Synthetic data**: Effect of RCN versus Massart noise on FILTERTRON and baselines

We evaluated FILTERTRON, gradient descent on the LeakyRelu loss, random forest[8], and logistic regression[9] on both synthetic and real-world datasets in the presence of (synthetic) Massart noise. Note that all of the methods considered *except* for random forest output simple halfspace classifiers; we included random forest as a benchmark for methods which are allowed to output somewhat more complex and less interpretable predictors. Because the theoretical guarantees in this work are in terms of zero-one error on the distribution *with* Massart corruptions, we measure the performance using this metric, i.e. both the training and test set labels are corrupted by the Massart adversary.

### G.1   Experiments on Synthetic Data

We evaluated the above algorithms on the following distribution. Let $\mathcal{D}_{\mathbf{x}}$ be a uniform mixture of $\mathcal{N}(0, \mathbf{I}_2)$ and $\mathcal{N}(0, \Sigma)$, where $\Sigma = \begin{pmatrix} 8 & 0.1 \\ 0.1 & 0.024 \end{pmatrix}$, and let $\mathbf{w} = (1, 0)$ be the true direction for the underlying halfspace. For various $\eta$, we considered the following $\eta$-Massart adversary: the labels for all $(x_1, x_2) \sim \mathcal{D}_{\mathbf{x}}$ for which $x_2 > 0.3$ are flipped with probability $\eta$, and the labels for all other points are not flipped.

For every $\eta \in [0.05, 0.1, \ldots, 0.45]$, we ran the following experiment 50 times: (1) draw 1250 samples from the mixture of Gaussians, label them according to halfspace $\mathbf{w}$, and randomly split them into a training set of size 1000 and a test set of size 250, (2) randomly flip the labels for the training and test sets according to this Massart adversary, (3) train on the noisy training set, and (4) evaluate according to zero-one loss on the noisy test set. For both FILTERTRON and gradient descent on LeakyRelu, we ran for a total of 2000 iterations with step size 0.05 and $\epsilon = 0.05$. We then ran the exact same experiment but with RCN rather than Massart noise. The results of these experiments are shown in Figure 2.

Regarding our implementation of random forest, we found that taking a smaller max depth led to better performance, so we chose this parameter to be five.

Note that whereas under RCN, the baselines are comparable to FILTERTRON especially for large $\eta$, logistic regression and gradient descent on LeakyRelu perform significantly worse under Massart noise. The intuition is that by reducing the noise along some portion of the distribution, in particular the portion mostly orthogonal to $\mathbf{w}$, we introduce a spurious correlation that encourages logistic regression and gradient descent on the LeakyRelu loss to move in the wrong direction. On the other hand, random forest appears to do quite well. We leave open the possibility that more sophisticated synthetic examples can cause FILTERTRON to outperform random forest.

These two experiments on synthetic data were conducted on a MacBook Pro with 2.6 GHz Dual-Core Intel Core i5 processor and 8 GB of RAM and each took roughly 15 minutes of CPU time, indicating that each run of FILTERTRON took roughly two seconds.

## G.2    Experiments on UCI Adult Dataset

We also evaluated the above algorithms on the UCI Adult dataset, obtained from the UCI Machine Learning Repository [DG17] and originally curated by [Koh96]; it consists of demographic information for $N = 48842$ individuals, with a total of 14 attributes including age, gender, education, and race, and the prediction task is to determine whether a given individual has annual income exceeding 50K. Henceforth we will refer to individuals with annual income exceeding (resp. at most) 50K as *high (resp. low) income*. In regards to our experiments, some important statistics about the individuals in the dataset are that 23.9% are high-income, 9.6% are African American, 1.2% are high-income and African American, 33.2% are Female, 3.6% are high-income and Female, 10.3% are Immigrants, and 2.0% are high-income Immigrants.

For various $\eta$ and various predicates $p$ on demographic information, we considered the following $\eta$-Massart adversary: for individuals who satisfy the predicate $p$ (the *target* group), do not flip the response, and for all other individuals, flip the response with probability $\eta$. The intuition is that because most individuals in the dataset are low-income, the corruptions will make individuals not satisfying the predicate $p$ appear to be higher-income on average, which may bias the learner against classifying individuals satisfying $p$ as high-income. We measured the performance of a classifier under this attack along two axes: A) their accuracy over the entire test set, and B) their accuracy over the high-income members of the target group in the test set.

Concretely, for every predicate $p$, we took a five-fold cross validation of the dataset, and for every $\eta \in [0, 0.1, 0.2, 0.3, 0.4]$ we repeated the following five times: (1) randomly flip the labels for the training and test set, (2) train on the noisy training set, and (3) evaluate according to $(A)$ and $(B)$ above. For both FILTERTRON and gradient descent on the LeakyRelu loss, we ran for 2000 iterations and chose the $\epsilon$ parameter by a naive grid search over $[0.05, 0.1, 0.15, 0.2]$.

For our implementation of random forest, we found that a larger maximum depth improved performance, so we took this parameter to be 20.

The predicates $p$ that we considered were (1) African American, (2) Female, and (3) Immigrant. Figure 3 plots the medians across each five-fold cross-validations, with error bars given by a single standard deviation. In all cases, while the algorithms evaluated achieve very similar test accuracy, FILTERTRON correctly classifies a noticeably larger fraction of high-income members of the target group than logistic regression or gradient descent on LeakyRelu, and is comparable to random forest. Note that in the sense of equality of opportunity [HPS16] (see Section G.3 for further discussion on fairness issues), the behavior of gradient descent on LeakyRelu is particularly problematic as its false negative rate among individuals outside of the target group is highest of all classifiers, in spite of its poor performance on the target group.

An important consideration in some applications is that certain demographic information may be hidden from the learner, either because they are unavailable or because they have been withheld for legal reasons. Motivated by this, we also considered how well our classifier would perform under such circumstances. We reran the experiment outlined above with the sole difference that after the training and test labels have been randomly flipped according to the Massart adversary targeting a predicate $p$, we pass to the learner the *censored* dataset obtained by removing all dimensions of the data corresponding to demographic information relevant to $p$. For instance, for the adversary

Figure 3: **UCI Adult**: Effect of three different Massart adversaries, targeting African Americans, Females, and Immigrants respectively, on the accuracy of FILTERTRON and baselines. Top figures indicate accuracy over entire test set, bottom figures indicate accuracy over the target group.

targeting immigrated, we removed data identifying the country of origin for individuals in the dataset. The rest of the design of the experiment is exactly the same as above, and Figure 4 depicts the results.

Note that the false negative rate among African Americans under logistic regression still degenerates dramatically as $\eta$ increases. Interestingly though, gradient descent on LeakyRelu no longer breaks down but in fact has lowest false negative rate among all classifiers at high $\eta$, though as with the previous experiment, its overall accuracy is slightly lower.

The experiments on the Adult dataset were conducted in a Kaggle kernel with a Tesla P100 GPU, and each predicate took roughly 40 minutes to run. All code is available at `https://github.com/secanth/massart`.

## G.3 Situating Our Results within the Fairness Literature

There is by now a mature literature on algorithmic fairness [DHP+12, HPS16, KMR17], with many mathematically well-defined notions of what it means to be fair that themselves come from different normative considerations. There is no one notion that clearly dominates the others, but rather it depends on the circumstances and sometimes they are even at odds with each other [Cho17, KMR17, MP19]. Our results are perhaps most closely related to the notion of *equal opportunity* [HPS16], where our experiments show that many off-the-shelf algorithms achieve high false negative rate on certain demographic groups when noise is added to the rest of the data. We view our work as making a tantalizing connection between robustness and fairness in the sense that tools from the burgeoning area of algorithmic robustness [DKK+19, LRV16], such as being able to tolerate noise rates that vary across the domain, may ultimately be a useful ingredient in mitigating certain kinds of unfairness that can arise from using well-known algorithms that merely attempt to learn a good predictor in aggregate. Our specific techniques are built on top of new efficient algorithms to search for portions of the distribution where a classifier is performing poorly and can be improved. We believe that even these tools may find other compelling applications, particularly because as Figure 4 shows, they do not need to explicitly rely on demographic information being present within the data, which is an important consideration in some applications.

Figure 4: **UCI Adult**: Results of the same experiment that generated Figure 3, with the sole difference that for each given target group, the corresponding demographic fields (race, gender, and nationality respectively) were removed from the dataset after Massart corruption

## H    Proof of Lemma C.3

### H.1    Tools from Empirical Process Theory

We recall a number of standard tools from empirical process theory, referencing the textbook [Ver18]; some alternative references include [vdG00, vH14]. Recall that a Rademacher random variable is equal to $+1$ with probability $1/2$ and $-1$ with probability $1/2$.

**Theorem H.1** (Symmetrization, Exercise 8.3.24 in [Ver18]). *For any class of measurable functions $\mathcal{F}$ valued in $\mathbb{R}$ and i.i.d. $X_1, \ldots, X_n$ copies of random variable $X$,*

$$\mathbb{E} \sup_{f \in \mathcal{F}} \left| \frac{1}{n} \sum_{i=1}^{n} f(X_i) - \mathbb{E}[f(X)] \right| \leq 2 \mathbb{E} \sup_{f \in \mathcal{F}} \left| \frac{1}{n} \sum_{i=1}^{n} \sigma_i f(X_i) \right|$$

*where the $\sigma_1, \ldots, \sigma_n$ are i.i.d. Rademacher random variables, independent of the $X_i$.*

**Theorem H.2** (Contraction Principle, Theorem 6.7.1 of [Ver18]). *For any vectors $\mathbf{x}_1, \ldots, \mathbf{x}_n$ in an arbitrary normed space,*

$$\mathbb{E} \left\| \sum_{i=1}^{n} a_i \sigma_i \mathbf{x}_i \right\| \leq \|a\|_\infty \mathbb{E} \left\| \sum_{i=1}^{n} \sigma_i \mathbf{x}_i \right\|$$

*where the expectation is over i.i.d. Rademacher random variables $\sigma_1, \ldots, \sigma_n$.*

**Definition H.3.** *Let $\Omega$ be a set and let $\mathcal{F}$ be a class of boolean functions on $\Omega$, i.e. functions of type $\Omega \to \{0, 1\}$. A finite subset $\Lambda \subset \Omega$ is* shattered *by $\mathcal{F}$ if the restriction of $\mathcal{F}$ to $\Lambda$ contains all functions $\Lambda \to \{0, 1\}$. The* VC dimension *of $\mathcal{F}$ is the size of the largest such $\Lambda$ which can be shattered by $\mathcal{F}$.*

**Theorem H.4** (McDiarmid's inequality, Theorem 2.9.1 of [Ver18]). *Supose $X_1, \ldots, X_n$ are independent random variables valued in set $\mathcal{X}$ and $\mathbf{X} = (X_1, \ldots, X_n)$. Suppose that for all $i$, the measurable function $f : \mathcal{X}^n \to \mathbb{R}$ satisfies the bounded difference property $|f(\mathbf{x}) - f(\mathbf{x}')| \leq L$ for all $\mathbf{x}, \mathbf{x}' \in \mathcal{X}$ differing in only one coordinate. Then*

$$\Pr[f(\mathbf{X}) - \mathbb{E}[f(\mathbf{X})] \geq t] \leq \exp\left( -\frac{2t^2}{nL^2} \right).$$

**Theorem H.5** (Theorem 8.3.23 of [Ver18])**.** *Let $\mathcal{F}$ be a class of Boolean functions with VC dimension $d$. If $X_1, \ldots, X_n$ are i.i.d. copies of random variable $X$ then*

$$\mathbb{E} \sup_{f \in \mathcal{F}} \left| \frac{1}{n} \sum_{i=1}^{n} f(X_i) - \mathbb{E}[f(X)] \right| \leq 2\mathbb{E} \sup_{f \in \mathcal{F}} \left| \frac{1}{n} \sum_{i=1}^{n} \sigma_i f(X_i) \right| = O\left( \sqrt{\frac{d}{n}} \right)$$

*where the $\sigma_i$ are i.i.d. Rademacher random variables.*

In the following Lemma, we compute the VC dimension of the class of slabs along a fixed direction.

**Lemma H.6.** *Fix $\mathbf{w} \in \mathbb{R}^d$. Let $\mathcal{F}_{\mathbf{w}}$ be the class of indicators of slabs $\mathcal{S}(\mathbf{w}, r)$ for all $r \geq 0$. The VC Dimension of $\mathcal{F}_{\mathbf{w}}$ is 1.*

*Proof.* The lower bound of 1 is clear, as any point $\mathbf{w}$ with $|\langle \mathbf{w}, \mathbf{x} \rangle| > 0$ is contained in every slab with $r > |\langle \mathbf{w}, \mathbf{x} \rangle|$ and not contained in any slab with $r < |\langle \mathbf{w}, \mathbf{x} \rangle|$. To show the VC dimension is strictly less than two: consider two points $\mathbf{x}, \mathbf{x}'$ and suppose without loss of generality that $|\langle \mathbf{w}, \mathbf{x} \rangle| \leq |\langle \mathbf{w}, \mathbf{x}' \rangle|$, we see that every slab containing $\mathbf{x}'$ contains $\mathbf{x}$, so it is impossible to shatter these points. $\square$

## H.2 Proof of Lemma C.3

At a high level, Lemma C.3 shows that the empirical process defined by looking at the LeakyRelu loss over slabs with $r \in R$ is no harder to control (in terms of sample complexity) than controlling what happens at the thinnest slab, for which the sample complexity is controlled by Bernstein's inequality. This happens because: (1) larger slabs receive more samples and so are very well-behaved, and (2) every slab is completely contained within every larger slab, so slabs with similar mass have a lot of overlap and behave similarly.

Based on this intuition, we split the proof of Lemma C.3 into two steps. First we prove the result for slabs with a relatively large amount of probability mass. Second, we show how to deduce the general result by a peeling argument, which groups slabs by probability mass. We further split the first step into two Lemmas — the first Lemma below contains the empirical process bound coming from slabs having bounded VC dimension.

**Lemma H.7.** *Let $\lambda \in [0, 1/2]$ be arbitrary. Suppose that $\mathbf{w}^*, \mathbf{w} \in \mathbb{R}^d$ and $\mathbf{X}$ is a random vector in $\mathbb{R}^d$ such that $|\langle \mathbf{w}^*, \mathbf{X} \rangle| \leq 1$ almost surely. Define $\mathcal{R} = \{r \geq 0 : \Pr[\mathbf{X} \in \mathcal{S}(\mathbf{w}, r)] \geq 1/2\}$. Suppose that*

$$\mathbb{E}_{\mathcal{D}}[\ell_\lambda(\mathbf{w}^*, \mathbf{X}) \cdot \mathbb{1}[\mathbf{X} \in \mathcal{S}(\mathbf{w}, r)]] \leq -\gamma$$

*for some $\gamma \in \mathbb{R}$ and all $r \in \mathcal{R}$. Then if $\hat{\mathcal{D}}$ is the empirical distribution formed by $n$ i.i.d. samples $\mathbf{X}_1, \ldots, \mathbf{X}_n$ from $\mathcal{D}$,*

$$\sup_{r \in \mathcal{R}} \mathbb{E}_{\hat{\mathcal{D}}}[\ell_\lambda(\mathbf{w}^*, \mathbf{X}) \cdot \mathbb{1}[\mathbf{X} \in \mathcal{S}(\mathbf{w}, r)]] \leq -\gamma + O\left( \sqrt{\log(2/\delta)/n} \right)$$

*with probability at least $1 - \delta$.*

*Proof.* By McDiarmid's inequality (Theorem H.4),

$$\sup_{r \in \mathcal{R}} \mathbb{E}_{\hat{\mathcal{D}}}[\ell_\lambda(\mathbf{w}^*, \mathbf{X}) \cdot \mathbb{1}[\mathbf{X} \in \mathcal{S}(\mathbf{w}, r)]] \leq \mathbb{E} \sup_{r \in \mathcal{R}} \mathbb{E}_{\hat{\mathcal{D}}}[\ell_\lambda(\mathbf{w}^*, \mathbf{X}) \cdot \mathbb{1}[\mathbf{X} \in \mathcal{S}(\mathbf{w}, r)]] + O(\sqrt{\log(2/\delta)/n})$$

using the bounded differences properties with respect to the independent samples $\mathbf{X}_1, \ldots, \mathbf{X}_n$ (it suffices to check this property for each particular value of $r \in \mathcal{R}$, and then it follows from the Lipschitz property of $\ell_\lambda$ and the fact $|\langle w^*, \mathbf{X} \rangle| \leq 1$ almost surely).

It remains to upper bound the first term. By the assumed upper bound on the true LeakyRelu loss, symmetrization (Theorem H.1), and contraction (Theorem H.2),

$$\mathbb{E}\sup_{r\in\mathcal{R}}\mathbb{E}_{\hat{\mathcal{D}}}[\ell_\lambda(\mathbf{w}^*,\mathbf{X})\cdot\mathbb{1}[\mathbf{X}\in\mathcal{S}(\mathbf{w},r)]]=\mathbb{E}\sup_{r\in\mathcal{R}}\frac{1}{n}\sum_{i=1}^{n}\ell_\lambda(\mathbf{w}^*,\mathbf{X}_i)\cdot\mathbb{1}[\mathbf{X}_i\in\mathcal{S}(\mathbf{w},r)]$$

$$\leq-\gamma+2\mathbb{E}\sup_{r\in\mathcal{R}}\left|\frac{1}{n}\sum_{i=1}^{n}\sigma_i\ell_\lambda(\mathbf{w}^*,\mathbf{X}_i)\cdot\mathbb{1}[\mathbf{X}_i\in\mathcal{S}(\mathbf{w},r)]\right|$$

$$\leq-\gamma+2\mathbb{E}\sup_{r\in\mathcal{R}}\left|\frac{1}{n}\sum_{i=1}^{n}\sigma_i\mathbb{1}[\mathbf{X}_i\in\mathcal{S}(\mathbf{w},r)]\right|.$$

Finally, by Lemma H.6 and Theorem H.5, the last term is upper bounded by $O(\sqrt{1/n})$. □

**Lemma H.8.** *Let $\lambda\in[0,1/2]$ be arbitrary. Suppose that $\mathbf{w}^*,\mathbf{w}\in\mathbb{R}^d$ and $\mathbf{X}$ is a random vector in $\mathbb{R}^d$ such that $|\langle\mathbf{w}^*,\mathbf{X}\rangle|\leq1$ almost surely. Define $\mathcal{R}=\{r\geq0:\Pr[\mathbf{X}\in\mathcal{S}(\mathbf{w},r)]\geq1/2\}$. Suppose that*

$$L_\lambda^{\mathcal{S}(\mathbf{w},r)}(\mathbf{w}^*)\leq-\gamma \tag{15}$$

*for some $\gamma>0$ and all $r\in\mathcal{R}$. Then if $\hat{\mathcal{D}}$ is the empirical distribution formed by $n$ i.i.d. samples $\mathbf{X}_1,\ldots,\mathbf{X}_n$ from $\mathcal{D}$,*

$$\sup_{r\in\mathcal{R}}\hat{L}_\lambda^{\mathcal{S}(\mathbf{w},r)}(\mathbf{w}^*)\leq-\gamma/4$$

*with probability at least $1-\delta$ as long as $n=\Omega(\log(2/\delta)/\gamma^2)$.*

*Proof.* Observe that

$$\hat{L}_\lambda^{\mathcal{S}(\mathbf{w},r)}(\mathbf{w}^*)=\frac{\mathbb{E}_{\hat{\mathcal{D}}}[\ell_\lambda(\mathbf{w}^*,\mathbf{X})\cdot\mathbb{1}[\mathbf{X}\in\mathcal{S}(\mathbf{w},r)]]}{\Pr_{\hat{\mathcal{D}}}[\mathbf{X}\in\mathcal{S}(\mathbf{w},r)]} \tag{16}$$

and the corresponding equation for $L_\lambda$ together with (15) shows that $\mathbb{E}_{\mathcal{D}}[\ell_\lambda(\mathbf{w}^*,\mathbf{X})\cdot\mathbb{1}[\mathbf{X}\in\mathcal{S}(\mathbf{w},r)]]\leq-\gamma/2$. The result then follows from Lemma H.7 and the assumed lower bound on $n$, since this upper bounds the numerator in (16) and the denominator is always at most 1. □

*Proof of Lemma C.3.* Define $R^*=\{r>0:\Pr_{\hat{\mathcal{D}}}[\mathbf{X}\in\mathcal{S}(\mathbf{w},r)]\geq\epsilon/2\}$. We partition $R^*$ as

$$R_k^*=\{r>0:\Pr_{\hat{\mathcal{D}}}[\mathbf{X}\in\mathcal{S}(\mathbf{w},r)]\in(\epsilon2^{k-1},\epsilon2^k]\}.$$

For each bucket $k$, let $r_k=\max\mathcal{R}_k^*$ which exists because the CDF is always right continuous. Let $p_k=\min(1/2,\Pr[\mathbf{X}\in\mathcal{S}(\mathbf{w},r_k)])\leq\epsilon2^k$ and observe that $\mathbb{V}_{Z\sim Ber(p_k)}[Z]\in[p_k/2,p_k]$. Using Bernstein's inequality (Theorem B.8), we have that

$$\Pr[\Pr_{\hat{\mathcal{D}}}[\mathbf{X}\in\mathcal{S}(\mathbf{w},r_k)]\leq p_k-t]\leq2\exp\left(\frac{-nt^2/2}{p_k+t/3}\right)$$

so taking $t=p_k/2$ we find

$$\Pr[\Pr_{\hat{\mathcal{D}}}[\mathbf{X}\in\mathcal{S}(\mathbf{w},r_k)]\leq p_k/2]\leq2\exp\left(-cnp_k\right)$$

for some absolute constant $c>0$.

Let $\delta_0>0$ be a constant to be optimized later and define $\delta_k=2^{-k}\delta_0$; applying Lemma H.8 with $\delta_k$ as the failure probability shows that conditional on having $\Omega(k\log(2/\delta_k)/\gamma^2)$ samples fall into $R_k^*$, with probability at least $1-\delta_k$ we have $\sup_{r\in R_k^*}\hat{L}_\lambda^{\mathcal{S}(\mathbf{w},r)}\leq-\gamma/4$. Requiring $\epsilon n=\Omega(\log(2/\delta_0)/\gamma^2)$, we see that the total probability of failure in bucket $k$ is at most $1-(1-\delta_k)(1-2\exp(-cnp_k))\leq\delta_k+2\exp(-cnp_k)$ since $p_kn\epsilon/2=\Omega(2^k\log(2/\delta_k)/\gamma^2)=\Omega(k\log(2/\delta_k)/\gamma^2)$, which means that (more than) enough samples fall into each bucket to apply Lemma H.8 with failure probability $\delta_k$.

Therefore applying the union bound, we have that $\sup_{r \in R^*} \hat{L}_\lambda^{\mathcal{S}(\mathbf{w},r)} \leq -\gamma/4$ with failure probability at most

$$\sum_{k \geq 0, \epsilon 2^k < 1} (\delta_k + 2\exp(-cnp_k)) \leq \sum_{k=0}^{\infty} \delta_k + \sum_{k=0}^{\infty} 2\exp(-cn\epsilon 2^{k-1}) \leq \delta_0 + c'\exp(-cn\epsilon)$$

for some absolute constant $c' > 0$, using that both infinite sums converge and are dominated by their first term (the first sum is a geometric series, for the second sum its terms shrink doubly exponentially fast towards zero). Using the lower bound on $\epsilon n$ (and recalling that $\gamma < 1$) gives that the failure probability is at most $c''\delta_0$ It follows that $\sup_{r \in R_k^*} \hat{L}^{\mathcal{S}(\mathbf{w},r)})_\lambda \leq -\gamma/4$ with probability at least $1 - \delta/2$ as long as $n = \Omega(\log(2/\delta)/\epsilon\gamma^2)$.

Finally, to prove the desired result we need to show that $\mathcal{R} \subset \mathcal{R}^*$ with probability $1 - \delta/2$. This follows under our assumed lower bound on $n$ by applying (similar to above) Bernstein's inequality to upper bound $\Pr[\mathcal{X} \in \mathcal{S}(\mathbf{w}, r_{-1})]$ for $r_{-1}$ defined following the convention above. Using the union bound, this proves the Lemma. $\qquad\square$

# I  Massart Noise and Non-Oblivious Adversaries

We note that all the algorithms presented, like that of [DGT19], are robust against the following stronger noise model. Formally,

**Definition I.1.** *(Non-oblivious Massart Adversary) let* $\hat{\mathcal{D}}_\mathbf{x} = \{\mathbf{x}_1, ..., \mathbf{x}_N\}$ *be* $N$ *draws from a distribution* $\mathcal{D}_\mathbf{x}$. *Let* $r_1, ..., r_N \sim Bern(\eta)$. *We then let the adversary choose any bit string* $z \in \{\pm 1\}^N$ *possibly depending on* $\hat{\mathcal{D}}_\mathbf{x}, \mathbf{w}^*, \{r_i\}_{i=1}^N$. *The label* $y_i$ *will be determined to be* $y_i = (1 - r_i)\sigma(\langle \mathbf{w}^*, x_i \rangle) + r_i z_i$. *We denote the full dataset* $\hat{\mathcal{D}} = \{(\mathbf{x}_i, y_i)\}_{i=1}^N$.

In the non-oblivious model we measure error in terms of empirical error on $\hat{\mathcal{D}}$. In a learning setting, we can interpret this as drawing a constant factor more samples corrupted by the non-oblivious adversary and splitting the dataset randomly into train and test. Our algorithmic guarantees then hold over the test data.

We can run through all our analyses replacing $\eta(\mathbf{x}_i)$ with $r_i z_i$. For succinctness we demonstrate this for the no margin proper halfspace learner. Note that our results concerning the behavior of the LeakyRelu loss on anulli and slabs such as Lemma C.4 do not use the Massart noise assumption. Thus, it suffices to prove that under the non oblivious adversary, the behavior of the LeakyRelu loss on $\mathbf{w}^*$ is unchanged.

**Lemma I.2.** *For a dataset* $\hat{\mathcal{D}}$ *corrupted by the non-oblivious massart adversary above, specified up to b bits of precision in d dimensions for which there are no* $\beta = \tilde{O}(\frac{db}{\epsilon})$ *outliers i.e for all* $\mathbf{x} \in \hat{\mathcal{D}}$, *we have* $\langle \mathbf{x}, u \rangle^2 \leq \beta \mathbb{E}[\langle \mathbf{x}, u \rangle^2]$ *over all unit vectors* $u \in \mathbb{S}^{d-1}$. *Then for* $\lambda \geq \eta + \epsilon$ *and* $N = \tilde{O}(\frac{d^3 b^3}{\epsilon^3})$, *we have with high probability over the randomness in* $r_1, ..., r_N$

$$LeakyRelu_\lambda(\mathbf{w}^*) \leq 0$$

*Proof.* We will use the notation $\mathbb{E}_{\mathbf{X} \sim \hat{\mathcal{D}}_\mathbf{x}}$ to represent an average over the $\mathbf{X}$ in $\hat{\mathcal{D}}_\mathbf{x}$.

$$LeakyRelu_\lambda(\mathbf{w}^*) = \mathbb{E}_{\mathbf{x} \sim \hat{\mathcal{D}}_\mathbf{x}}[(\text{err}(\mathbf{w}^*) - \lambda)|\langle \mathbf{w}^*, \mathbf{x} \rangle|]$$
$$= \mathbb{E}_{\mathbf{x} \sim \hat{\mathcal{D}}_\mathbf{x}}[(r_i z_i - \lambda)|\langle \mathbf{w}^*, \mathbf{x} \rangle|]$$
$$\leq \mathbb{E}_{\mathbf{x} \sim \hat{\mathcal{D}}_\mathbf{x}}[(r_i - \lambda)|\langle \mathbf{w}^*, \mathbf{x} \rangle|]$$

In the last inequality we make use of the fact $r_i z_i \leq r_i$. Note that we have made no appeals to concentration up to this point. Now it suffices to show $\mathbb{E}_{\mathbf{x} \sim \hat{\mathcal{D}}}[(r_i - \lambda)|\langle \mathbf{w}^*, \mathbf{x} \rangle|] < 0$ with high probability over the randomness in in $r$. This follows from a standard application of bernstein as we did in Theorem C.18. $\qquad\square$

## Footnotes

[4]Crucially, we can *distinguish* which case we are in without knowing these quantities (see Lemma F.3).

[5]It is standard in such settings to assume $\mathcal{D}_{\mathbf{x}}$ has bounded support. We can reduce from this to the unit ball case by normalizing points in the support and scaling $L$ appropriately.

[6]More precisely, our sample complexity is $\operatorname{poly}(L', \epsilon^{-1}, (\zeta \vee \epsilon)^{-1}) \cdot \log(1/\delta)$, where $L' \triangleq L \cdot \inf_{\mathcal{X}' \subseteq \mathcal{X}} \lambda(\mathcal{X}')$ for $\lambda(\mathcal{X}')$ defined in (13) below. In particular, for Massart *halfspaces*, we recover the guarantees of [DGT19].

[7]There is a small subtlety that if the mass of this half is too small, then we will instead split off the other, larger half of the annulus as its conditional error will, by a union bound, be comparable to that of the whole annulus (see Step 14).

[8]We used the implementation in scikit-learn and tuned the max depth differently for the synthetic and numerical experiments, see below

[9]We used the LIBLINEAR library solver in scikit-learn with regularization strength $1/C = N/50$, 200 iterations, $L_2$ penalty, and 0.1 tolerance. We chose to use very little regularization because the datasets we considered are fairly low-dimensional; increasing the regularization makes the logistic regression accuracy decrease.


[Supplementary Material 2 · censormain.pdf]



*Figure: Plots of accuracy versus noise rate $\eta$ for three adversaries across different subgroups.*

Top row (adv 0): "adv 0: overall", "adv 0: >50K aframer", "adv 0: >50K, not aframer"

Middle row (adv 1): "adv 1: overall", "adv 1: >50K female", "adv 1: >50K, not female"

Bottom row (adv 2): "adv 2: overall", "adv 2: >50K immigrant", "adv 2: >50K, not immigrant"

y-axis: accuracy

x-axis: noise rate $\eta$

Legend:
- rf
- lreg
- our
- rcn

[Supplementary Material 3 · noncensormain.pdf]



adv 0: overall | adv 0: >50K aframer | adv 0: >50K, not aframer

adv 1: overall | adv 1: >50K female | adv 1: >50K, not female

adv 2: overall | adv 2: >50K immigrant | adv 2: >50K, not immigrant

accuracy

noise rate $\eta$

rf
lreg
our
rcn