[Reviews · NeurIPS 2020]

Review 1

Summary and Contributions: This work studies learning under Massart noise, following up on the recent breakthrough algorithm for learning halfspaces by Diakonikolas et al. In this work, a vastly simpler proper halfspace learning algorithm is proposed, as well as algorithms for learning generalized linear models under Massart noise. These algorithms can only guarantee accuracy up to the (maximum) noise rate. This work also presents a connection between learning under Massart noise and (correlational) statistical query algorithms/evolvability, and shows that no distribution-independent algorithm that makes only a polynomial number of queries can match the error rate of the optimal halfspace (which may be lower than the noise rate). Finally, some experiments are presented, demonstrating that different (Massart-style) noise rates on different populations can lead standard classifiers to produce different error rates across populations, but that the algorithm presented here (along with the uninterpretable random forest) is resilient to this effect.

Strengths: This is a bundle of several results, each of which are pretty interesting in their own right. The main algorithm for learning halfspaces is a pretty strong result. It is vastly simpler than the Diakonikolas et al. algorithm, provides a single halfspace as a hypothesis, and actually evades a negative result of Diakonikolas et al. showing that minimizing a fixed surrogate loss cannot succeed. The extension to GLMs and the connection to CSQ/evolvability, used to obtain the lower bound for matching OPT (< noise rate), are indeed also quite nice. Finally, the discussion of the connection to algorithmic fairness is interesting.

Weaknesses: My only complaint is that the connection to fairness could have been investigated a little more thoroughly. A single effect resulting from explicit Massart-like noise is presented in the experiments, but the introduction suggests some broader motivation -- I am assuming that the suggestion is that the misreporting should be viewed as Massart noise, but I am not sure if the misreporting due to, e.g., low levels of trust should be viewed as Massart-style noise. Likewise, in the strategic classification example, I am not sure if Massart-style noise is a good model. I would think that to support these kinds of claims, you should demonstrate that the Massart-resilient learner achieves for example, more consistent accuracy across groups in the kinds of data where these effects have been demonstrated. It is an intriguing suggestion, but somewhat underdeveloped here.

Correctness: The body of the paper is somewhat scant on details. A claim about distillation of improper hypotheses to proper hypotheses is made in the introduction and not discussed in any further detail in the body of the paper. It's hard to judge the correctness of most of the work based on what's written in the body; it's plausible but not truly convincing. I recognize that the proofs do appear in the supplemental material, and I recognize that the page limit does not permit the authors to present the work as they'd wish. Still, I would hope for more discussion of what enables the main claims to be obtained.

Clarity: Some parts of the paper are very well written and some parts are rough. I appreciated the overview of the proper halfspace learning algorithm and the connection to the CSQ model. The section on learning GLMs gives a fair overview, but it's a bit too high level and vague at times, to take much away (continuing my complaint above). Likewise, the discussion of fairness could have benefitted from a more careful discussion of how exactly Massart-resilient learning protects against the effect described at the top of p.8. I think this paper would have benefitted from giving a more thorough presentation of more limited claims, perhaps even from being split into two. The results seem strong enough to have supported more than one submission.

Relation to Prior Work: Yes.

Reproducibility: Yes

Additional Feedback: Update post-response: I cannot envision a world in which this paper is rejected on the basis of my relatively low score of 7. But, I really do believe that this should have been split in two companion papers, perhaps with the text of the first paper included in the supplemental material of the second. It does not matter that the algorithm for GLMs shares ideas with the proper learning algorithm; obviously, one often builds upon the ideas of other, prior works. I stress again, it's hard to get much out of the presentation here, to the point that the paper is more "planting a flag" on the result than communicating the ideas involved.


Review 2

Summary and Contributions: The paper extends the recent work of Diakonikolas et al on learning halfspaces with Massart noise. In particular the authors provide an algorithm that properly learns halfspaces under Massart noise. Furthermore, they provide an efficient algorithm for generalized linear models under a noise model that yields Massart noise as a special case. The authors also provide a super-polynomial lower bound on the number of queries needed for learning halfspaces distribution-independently under Massart noise. Finally, the authors evaluate empirically their work on the UCI Adult dataset.

Strengths: This seems to be a very interesting paper with connections to different topics in learning theory, including evolvability and fairness. The algorithms for learning halfspaces, as well as learning generalized linear models are the main contributions of the paper. The paper is relevant to the NeurIPS community (though it appears to be a more computational learning theory type of paper), and there is an extensive appendix (which I must admit I only skimmed through some claims and proofs) that defines the different topics and provides the details to the various claims.

Weaknesses: The only weakness that I can think of is the fact that learning halfspaces under Massart noise has already been done by Diakonikolas et al. last year -- though improperly. Here the contribution is that learning halfspace can be done properly. Of course, in the course of providing this proper learning result the authors also shed light to issues and questions that remained open from the work of Diakonikolas et al; therefore, in the end, the whole approach is definitely worth it.

Correctness: There is an extensive appendix with the claims and the proofs. I skimmed through several of them and it appears to be ok. Similarly, the empirical methodology appears to be correct.

Clarity: The paper is well-written and this appears to be the case in the appendix, to the extent that I read some parts. I only have a few remarks. Line 19: "hypothesis" --> ground truth function Lines 39-42: The Massart noise model was known/used earlier. I think the correct reference is Robert Sloan's paper "Four types of noise in data for PAC learning" where the noise model was introduced and is called as the "malicious misclassification noise model" there. Line 155: D is defined to be the distribution in line 169. This should be done here as well.

Relation to Prior Work: Yes, the paper positions well its results with respect to the related work.

Reproducibility: Yes

Additional Feedback: After rebuttal: Thank you for a very interesting paper.


Review 3

Summary and Contributions: This makes the recent breakthrough results on learning halfspaces proper and also gives an SQ lower bound for getting OPT + \epsilon.

Strengths: Clearly a great result. A very important improvement on recent work due to Diakonikolas et al. Well written. Several clever ideas and an interesting tie-in to evolvability.

Weaknesses: None

Correctness: Yes

Clarity: Yes

Relation to Prior Work: Yes

Reproducibility: Yes

Additional Feedback: A must accept.


Review 4

Summary and Contributions: The paper extends results on learnability with Massart noise. While previous work settled the open question of learnability of halfspaces by showing an improper learner exists, this work provides a proper learner. Another extension this work offers is to the class of Generalized Linear Models, where an improper learner for GLMs under Massart noise is given. Finally, a lower bound is given for the number of Correlation Statistical Queries required for learning halfspaces under Massart noise, using a nice connection to previous work on Correlational Statistical Queries.

Strengths: The paper provides insights into interesting questions that are relevant to the learning theory community, and are receiving increased attention following the developments in last year's paper about learning with Massart noise. Technically, the proofs seem to contain several novel developments over the work of Diakonikolas et al. The results on GLMs and connections to Statistical Queries are also novel and significant. Although I am not highly knowledgable on these specific aspects of learning theory, the paper looks like a solid contribution to me.

Weaknesses: I did not find very clear weaknesses with the paper, although I do think it may be possible to discuss the connection between its different parts in more detail. For instance, It would be nice to give a few words about why the knowledge distillation approach from the paper is inapplicable to get a proper learner for GLMs. Or more broadly, are there any technical insights that the analysis in the paper gives towards resolving the question of proper learning in GLMs? In my humble opinion, a small table summing up the known bounds under the different settings, or a discussion at the end of the paper, can also be helpful for readers who want to get an overview of the results and open issues.

Correctness: The claims are correct and backed up by rigorous proofs.

Clarity: For a paper that is rich in technical details, the paper is not hard to read. As pointed out earlier, I think that it could be useful to have a condensed overview of the results and the conclusions.

Relation to Prior Work: Relation to prior work is discussed adequately.

Reproducibility: Yes

Additional Feedback: Small comments: 1) Equation numberings and hyperlinks seem to be malfunctioning (e.g. I couldn't find any of the numberings on the equations). 2) In line 131 it is stated that \sigma=0 is essentially the same as agnostic learning. While I can see why this makes sense, it doesn't look like an entirely trivial statement (unless I am missing something here). It might be good to consider adding a short comment about that.

[Author Response · NeurIPS 2020]

**Author Response: Classification Under Misspecification: Halfspaces, Generalized Linear Models, and Evolvability**

We would like to thank all the reviewers for taking the time to understand the contributions of our paper, and for their helpful comments and/or suggestions. We do not have much to add, but would just like to emphasize a few points in response to the reviews.

We think our contribution relative to the breakthrough work of Diakonikolas et al. is not just that our algorithm is proper or that the insights behind it lead to algorithms for more general concept classes, but that by avoiding partitioning the domain into a polynomial number of regions, it actually becomes practical and something that we can run on real data.

We think that the experimental results are striking, but still only a proof-of-concept in the sense that we added the noise to the data ourselves. A truly compelling demonstration would be, like the first reviewer said, if we could find some real data where our algorithm works well and is demonstrably more fair. This is a direction we are actively pursuing, but we feel that it is a substantial project and would likely be a separate paper if it is successful. Nevertheless some works in fairness work with a graphical model whose causal structure produces confounding effects that lead off-the-shelf algorithms to produce unfair decision rules, see e.g. Kusner et al. [2017]. These types of models naturally lead to situations where noisy observations of some latent quality score might be more variable for some demographics than for others.

Also, we agree that it is hard to do justice to all the technical ingredients in just 8 pages. We attempted to give a more detailed outline for our proper learner, and just some of the key ideas for GLMs. It is an interesting suggestion that we could have split it into two papers. However the results actually build on each other, e.g. our algorithm for GLMs in the $\zeta = 0$ case depends on some knowledge distillation primitives which in turn use our proper learning algorithm for halfspaces.

# References

Matt J Kusner, Joshua Loftus, Chris Russell, and Ricardo Silva. Counterfactual fairness. In *Advances in neural information processing systems*, pages 4066–4076, 2017.


[Meta-Review · NeurIPS 2020]

This paper makes a solid contribution to the literature on efficiently learning halfspaces with noise, extending a paper from last year on learning with Massart noise by proposing a simpler and proper learning algorithm. It also generalizes beyond Massart noise to some extent, and establishes a lower bound for SQ learning. The reviewers are unanimous in their praise of the paper.